# Segmented filamentous bacteria are worldwide human gut commensals

Shashi Kiran [1,15], Ana Raquel Cruz [1,15], Alice Daniau [1], Bing Ma [2], Martial Marbouty [3], Juliana Pipoli Da Fonseca [4], Agnès Legrand [1], Lyam Baudry [3], Thomas Cokelaer [4,5], Matthieu Bensussan [6], Maryse Moya-Nilges[7], Julian Garneau [8], Marc Monot [4], John B. Ochieng [9], Martin Antonio [10], Boubou Tamboura [11], Willem M. de Vos [12,13], Anne Salonen [12], Jahangir Hossain [10], Richard Omore [9], Samba O. Sow [11], Philippe J. Sansonetti [14], Jacques Ravel [2], Nadine Cerf-Bensussan [6] & Pamela Schnupf [1,15] ✉

Segmented filamentous bacteria (SFB) describe morphologically similar gut commensals found in mammals, fish and birds. In mice, SFB intimately colonizes the ileal epithelium at the time of weaning and elicits a strong pleiotropic immune activation that fosters colonization resistance while augmenting disease severity in various disease models. SFB is therefore critical in both health and disease but information regarding SFB in humans remains limited. Here, we first identify and characterize a human SFB species with SFB-specific morphology, including the hook-like tip structure that mediates attachment, and unique genome features, including a starch and glycogen degradation module. This species, which we name Anisomitus miae and establish as the nomenclature type for the SFB genus, is within a SFB lineage common across Africa. We then bioinformatically identify, based on the 16S rRNA gene V3-V4 variable region sequence, four major, and two minor, human SFB lineages in forty-four countries distributed across all six inhabited continents. We provide evidence towards the co-colonization potential of the SFB lineages and their colonization dynamics, including a potent but short-lived colonization peak in children between one to five years of age. This study establishes the presence of multiple SFB species in the human population and SFB as a minor but widespread group of commensals in humans.

SFB are aerotolerant, spore-forming Clostridia-related intestinal anaerobes present in the gut microbiota of diverse phylogenetic groups such as fish, birds, and mammals[1]. SFB remain difficult to study and enigmatic due to a general lack of available tools and limited in vitro culturing conditions[2]. Best characterized in the mouse model, where SFB can be propagated under monocolonization conditions in germfree mice, SFB is uniquely immunostimulatory with a particularly strong stimulation of intestinal Th17 and IgA responses, and a substantial effect on the intestinal homeostatic immune cell composition and reactivity[3]. Through intimate attachment to the ileal epithelium, SFB can, by itself, largely recapitulate the immunostimulatory potential of a complex microbiota in the innate and adaptive immune compartments[4–8]. SFB is a strong inducer of the weaning reaction that prevents pathological imprinting and increased susceptibility to colitis, allergic inflammation, and cancer[9]. SFB is also one of the foremost commensals associated with colonization resistance against

pathogens and has been linked to the protection of mice against bacteria, viruses, and protozoa in the gut[6,10,11], as well as protection from bacterial, viral, and fungal infection in the lung[12–14]. The effect of SFB on the host immune system is, however, complex, and SFB can augment the severity in several murine models of disease[15–20]. In mice, SFB is therefore a key gut commensal critical in both health and disease.

SFB has a complex life cycle. Teardrop-shaped unicellular SFB, called intracellular offsprings (IOs), attach via an unusual tip structure[21] to the absorptive epithelial cells and cells overlying Peyer's patches in the ileum in a species-specific manner[2,22,23]. IOs grow out into filaments that divide and differentiate into new IOs that are released from the distal end of the filament or may form spores prior to their release[24–26]. SFB numbers increase rapidly around the late weaning period but decrease again after a short colonization peak as the host immune system matures and the gut microbiota becomes more complex[23,27,28]. Due to the difficulties of growing and manipulating these bacteria, the complete genome sequence of SFB is still limited to three mouse SFB and one rat SFB[29–31]. Incomplete genome sequences are available for additional mouse SFB, turkey SFB and human SFB from Sweden (Human-SFB-SE)[32–35]. SFB have a reduced genome size of ~1.5–1.7 Mb[32,34] and are metabolically unique symbionts that, based on their reduced number of predicted coding sequences (CDS)[35] and metabolic capabilities[34], lie in between obligate and facultative bacterial symbionts, underscoring their intimate relationship with the host despite their limited penetrative capacities.

Evidence towards the existence of SFB as part of the human gut microbiota remains sparse. SFB have been detected in human fecal and intestinal samples in a few countries[36–43], albeit most often through the use of PCR of the 16S rRNA gene with SFB-specific primers, which can be error-prone[44]. Currently, the strongest genomic evidence for the existence of SFB in humans is the draft SFB genome sequence from Sweden, reported in 2020[33], but searches for this genome in metagenome databases have yielded few results[33] and a link between the genome sequence and the bacterium's morphology has not been established. The effect of human SFB on the host immune system also remains unclear. Higher Th17 and IgA responses in Chinese children have been correlated with higher levels of SFB, based on 16S rRNA gene amplification and proteomics[38]. However, morphological analysis through fluorescent in situ hybridization revealed either no filaments[36] or filaments but also bacteria with a rod shape instead of the characteristic teardrop-shaped SFB morphology[38]. Higher Th17 and IgA responses in Chinese children have also been correlated with higher levels of SFB-like bacteria attached to the ileum; however, 16S rRNA gene amplicon analysis did not identify a candidate for the adhesive bacteria with the SFB-like morphology[45]. Scanning electron microscopy has, for now, not identified the characteristic tip structure[38,45,46]. At the same time, associations of SFB with medical pathologies such as inflammatory bowel disease and depression are slowly emerging[37,39,41,42], albeit predominantly through the use of SFB-specific PCR.

Here, we identify and characterize an SFB species from the fecal material of two Malian children on the morphological and genomic level. This species, here named *Anisomitus miae* and referred to as Human-SFB-ML, displays the characteristic SFB morphology, including the tip structure involved in attachment, a thin-and-smooth to thick-and-bulbous filament transition, and spore formation. The Human-SFB-ML genome shares a core genome with other SFB species from mouse, rat, turkey, and human. This links the SFB-specific morphology of Human-SFB-ML to its genome position within the SFB clade. Beyond the core genome, the Human-SFB-ML genome also includes previously undescribed features such as a module for the digestion of glycogen and starch and the uptake of maltooligosaccharides, as well as additional features potentially related to the protection against environmental stresses. Through bioinformatic analysis of metagenomic data, we find evidence of the Human-SFB-ML genome in fecal samples of

individuals from numerous African countries, including indigenous cultures, thereby pointing towards an ancient association. When we extend the bioinformatic analysis to publicly available 16S rRNA gene amplicon datasets of the V1-V2 and V3-V4 variable region, we identify the presence of four major and at least two minor SFB lineages worldwide. Besides Human-SFB-refML and Human-SFB-refSE, the other SFB lineages are phylogenetically close to SFB from mouse, rat and chicken and we find supporting genomic evidence for the sharing of a potentially closely related SFB species between humans and turkey in a human paleofecal sample. Metadata analysis of the 95 SFB-positive bioprojects identified, and particularly time course studies in children, reveals a strong but only ~1 month-long colonization peak of SFB in children within the first 5 years, but particularly between 12 and 24 months of age. In addition, we bioinformatically identify the co-colonization of different SFB lineages within an individual and provide support for colonization dynamics similar to what has been described for mouse SFB. Together, we thereby establish the existence of a bacterial species with SFB-specific morphology and genome sequence in humans and show that multiple SFB species and at least six SFB lineages are associated with the human gut microbiota. In addition, we find that the SFB lineages are minor constituents of the fecal microbiota, while numerous studies, and particularly time-course experiments, ultra-deep sequencing studies, and studies with intestinal biopsies, reveal a high prevalence of SFB in many countries and in all inhabited continents.

## Results

### Identification of a human SFB species in Africa

The nature and extent of SFB in the human population remain unclear. Using bioinformatic analysis and nucleotide similarity to the Mouse-SFB-NL full-length 16S rRNA gene sequence, we identified potential SFB-positive fecal samples in children living in Mali, Kenya, and The Gambia in a study of young children with dysentery (Supplementary Fig. 1a–d)[47]. To verify and further characterize this finding, we acquired both SFB 16S rRNA gene amplicon-positive and negative fecal samples from the Gambia (GM), Kenya (KE), and Mali (ML)[47]. Using Pacbio sequencing, an identical full-length 16S rRNA gene sequence, which matched the previously identified 16S rRNA gene amplicon reads, was obtained in all four SFB-positive fecal samples from the Gambia, Kenya, and Mali, but not in fecal samples negative for SFB. To identify SFB 16S rRNA gene reference genes, 16S rRNA gene sequences that covered at least the V1-V4 region and showed at least a 92% nucleotide identity to the Mouse-SFB-NL 16S rRNA gene were identified in the NCBI nucleotide database using the V1-V4 region of the Mouse-SFB-NL 16S rRNA gene sequence as the query. All 16S rRNA gene sequences identified fell within the SFB clade on a maximum likelihood phylogenetic tree (Supplementary Fig. 2). The SFB reference sequences retained from various hosts covered the V1-V9 variable regions, unless not available, in which case representative sequences covering the V1-V4 variable regions were retained for subsequent analysis. These 16S rRNA gene reference sequences of the SFB clade are here referred to as "host"-SFB-"geographic origin of host" (Supplementary Fig. 3). As expected, the African human SFB 16S rRNA gene sequences fell within the SFB clade when using as an outgroup four *Clostridium* species representing different *Clostridium* clades (Fig. 1a) or the closest bacterial species to the SFB clade in the EzBioCloud (Supplementary Fig. 4a) or Silva 16S rRNA gene databases (Supplementary Fig. 4b). The African human SFB clustered with the 16S rRNA gene sequence of pigs, albeit with only a 96.2% nucleotide identity (Fig. 1b, Supplementary Data 1a). This clustering was broadly maintained when considering only the V1-V2 region (Supplementary Fig. 5), which is highly variable (Supplementary Fig. 6), while the African SFB clustered most closely to chicken SFB when considering only the V4 or the V3-V4 regions (Supplementary Fig. 5b, c, Supplementary Data 1). The V1-V2 16S rRNA variable region is therefore a better proxy for the full-length 16S rRNA gene in SFB phylogenetic

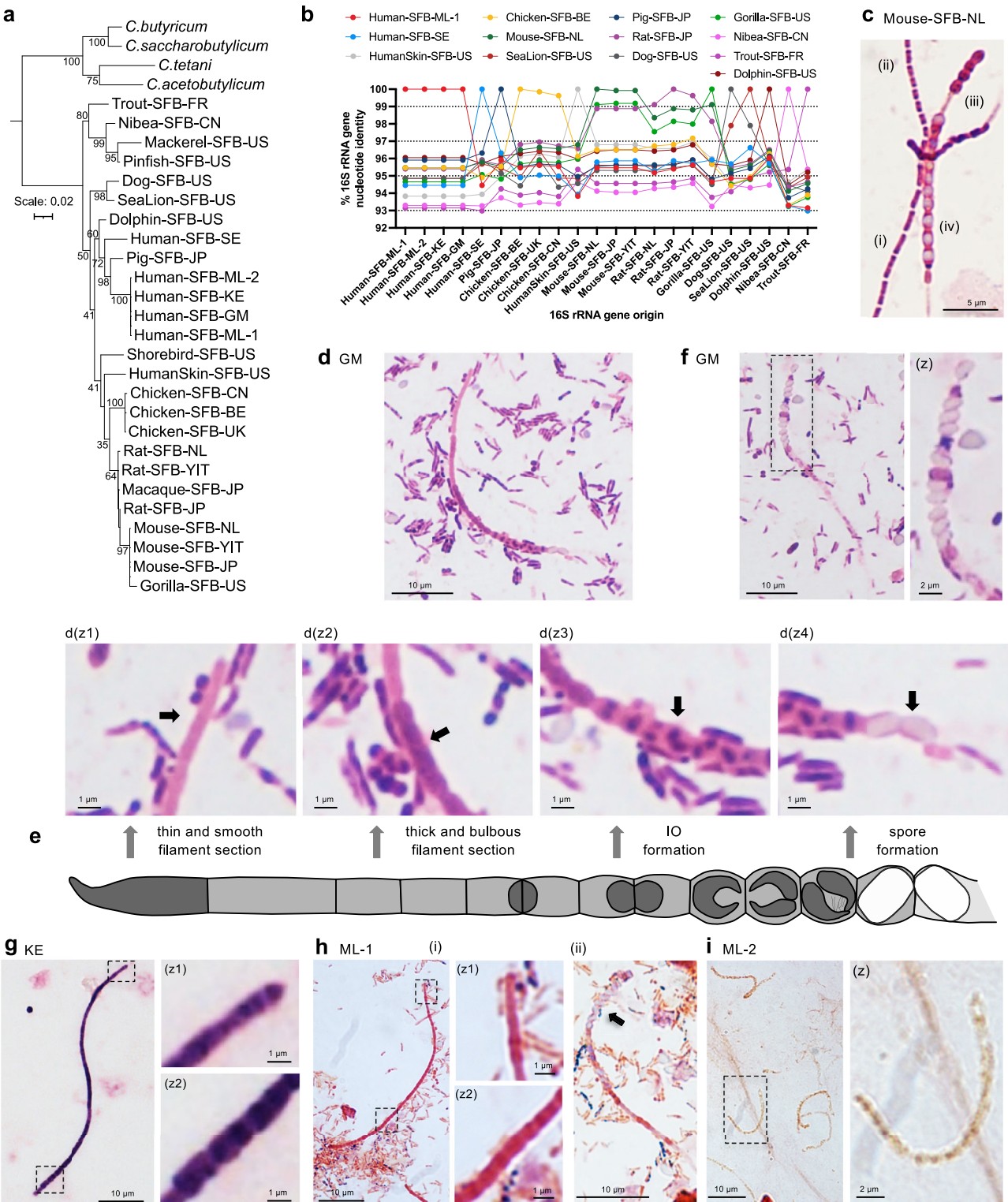

**c** Mouse-SFB-NL

**d** GM

**f** GM

**d(z1)** thin and smooth filament section

**d(z2)** thick and bulbous filament section

**d(z3)** IO formation

**d(z4)** spore formation

**e**

**g** KE

**h** ML-1

**i** ML-2

analysis as compared to the other variable regions. The African human SFB 16S rRNA gene sequences, referred to as Human-SFB-ML hereafter, share less than 97% nucleotide identity with all other SFB 16S rRNA gene sequences of human or animal origin (Fig. 1b, Supplementary Data 1), establishing it as a new SFB species[48].

The 16S rRNA gene sequence of Human-SFB-ML and the human SFB identified in Sweden (Human-SFB-SE) are markedly different at only a 94.7% nucleotide identity (Fig. 1b, Supplementary Data 1a). A similar pattern of low nucleotide identity is seen for the other 16S rRNA gene sequence of human origin, HumanSkin-SFB-US, from the skin of

an American child[49]. The low nucleotide identity between SFB 16S rRNA gene sequences of human origin, and particularly within the regions targeted by popular SFB primer sets (for example, F799/R1008), has important implications for the detection of SFB using PCR-based methods (Supplementary Fig. 7). As such, detection of different human SFB necessitates modified primer sequences.

The low 16S rRNA gene sequence identity of human SFB from Africa and Sweden was unexpected and contrasts with the high sequence identity shared between SFB from the same animal hosts of geographically distant regions, such as Mouse-SFB (Netherlands vs

**Fig. 1 | Identification and characterization of a human SFB species found in Africa.** Analysis of fecal samples from bioproject PRJNA234437. **a** Maximum likelihood phylogenetic tree of 16S rRNA gene sequences from SFB from various hosts; includes *Clostridium* outgroup species (italic). Tree includes bootstrap values, and the scale is nucleotide substitutions per nucleotide position. Sequences are trimmed to the 16S rRNA gene 1470 bp position but includes shorter sequences, including only the V1-V4 region for mackerel, pinfish, macaque, and shorebird (Supplementary Fig. 3). **b** Percent nucleotide identities across SFB 16S rRNA gene sequences of 1366 bp in length from various hosts (Supplementary Fig. 3). **c** Gram stain of Mouse-SFB-NL from the intestinal content of monocolonized mice. Highlights SFB filament sections that are **c**(i/ii) thin and smooth with **c**(i) long primary and **c**(ii) short secondary segments, as well as **c**(iii) thick and bulbous sections characteristic of differentiation, and **c**(iv) a filament section containing spores. **d** Gram stain of The Gambia (GM) sample 102358 from a non-diarrheal control group child showing a complete filament and zooms (z) to the (z1) smooth filament end; (z2) smooth to bulbous transition; (z3) heavily de-stained bulbous segments

towards filament end with crystal violet staining reminiscent of intracellular offsprings; and (z4) white unstained spores at filament end. (z1-z4) Arrows highlight the various features. **e** Schematic representation of inferred SFB filament stages. **f** Gram stain of a disintegrating filament in The Gambia sample 102358 with large oval spores at filament end, including a zoom to highlight spores. **g** Gram stain of the Kenya (KE) sample 401080 from a non-dysentery diarrheal group child showing a filament with a zoom (z1) to the thin, apparently tip-like, end and (z2) to the bulbous filament end. **h** Gram stains of the Mali (ML) sample 200340 (S340, Human-SFB-ML-1) from a non-diarrheal control group child, including **h**(i) a filament with a zoom to the characteristically (z1) thin and (z2) bulbous phenotype of SFB, and **h**(ii) a filament with spores at one end, highlighted with a black arrow. **i** Gram stain of the Mali sample 200195 (S195, Human-SFB-ML-2) from a non-dysentery diarrheal group child showing filamentous bacteria and a zoom (z) of one filament. **a, b, d–i** Labeling includes the two-letter country codes of the sample of origin, except YIT, which also originated from Japan. Images in (**d**) and (**g**) include increased bilinear interpolation for clarity.

Japan, 99.9%), Rat-SFB (Netherlands vs Japan, >99.1%), and Chicken-SFB (Belgium versus the United Kingdom versus China, >99.8%) (Fig. 1b, Supplementary Data 1). Even within more distantly related organisms such as mouse and rat, the SFB 16S rRNA gene sequence nucleotide identity is relatively high at over 98.2% (Fig. 1b, Supplementary Data 1). Meanwhile, SFB 16S rRNA gene sequences from phylogenetically distant hosts generally share a low (-91-96%) nucleotide identity (Fig. 1b, Supplementary Data 1). However, notable exceptions are the 99.1% nucleotide sequence identity shared between mouse SFB and the available 1366 bp of Gorilla-SFB-US, and the 99.8% nucleotide sequence identity shared between rat SFB and the available 970 bp of Macaque-SFB-JP (Supplementary Fig. 3, Supplementary Fig. 6, Supplementary Data 1)[50], suggesting the sharing of a similar or the same SFB species in multiple hosts. For human SFB, these data show that multiple species within the SFB clade exist within the human population and validate our workflow and criteria used to bioinformatically identify SFB-positive samples in publicly available 16S rRNA gene amplicon sequence deposits.

### Human-SFB-ML is morphologically similar to SFB in mice

To characterize SFB on a morphological level, both SFB-positive and negative human fecal samples from The Gambia, Kenya, and Mali[47] were first analyzed using the Gram stain. Filaments with SFB-specific morphology at varying stages of differentiation, similar to what is seen for Mouse-SFB-NL under monocolonization conditions in mice (Fig. 1c, Supplementary Fig. 8), were present in SFB-positive samples of all three African countries (Fig. 1d, f–i, Supplementary Fig. 9 and Supplementary Fig. 10). Long filaments (normally >30 μm) with the thin-and-smooth to thick-and-bulbous filament transition, a characteristic for the differentiation of filament segments into unicellular IOs[25,26,51], were more readily identified in fecal samples that, based on 16S rRNA gene amplicon analysis, had a higher SFB relative abundance. SFB-like filaments were never observed in the eight human fecal samples that were SFB-negative by 16S rRNA gene amplicon analysis. In stainings of the Gambia fecal sample (Fig. 1d, f, Supplementary Fig. 9a), which was subject to excessive de-staining during the procedure leading to the removal of the crystal violet from the filament cell wall, filaments showed many morphological features (Fig. 1e) characteristic of SFB in mice (Fig. 1c, Supplementary Fig. 8) including (Fig. 1d (z1–4)): (z1) a smooth and thin filament morphology at one end that includes an apparent tip; (z2) a thin and smooth to thick bulbous filament transition; (z3) a thick bulbous end displaying within a bulbous segment staining reminiscent of teardrop-shaped IOs in their non-septated state[52], and (z4) stain-resistant spores at the disintegrating filament end. Spores, which were also found in other disintegrating filaments (Fig. 1f, Supplementary Fig. 9a (iii)), were, at -0.9 μm by 1.9 μm, larger and more rectangular than the -1.0 μm in diameter spores observed in mice[51] (Fig. 1c, Supplementary Fig. 8b, c, e, f) and rats[25], two host species

in which IO formation has been shown to precede spore formation[25,51]. A similar thin to bulbous biphasic morphology was present in the Kenya diarrheal sample, including an apparent tip-like morphology at the thin end (Fig. 1g, Supplementary Fig. 9b (ii)z1, (iii)z)). Filaments of the Mali non-diarrheal fecal sample (ML-1) likewise showed the thin and bulbous end phenotypes (Fig. 1h (i), Supplementary Fig. 10a) and a large spore content (Fig. 1h (ii)), shown at higher magnification in Supplementary Fig. 10a (i)). Conversely, the second Mali fecal sample (ML-2), derived from a child with a non-dysentery diarrhea, contained segmented filaments that appeared damaged and stained weakly with the Gram stain (Fig. 1i, Supplementary Fig. 10b) but could also include spores (Supplementary Fig. 10b (iii, v)). The two fecal samples from the Mali cohort were chosen for further analysis as they came from individuals of the same region, thereby serving to increase the rigor of the analysis and demonstrate reproducibility.

In the non-diarrheal sample Human-SFB-ML-1, fluorescent in situ hybridization (FISH) with a commonly used pan SFB-specific probe[53,54] (Supplementary Fig. 3) identified filaments displaying the smooth to bulbous morphology transition (Fig. 2a). In addition, specific staining was observed for a long uneven filament (Fig. 2b (i, ii)), a possibly broken filament remnant (Fig. 2b (i, ii), Supplementary Fig. 10c), and small bacteria with an apparent teardrop-shape (Fig. 2b (i)(z1, z2)), reminiscent of IOs (Supplementary Fig. 8a (z1), b (z1)) in the dense aggregate of bacteria (Supplementary Fig. 10d). These phenotypes are similar to what is described for SFB FISH in chickens[53]. Scanning electron microscopy (SEM) analysis validated the qualitative difference in SFB filaments of the two Mali samples, the characteristically segmented morphology of the filaments (Fig. 1c–e), and the characteristic SFB tip structure that in SFB from other vertebrate hosts is involved in host attachment (Fig. 2c, d)[25,51,55–57]. The smooth and thin bacterial filament end harboring tips were -0.7 to 1.0 μm in width (Fig. 2c), as reported for murine SFB[25,51]. In addition, SEM showed the smooth to bulbous morphological transition within a single filament (Fig. 2c (ii), Fig. 2e (i, ii)), heterogeneity in the bulbous sections (Fig. 2e (iii)), and the particularly pronounced bulbous phenotype reminiscent of spore-containing filaments (Fig. 2e (iv)), as seen, using the Gram stain, in the same fecal sample (Fig. 1h, Supplementary Fig. 10a (i)) and the Gambia fecal sample (Fig. 1f, Supplementary Fig. 9a (iii)). These data show that the characteristic SFB morphology is present in filamentous bacteria from The Gambia, Kenya, and Mali. Confirmation of the unusual tip structure for Human-SFB-ML furthermore strongly supports a similar life-cycle of Human-SFB-ML as SFB from mice, including intimate attachment to the intestinal epithelium.

### Human-SFB-ML genomes are more closely related to turkey SFB than to rodent SFB or Human-SFB-SE

We next aimed to obtain the SFB genome sequence from the two Malian fecal samples. Using chromosome conformation capture (3 C),

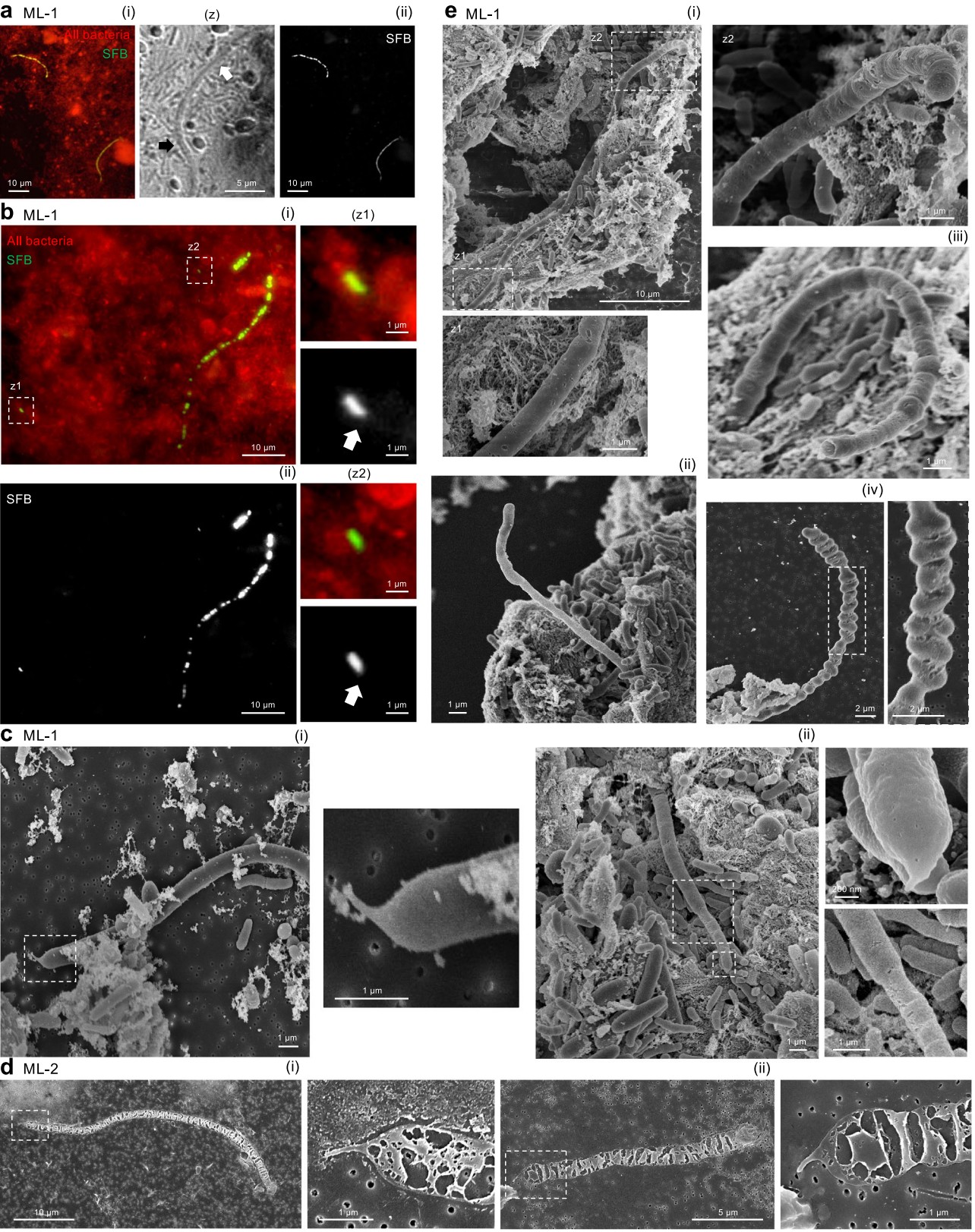

two draft human SFB genome sequences were obtained that showed relatedness to the SFB genomes using a Blast Score Ratio (BSR) analysis[58], a sensitive method based on the protein sequence coverage and amino acid identity score (Supplementary Fig. 11, Supplementary Table 1a). SFB genomic DNA in the Malian fecal samples ML-1 (S340) and ML-2 (S195) was then captured using a pan-SFB genome RNA bait set and sequenced using both Illumina and PacBio technologies. De novo assembly of Illumina reads, combined with guided assemblies using PacBio sequences, was applied for a final assembly yielding the human SFB genome sequence of Human-SFB-ML-1 in 13 contigs and Human-SFB-ML-2 in 11 contigs (Supplementary Table 1b). The genomes of Human-SFB-ML-1 and ML-2 have a CheckM v1.1.0[59] completion/contamination prediction of 99.85%/0.62% and 99.74%/1.02%, respectively.

**Fig. 2 | Morphological analysis of SFB found in Malian children. a, b** Fluorescent in situ hybridization of the ML-1 fecal sample showing staining with **a, b**(i) the SFB-specific (green) and eubacterial (EUB338) (red) 16S rDNA probes or **a, b**(ii) only the SFB-specific eubacterial probe in black and white. **a** Image with two SFB filaments, including a zoom (z) in brightfield highlighting the smooth (white arrow) and bulbous (black arrow) morphology of the filament. **b** Image with an SFB filament of uneven thickness, a potential SFB filament fragment, and two small SFB highlighted (z1/z2) in both color and, for the SFB probe only, in black and white. The small SFB has an apparent teardrop-shaped morphology (white arrows). Images in (z1/z2) include increased bilinear interpolation for clarity. **c** SEM images **c**(i/ii) of the Human-SFB-ML-1 (non-diarrheal) fecal sample showing the smooth end and of a filament with zooms to the characteristically SFB tip structure, and for **l**(ii) also a zoom of filament thickening. **d** SEM images (i/ii) of the Human-SFB-ML-2 (diarrheal) fecal sample showing disintegrating filaments with zooms of the tip structure. **e** SEM images of the Human-SFB-ML-1 fecal sample showing **e**(i/ii) filaments with a contrasting smooth and bulbous morphology along the filament with zooms of opposite filament ends included for (**e**(i)); **e**(iii) a filament with a characteristic segmented phenotype of SFB, and **e**(iv) a filament, including a zoom, with an irregular and large bulbous segmented phenotype similar to the spore-containing filament segment in the Gram stains of Fig. 1f and Fig. 1h(ii).

The Human-SFB-ML genomes are similar in size to SFB from other vertebrate hosts (Supplementary Table 1b). With only a limited number of genome sequences available, phylogenetic analysis of the SFB core genome places the Human-SFB-ML genomes closest to Turkey-SFB-US (Fig. 3a). A similar relationship was obtained using a phylogenetic analysis of the RNA polymerase subunits RpoB and RpoC (Supplementary Fig. 12a, b) and average amino acid identity (AAI) (Supplementary Fig. 12c). Using whole-genome average nucleotide identity (ANI), bacteria of the same species generally show an ANI of greater than 95%, while different species are below 83%[60]. The Human-SFB-ML genomes, which share a high sequence identity at 99.5% ANI with each other, share only a ~78.4% ANI with turkey SFB, and a 77.2 to 77.8% ANI with rodent SFB and Human-SFB-SE (Fig. 3b, Supplementary Table 2a). At the amino acid level, the deduced Human-SFB-ML proteomes share an AAI of ~99.2% with each other but only ~73% with Turkey-SFB-US and ~67% with both rodent SFB and Human-SFB-SE (Supplementary Fig. 12d, Supplementary Table 2b). These data confirm, on the genome level, Human-SFB-ML as a new SFB species and, to our knowledge, identify humans as the first host species harboring multiple SFB species.

The Human-SFB-ML genomes were registered as a new genus and species under the Code of Nomenclature of Prokaryotes Described from Sequence Data (SeqCode)[61] and named *Anisomitus miae* (seq-co.de/r:zlg3gn05). While SFB have long been called Candidatus *Arthromitus* in the literature[62], *Arthromitus* also describes non-SFB bacteria in termites[63]. The name Candidatus *Dwaynsavagella* has been used more recently[64] but, in keeping with a more descriptive name, we chose the genus name *Anisomitus* proposed, in 1925, by Pierre-Paul Grassé to describe an SFB-like organism attached to the intestinal epithelium of the domestic duck and to distinguish this organism from *Arthromitus*-like organisms found in the gut of termites[65].

### Description of Anisomitus gen. nov

*A.ni.so.mi'tus.* Gr. masc. adj. *anisos*, uneven; Gr. masc. n. *mitos*, thread; N.L. masc. n. *Anisomitus*, referring to uneven filamentous form.

Genus of bacteria commonly referred to as segmented filamentous bacteria (SFB); known also as *Candidatus Arthromitus* (Greek for "jointed thread"), as proposed by Snel et al. As *Arthromitus* also describes non-SFB bacteria[63], the genus name *Anisomitus* was chosen in keeping with a morphologically descriptive name for SFB and in recognition of the work by the French zoologist Pierre-Paul Grassé, who described SFB attached to the intestinal epithelium of the domestic duck in 1925[65]. SFB are Gram variable and spore-forming bacteria that, based on 16S rRNA gene sequence analysis, form a monophyletic group within the *Clostridiaceae*. SFB grow from unicellular bacteria of approximately 1 μm in length into filaments reaching over 80 μm in length. Particular characteristics include a hook-like tip structure, present on both the unicellular and filamentous forms, as well as a biphasic filament morphology of thin and smooth to thick and bulbous as the filament ages and unicellular bacteria develop inside the filament to either form spores or to be released from the filament as IOs. The genus is represented by closed genome sequences of SFB from mice[29–31] and rat[30] as well as metagenome-assembled genomes from the gut of humans[33] and birds[32,64,66]. Genomes are reduced and range in size from approximately 1.5–1.7 Mb, lack nearly all components of the TCA cycle, but include genes involved in flagella synthesis and chemotaxis. The nomenclatural type species for this genus is *Anisomitus miae*.

### Description of Anisomitus miae gen. nov. sp. nov

*Anisomitus miae* (*A.ni.so.mi'tus*. Gr. masc. adj. *anisos*, uneven; Gr. masc. n. *mitos*, thread; N.L. masc. n. *Anisomitus*, referring to uneven filamentous form; *mi'ae*. L. gen. fem. n. *miae*, of Mia, daughter). This bacterial species is Gram variable and spore-forming, hybridizes with the 16S rRNA-targeted oligonucleotide probe 5′-GGG TAC TTA TTG CGT TTG CGA CGG CAC-3[54], and has a 16S rRNA gene sequence that clusters within the monophyletic group in the Clostridiaceae that includes SFB from hosts such as the mouse, rat, turkey, chicken, and human. The genome is ~1.6–1.7 Mb in size with a GC content of ~30.6%. This species includes all bacteria with genomes that show ≥95% average nucleotide identity to the type genome of the MAG ID Human-SFB-ML-1 from the biosample SAMN41134099 available at NCBI.

### Human-SFB-ML share unique features with Turkey-SFB-US and Human-SFB-SE as compared to rodent SFB

The Human-SFB-ML genomes have a similar size and number of predicted coding sequences to other SFB species (Supplementary Table 1). The SFB core genome consists of 782 gene clusters or 925 gene clusters when not considering the draft genome of Human-SFB-SE (Fig. 3c). Outside the core genome, the Human-SFB-ML genomes are most similar to each other, exclusively sharing 411 gene clusters when all SFB genomes are considered (Fig. 3c), or 607 gene clusters when compared only to the Human-SFB-SE genome (Fig. 3d). Pangenome analysis using Anvi'o illustrates the presence of the shared core SFB genome (Core, 925 gene clusters) as well as the proportion of gene clusters shared between two to four of the five phylogenetic groups (Cross-phylo, 647 gene clusters), and those within only one phylogenetic group (Phylo-specific, 1315 gene clusters) (Fig. 3e). In addition, the pangenome plot includes an overview of some of the genome features (Supplementary Table 1b) and the combined geometric and functional homogeneity analysis, a measure of the similarity of the genes within the gene clusters. Together these data indicate an overall broadly conserved genome composition of Human-SFB-ML compared to other SFB genomes.

The Human-SFB-ML genomes include expected genes related to the characteristic life-cycle of SFB, such as genes related to sporulation and germination, chemotaxis, and flagella-mediated motility (Supplementary Data 2 and Supplementary Data 3), and have an overall similar number of genes within the different cluster of orthologous gene (COG) functional categories as compared to SFB genomes from other vertebrate hosts (Fig. 3f, Supplementary Data 3). An exception is a lower number of genes associated with carbohydrate (G) and inorganic ion (P) metabolism and transport (Fig. 3f). Within the accessory genes, Human-SFB-ML shares a number of genes with only a subset of the other SFB species; for example Human-SFB-ML shares uniquely only 10 genes with all rodent SFB, 30 genes with Turkey-SFB-US, 38 genes with

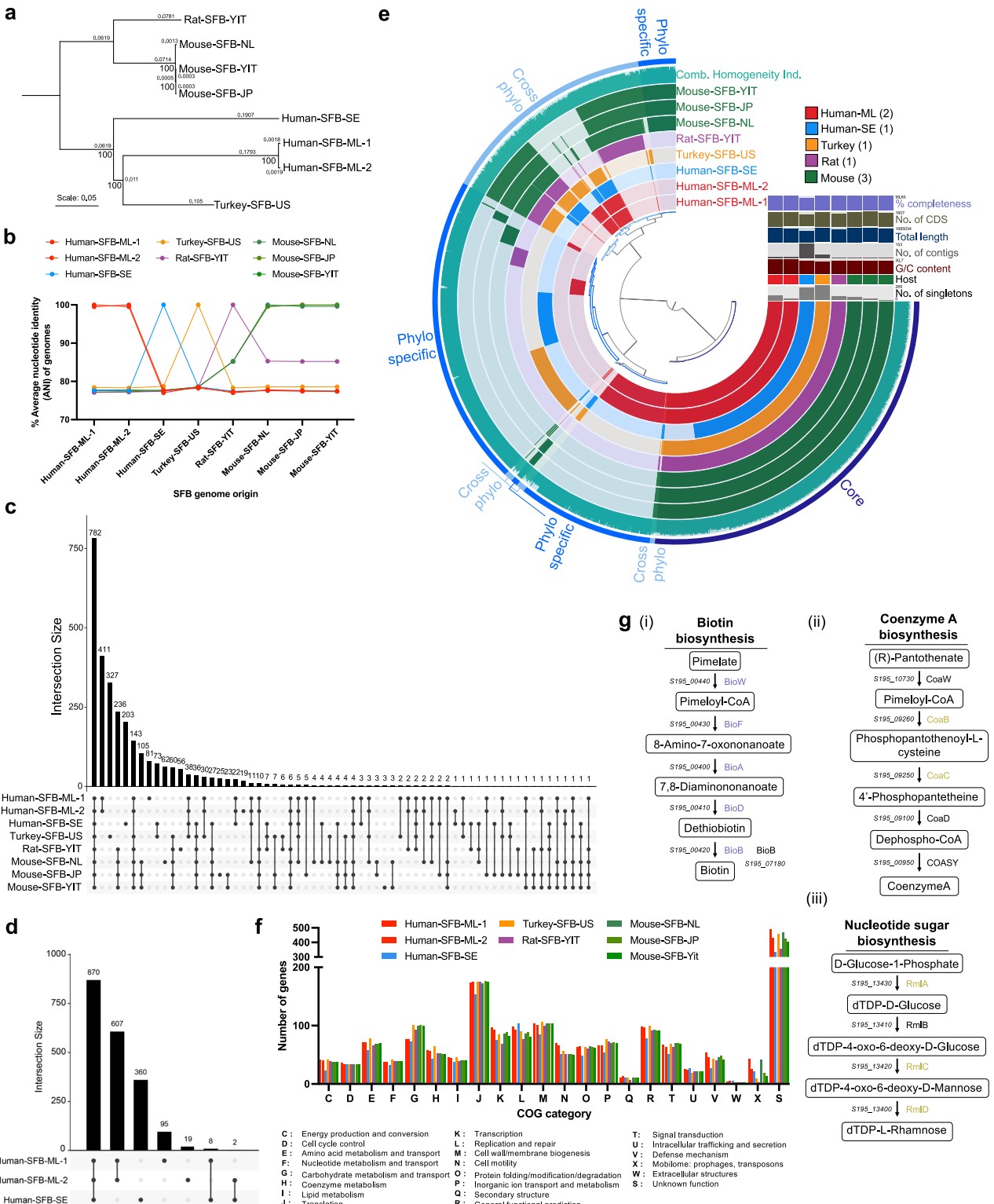

**Fig. 3 | Genome analysis across SFB from various hosts. a** Maximum likelihood phylogenetic tree of 782 core SFB genes from various hosts, including bootstrap values and branch lengths. Scale: nucleotide substitutions per nucleotide position. **b** Percent average nucleotide identity (ANI) across SFB genomes from various hosts. **c** UpSet plot of the SFB genomes from various hosts. Intersections correspond to gene clusters shared between the indicated SFB genomes. **d** UpSet plot of the SFB genomes from human origin. **e** SFB pangenome analysis using Anvi'o and highlighting a number of genome features and core, cross-phylo, and phylo-specific gene clusters. No. number, Singletons gene clusters present only in one genome, CDS coding sequences. Completeness is based on CheckM v1.1.0. **f** Number of genes assigned to different COG functional categories for SFB from various hosts. **g** Schematics of metabolic pathways found in human and turkey, but not rodent, SFB genomes: **g**(i) biotin biosynthesis, **g**(ii) coenzyme A biosynthesis, and **g**(iii) rhamnose biosynthesis. Predicted enzymes in black are part of the SFB core genome; those in purple are only predicted in the Human-SFB-ML, Human-SFB-SE, and Tukey-SFB-US genomes; and those in yellow-green are only predicted in the Human-SFB-ML and Turkey-SFB-US genomes. The Human-SFB-ML-2 gene locus tag annotation is provided as a reference.

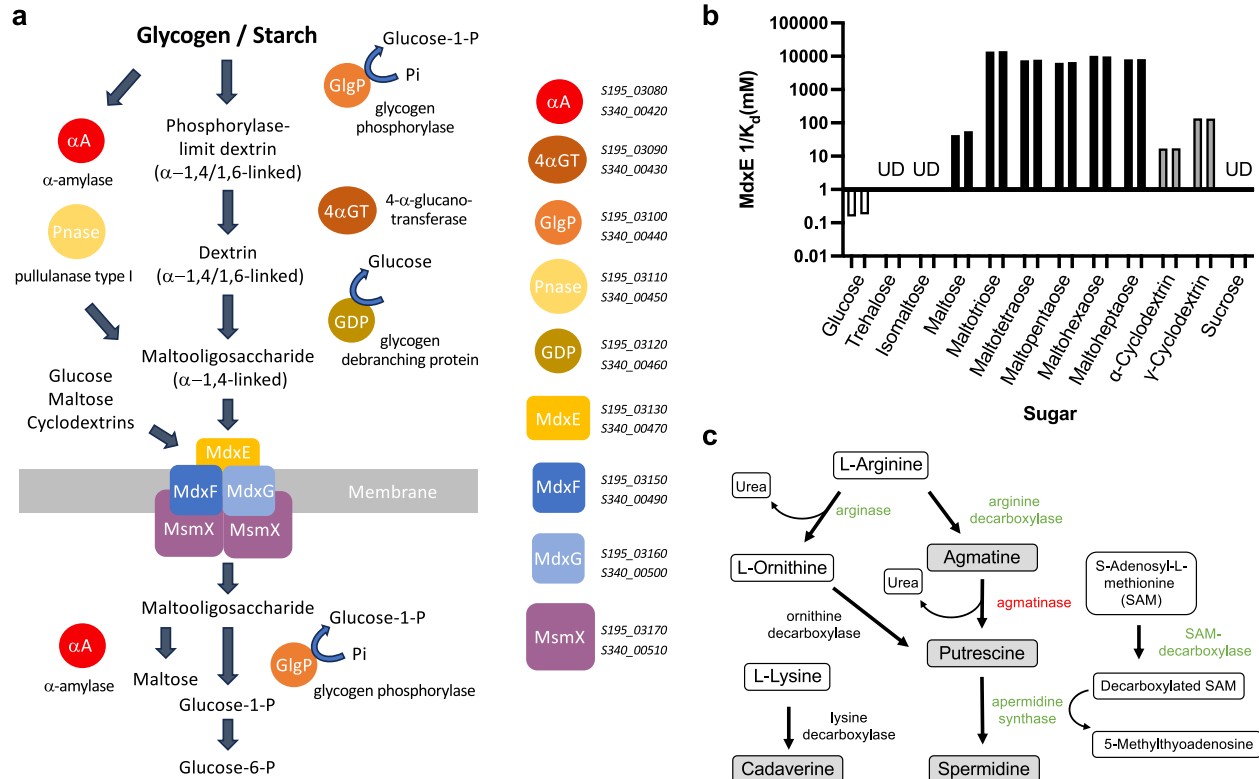

**Fig. 4 | Human-SFB-ML-specific genome features. a** Schematic of the components of the glycogen and starch utilization module identified in the Human-SFB-ML genomes. Includes the gene locus tags for Human-SFB-ML-1 (S340) and 2 (S195) for reference. **b** Sugar-binding affinities of Human-SFB-ML MdxE showing the results for each substrate from two independent experiments for a single purified protein sample. Carbohydrates with a α-(1,4) linkage of glucose subunits are colored in black when linear and gray when circular; Trehalose and isomaltose are dimers of glucose with a α-(1,1) and α-(1,6) linkage, respectively. **c** Schematic of the biosynthetic pathways for polyamines. Predicted enzymes present in all eight genomes are in green, a Orn/Lys/Arg decarboxylase predicted to be present in all genomes but whose substrate specificity is unclear is in black, and the enzyme predicted to be present only in the Human-SFB-ML genomes is in red. Polyamines are shaded in gray.

both Human-SFB-SE and Turkey-SFB-US, and 22 genes only with Human-SFB-SE (Fig. 3c). Among the genes shared only between the Human-SFB-ML genomes and Human-SFB-SE, we identified essential elements for a putative conjugation system of the FATA subtype (Supplementary Fig. 13a, b)[67,68]. However, these genes are located in a high GC content region that is part of a locus with features of other mobile genetic elements (Supplementary Fig. 14). Notable differences between the SFB genomes include a divergent biotin acquisition strategy for the human SFB and turkey SFB genomes as compared to rodent SFB. The human SFB and turkey SFB genomes all include the necessary genes required for biotin synthesis, but are devoid of a critical component in the biotin uptake ABC transporter present in the rodent SFB genomes (Fig. 3g (i), Supplementary Table 3)[33]. In addition, the Human-SFB-ML and Turkey-SFB-US genomes are enriched in glycosyl transferases (Supplementary Fig. 13c), have additional genes involved in folic acid biosynthesis (Supplementary Fig. 13d), share a full acetyl-CoA biosynthetic pathway for the generation of the acyl-carrier protein from pantothenate (Fig. 3g (ii)), and encode the metabolic pathway for the generation of dTDP-L-rhamnose (Fig. 3g (iii)), which may contribute to differences in the cell wall polysaccharide composition of Human-SFB-ML and turkey SFB as compared to rodent SFB.

### Human-SFB-ML has previously undescribed SFB genome features

There is a substantial heterogeneity in the repertoire of PTS and ABC transporters across the SFB genomes within the phylogenetic groups of turkey SFB, mouse SFB, rat SFB, Human-SFB-SE, and Human-SFB-ML

(Supplementary Table 3). The Human-SFB-ML genomes were devoid of a number of sugar import functions present in other SFB genomes and have a low number of glycosyl hydrolases (Supplementary Fig. 13c, Supplementary Table 3). While a definitive analysis of missing genes necessitates full genome closure, the absence of these genes in both Human-SFB-ML genomes does, however, provide additional confidence. Strikingly, the Human-SFB-ML genomes include instead a full set of genes involved in the degradation of glycogen and starch and the import of maltooligosaccharides (Fig. 4a, Supplementary Table 3). Components of this module had the closest hits to those in *Clostridium, Epulopiscium*, and *Vallitalea* with a range of 51–74% amino acid sequence identity. Using surface plasmon resonance, the recombinant substrate-binding protein MdxE (Supplementary Fig. 15a, b) of the maltooligosaccharide ABC transporter displayed weak binding to glucose, substantial binding to maltose and α and γ-cyclodextrins, and high affinity to a range of 1,4-linked maltooligosaccharides (Fig. 4b, Supplementary Fig. 15, Supplementary Table 4). No binding capacity was detected for sucrose, trehalose, or isomaltose. Together, these data support the potential of Human-SFB-ML to utilize glycogen and starch and their degradative products as carbon and energy sources.

In addition, beyond the presence of an alkanesulfonate ABC transporter (Supplementary Table 3) to scavenge sulfur, Human-SFB-ML may have additional protection against environmental stresses. SFB across all hosts have an arsenal of enzymes, at various copy numbers, predicted to be involved in the scavenging or detoxification of reactive oxygen species to protect the organism from oxidative damage (Supplementary Table 5). These include genes encoding

catalase, peroxiredoxin, thioredoxin and thioredoxin reductase, a glutaredoxin-related protein, rubredoxin and rubrerythrin, a homolog for the peroxide stress protein YaaA[69], and an arginase, which, through its catabolism of L-arginine, is hypothesized to limit this amino acid for the host synthesis of nitric oxide, a compound involved in host defense and host immune responses[35]. The Human-SFB-ML genomes share all these functions but in addition include flavodoxins, a glutathione peroxidase, three nitroreductases, an osmoprotectant ABC transporter, and an agmatinase (Fig. 4c, Supplementary Fig. 13e, Supplementary Table 5). While the SFB genomes all have two arginine/lysine/ornithine decarboxylases, the predicted presence of agmatinase completes the pathway for the production of the polyamines putrescine and spermidine (Fig. 4c) from agmatine, which together with cadaverine may play a critical role in a large number of biological functions, including resistance to stresses such as oxidative stress[70]. The presence of these additional genes suggests that Human-SFB-ML may be able to withstand a more hostile environmental niche.

### SFB are members of the human gut microbiota worldwide

To obtain a better understanding of the global distribution of SFB within the human population, we next expanded our developed workflow to identify human SFB in other 16S rRNA gene amplicon datasets. For this, we first analyzed the range of nucleotide identity in the variable regions of the 16S rRNA gene sequences within the SFB clade (Supplementary Data 1) and as compared to members of the nearest *Clostridia* group (Fig. 5a). A clear separation of nucleotide identity between the 16S rRNA gene sequences of those within and outside of the SFB clade was apparent (Fig. 5a). The 16S rRNA gene reference sequences of the SFB clade shared a nucleotide identity as low as 85% for the V1-V2 region and as low as ~95% for the V3-V4 and V4 regions (Fig. 5a, Supplementary Data 1c−f). Conversely, the nearest non-SFB clade *Clostridium* species in the NCBI nucleotide and EzBioCloud databases, including our *Clostridium* reference sequences, shared a nucleotide identity only as high as ~78% for the V1-V2 region, as high as ~88% for the V3-V4 region, and as high as ~87% for the V4 region (Fig. 5a).

To search publicly available 16S rRNA gene amplicon databases for SFB, we focused on bioprojects with 16S rRNA read lengths of at least 200 bp, used a cut-off in nucleotide identity of 91% for the V1-V2 region and 94% for the V4 or V3-V4 regions over 90% of the read length, and included four 16S rRNA gene sequences (Human-SFB-ML, Human-SFB-SE, Chicken-SFB-BE and Mouse-SFB-NL) as the reference SFB (Supplementary Fig. 16a). Using these criteria, all 16S rRNA reads identified had SFB as the closest hit when BLAST was performed against the NCBI nt database, excluding an incorrectly annotated bacterial genome containing a mouse SFB rRNA gene contig (Supplementary Fig. 17, Supplementary Table 6), and fell within the SFB clade on a maximum likelihood phylogenetic tree. Overall, we identified SFB 16S rRNA reads in 79 bioprojects as well as two SFB sequences of human intestinal origin in the NCBI GenBank database (Supplementary Data 4). The number of independent SFB-positive bioprojects per country ranged between 1 and 21, with bioprojects consisting of 1 to 173 SFB-positive samples (also referred to as sequence read archives or SRAs). In addition, SFB were identified in fecal samples of children from Uganda in a multiplex 16S rRNA gene amplicon dataset, and in children from Ethiopia and Niger and an adult from France in metagenomic RNA sequencing datasets (Supplementary Data 5, Supplementary Data 6). Metagenomic samples were considered SFB-positive only if mapping of paired reads at a mapping stringency of 100% for read nucleotide identity and read length led to a consensus sequence that fell within the SFB clade on a maximum likelihood phylogenetic tree and whose first classified hit on the NCBI nt database was SFB. SFB were thereby identified in forty-two countries (Fig. 5b, Supplementary Data 7), establishing SFB as gut commensal bacteria present in humans across all inhabited continents.

### Humans harbor four main and at least two minor SFB lineages in their gut microbiota

To further analyze the SFB-positive bioprojects, the dominant SFB 16S rRNA read in each bioproject was identified. The majority of the SFB 16S rRNA reads had 99% to 100% nucleotide sequence identity to one of the four reference SFB 16S gene sequences (Human-SFB-ML, Human-SFB-SE, Mouse-SFB-NL, and Chicken-SFB-BE) (Supplementary Data 8). Phylogenetic analysis of the dominant SFB 16S rRNA reads placed all bioprojects of the V1-V2 region (Supplementary Fig. 18) and most of the bioprojects that include the V4 region (Fig. 5c) in the same clade as one of these four reference SFB 16S rRNA gene sequences. For a more detailed analysis, the dominant sequence of each individual SRA was identified and assigned to an SFB lineage based on the highest percent nucleotide identity to one of the fifteen reference SFB 16S rRNA gene sequences (Supplementary Data 4, Supplementary Fig. 16b). Eight samples could not be placed within a SFB lineage using nucleotide identity alone, predominantly due to the high nucleotide identity in the V4 and V3-V4 16S rRNA gene region of mouse SFB and HumanSkin-SFB-US. All other 872 samples showed congruency in SFB reference assignment between nucleotide identity and phylogenetic analysis (Supplementary Data 4). Maximum-likelihood phylogenetics for SRAs spanning the 16S rRNA gene V1-V2 (Supplementary Fig. 19), V4 (Fig. 5d(i), Supplementary Fig. 20a) or V3-V4 (Fig. 5d(ii), Supplementary Fig. 20b) regions placed the majority of the samples within the clusters of Human-SFB-ML, Human-SFB-SE, Mouse-SFB-NL and Chicken-SFB-BE. Focusing on the SRAs with V3-V4 16S rRNA reads, additional SFB lineages became apparent (Fig. 5d, Supplementary Fig. 20b). A Human-SFB-refRat lineage with predominantly 100% nucleotide sequence identity to Rat-SFB-YIT was found in 8 samples from 2 independent studies originating in China and India. In addition, a discernible clade became apparent near the chicken SFB clade (dark yellow) and designated as a chicken-related lineage (ChickenRel), with samples from four independent studies showing 98.4−99.5% nucleotide sequence identity to Chicken-SFB-BE. Within the mouse SFB clade, a grouping (light green) was also apparent, originating from a single study in India. This grouping was not considered a separate lineage, as phylogenetic separation was less pronounced. Only three other samples were closer to other SFB 16S rRNA gene reference sequences (trout, dog, and shorebird), albeit with a low nucleotide identity and thus also not designated as separate lineages. The distribution of the six robust SFB lineages identified was plotted (Fig. 5e, Supplementary Data 7), and the dominant SFB lineage per country was used to color the country on the world map (Supplementary Fig. 21). Together, these findings support the existence, based on the V3-V4 16S rRNA gene region, of at least six human SFB lineages in humans worldwide, with the main lineages being those with the highest similarity to Human-SFB-ML, Human-SFB-SE, mouse SFB, and chicken SFB.

### Identification of the Human-SFB-ML genome in metagenomic datasets

To validate our 16S rRNA gene-based approach for the identification of SFB, we extended our analysis to the genome level and searched for the Human-SFB-ML genome in publicly available metagenomic datasets (Supplementary Fig. 22a). As previously done for the RNA metagenomic data, metagenomic samples were considered SFB-positive only if mapping of exclusively paired reads with a mapping stringency of 99−100% nucleotide identity over 100% read length to a full-length SFB 16S rRNA gene reference sequence led to a consensus sequence whose first hit on the NCBI nt database was SFB and phylogenetically clustered with the SFB clade. Fourteen SFB-positive samples were found in Zimbabwean children in a study of stunting[71] (Supplementary Data 5, Supplementary Data 6). In a single fecal sample from a 6-month-old infant, mapping of paired-end metagenomic reads to the full-length Human-SFB-ML 16S rRNA gene sequence led to a consensus sequence with 100% coverage and a 99.9% nucleotide sequence

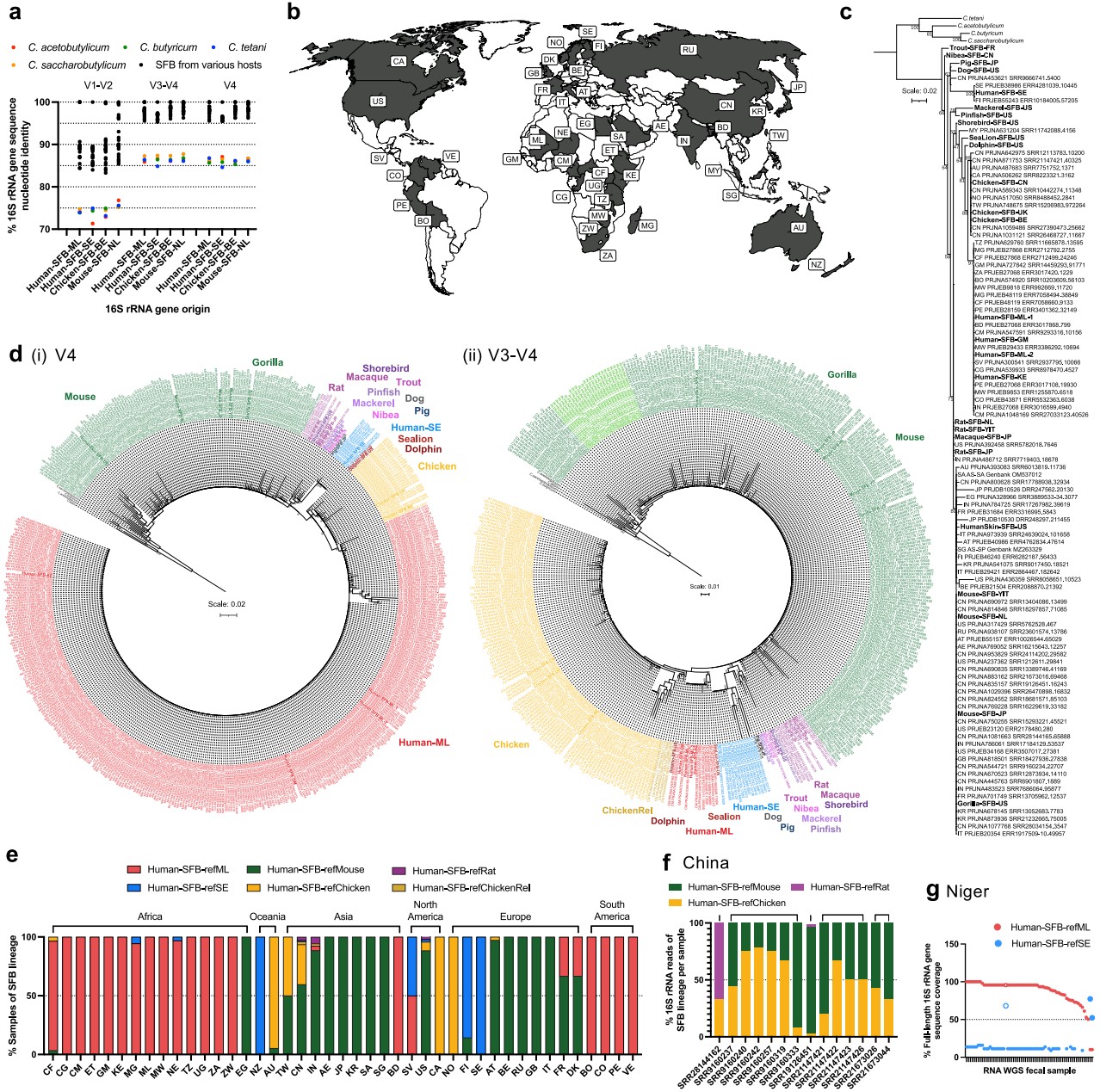

**Fig. 5 | Global SFB presence based on 16S rRNA gene sequence analysis.**
**a** Percent nucleotide identity of 16S rRNA gene variable regions for the Human-SFB-ML, Human-SFB-SE, Chicken-SFB-BE, and Mouse-SFB-NL 16S rRNA gene sequences as compared to the same variable regions in the SFB reference sequences from various hosts (Supplementary Fig. 3) and four *Clostridium* outgroup species.
**b** Countries with individuals positive for SFB. Analysis based predominantly on 16S rRNA gene sequence analysis (Supplementary Data 7). **c** Maximum likelihood phylogenetic tree of the V4 region of SFB reference 16S rRNA gene sequences (bold) and the dominant 16S rRNA gene read of all 16S rRNA gene amplicon-based bioprojects of Supplementary Data 4 with >200 bp reads that included the V4 region; includes *Clostridium* outgroup species (italic). **d** Maximum likelihood phylogenetic trees of the dominant SFB 16S rRNA gene read for all samples with >200 bp reads in the **d**(i) V4 and **d**(ii) V3-V4 16S rRNA gene variable region (Supplementary Data 4) as well as the SFB reference 16S rRNA gene sequences (bold) (Supplementary Fig. 3) and *Clostridium* outgroup species (black and italic). Samples are colored based on the host of their nearest SFB 16S rRNA gene reference sequence. For **d**(ii), an additional separate lineage most closely related to chicken

SFB (ChickenRel) is included, and a sub-clustering within the mouse SFB clade is highlighted (light green). **e** SFB lineage distribution per country based on the percent of each lineage within the SFB-positive samples from bioprojects included in Supplementary Data 7. SFB lineage co-occurrence within a sample for **f** bioprojects from China based on 16S rRNA gene amplicon data (Supplementary Table 7), and **g** the bioproject PRJNA549968 from Niger based on RNA metagenomic sequencing data and percent 16S rRNA gene coverage using PebbleScout analysis. For (**f**), samples from the five separate projects are grouped per bioproject above the graph, and the sample reference is provided; for (**g**) open circle denotes the sample with Human-SFB-refSE and Human-SFB-refML co-occurrence. Samples above the dotted line are positive for the indicated human SFB lineage. Sample names are listed in Supplementary Data 6. **c**, **d** SFB reads are labeled with the two-letter country code of the country of origin of the sample, the bioproject number, the SRA number, and the sequence read number. **b**, **e** Countries are labeled based on their two-letter country code. **c**, **d** ML trees include bootstrap values, and the scale is nucleotide substitutions per nucleotide position.

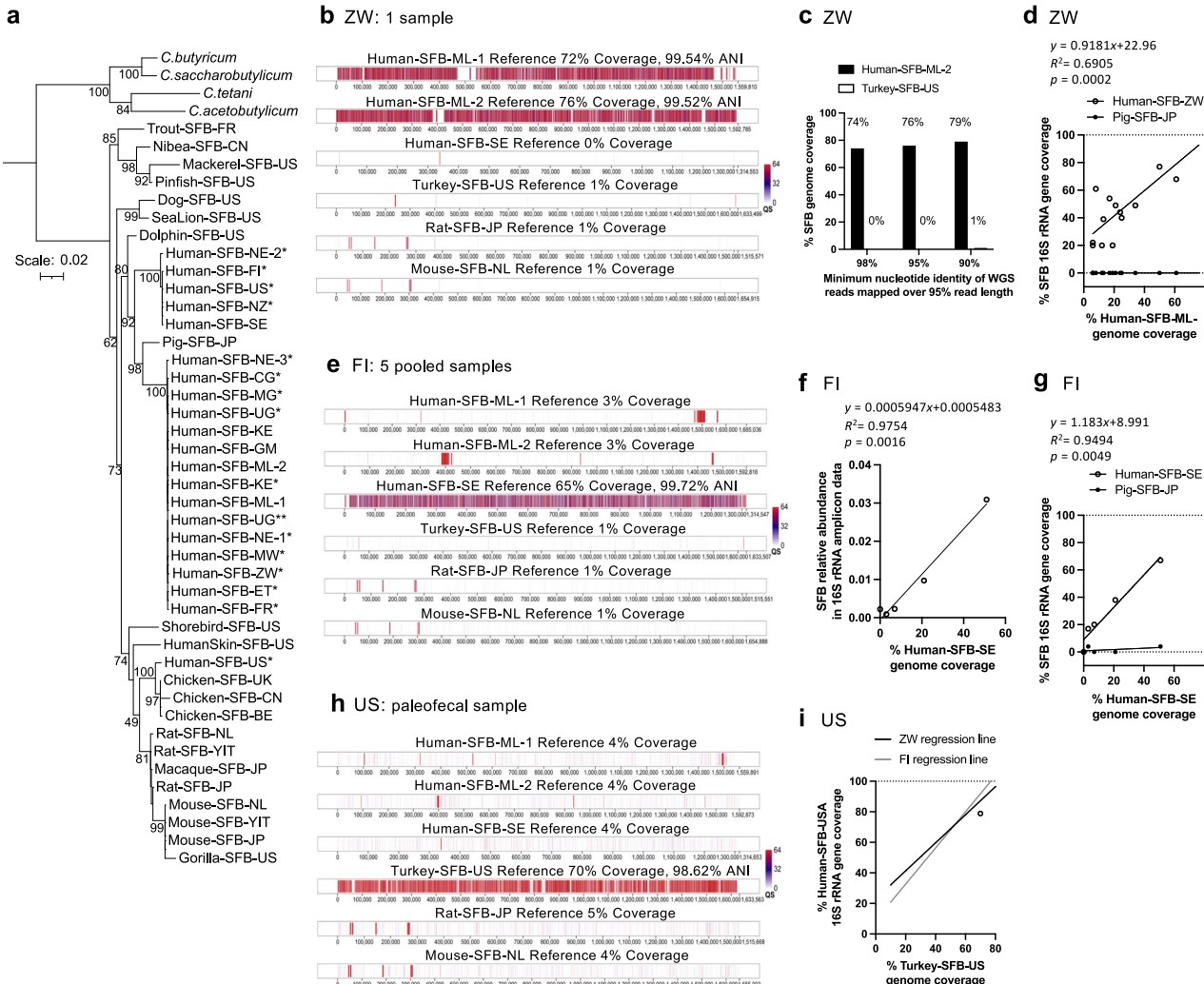

**Fig. 6 | Identification of SFB in metagenomic datasets. a** Maximum likelihood phylogenetic tree of reference SFB 16S rRNA gene sequences and 16S rRNA gene sequences obtained from (*) metagenomic read mapping or (**) multiplex 16S rRNA gene amplicon analysis; includes *Clostridium* outgroup species (italic). Tree includes bootstrap values and the scale is nucleotide substitutions per nucleotide position. **b**–**d** Analysis of the Zimbabwe bioproject PRJEB51728. **b** Mapping of trimmed sequence reads of the SRA ERR10900747 to the SFB genome of various hosts. **c** Comparison between the percent SFB genome coverage for Human-SFB-ML-2 and Turkey-SFB-US using variable read mapping stringencies of 98–90% for nucleotide identity. **d** Correlation between the percent coverage of the Human-SFB-ML-2 genome and the Human-SFB-ZW and Pig-SFB-JP 16S rRNA gene sequences obtained through metagenomic read mapping. Includes a simple linear regression line. **e**–**g** Analysis of the Finland bioproject PRJEB70237. **e** Mapping of trimmed sequence reads of five fecal samples (Supplementary Fig. 26c) to the SFB genome of various hosts. **f** Correlation between the percent Human-SFB-SE genome coverage obtained through metagenomic read mapping and the SFB relative abundance in

the 16S rRNA gene amplicon bioproject PRJEB55243 dataset of the same five samples. **g** Correlation between the percent coverage of the Human-SFB-SE genome and the Human-SFB-SE and Pig-SFB-JP 16S rRNA gene sequences obtained through metagenomic read mapping. Includes a simple linear regression line. **h**, **i** Analysis of the human paleofecal SRA SRR12557707 from the USA (US). **h** Mapping of trimmed metagenomic reads to the SFB genomes of various hosts. **i** Comparison between the percent coverage of the Turkey-SFB-US and 1491 bp Human-SFB-US paleofecal 16S rRNA gene sequence obtained through metagenomic read mapping. Includes the simple linear regression lines obtained for the equivalent analysis with the samples from Zimbabwe (ZW) and Finland (FI) in **d**, **g**, respectively, for comparison purposes. **b**–**i** Mapping of cleaned and trimmed metagenomic reads was performed at a 95% nucleotide sequence identity and read length for SFB genome mapping, except in (**c**), and at 100% nucleotide sequence identity and read length of only paired-end reads for SFB 16S rRNA gene sequence mapping using the CLC Genomics Workbench software package. QS Quality score.

identity that fell within the Human-SFB-ML clade (Fig. 6a, Supplementary Data 5). Mapping of the metagenomic reads onto the SFB genomes, with mapping parameters of a minimum of 95% nucleotide identity over 95% read length, led to a consensus sequence that covered 76% of the Human-SFB-ML-2 genome with an ANI of 99.5% (Fig. 6b). Conversely, less than 1% of the other SFB genomes available were represented under identical mapping conditions (Fig. 6b). This low coverage is expected given the low whole genome ANI across SFB from various hosts (Fig. 3b, Supplementary Table 2a). The even distribution of metagenomic reads along the Human-SFB-ML genomes,

the lack of read mapping to other SFB genomes, and a general lack of substantial genome coverage with more stringent or more relaxed mapping parameters (Fig. 6c), suggest that the lack of full genome coverage is due to an insufficient sequencing depth and not due to more variable regions in the genome. In support, combining the metagenomic reads of the 14 SFB-positive Zimbabwean fecal samples increased coverage of the Human-SFB-ML-2 genome to 98% while maintaining 99.6% ANI and only 1% genome coverage of the most closely related genome available, the Turkey-SFB-US genome (Supplementary Fig. 23a, Supplementary Data 6). Notably, the consensus

sequence of the single sample, as well as the 14 pooled samples, included Human-SFB-ML-specific genome features such as the agmatinase and starch utilization module (Supplementary Fig. 24, Supplementary Data 2), supporting the conservation of these features in the African human SFB lineage. In addition, there was a good correlation between the percent coverage of the Human-SFB-ML-2 genome and the Zimbabwe 16S rRNA gene consensus sequence (R-squared = 0.6905, p-value = 0.0002), while no coverage or correlation was obtained with the otherwise closest SFB 16S rRNA gene sequence (Pig-SFB-JP) (Fig. 6d, Supplementary Data 5). Based on the frequency of mapped metagenomic reads, the SFB frequency in the Zimbabwe samples was less than 0.25% and correlated well (R-squared = 0.8690, p-value = <0.0001) with the percent coverage of the Human-SFB-ML-2 genome (Supplementary Fig. 23b). These results further validate the metagenomic read mapping parameters, support the notion that a lack of genome coverage is due to insufficient sequencing depth, and identify the Human-SFB-refML lineage on the genome level in Zimbabwe.

The Human-SFB-ML genome was additionally identified in fecal samples from children of the Hadza hunter-gatherer tribe in Tanzania[72], in fecal samples from children from Niger, Uganda, and Kenya, and, with PebbleScout[73], in fecal samples of individuals from the Republic of the Congo and children from Madagascar and Malawi (Supplementary Data 5, Supplementary Data 6). Using the same stringent 16S rRNA read mapping conditions, a single fecal sample from Malawi, Madagascar, the Republic of the Congo, Tanzania, Kenya, Uganda, and Niger led to 13%, 24%, 37%, 54%, 54%, 66% and 93% Human-SFB-ML-2 genome coverage, respectively, and less than or equal to 1% of the other SFB genomes (Supplementary Fig. 23 c–e, Supplementary Fig. 25a–d). In these samples, the Human-SFB-ML 16S rRNA gene sequence was identified at a 99.9% nucleotide identity and with an overall similar trend of increasing 16S rRNA gene sequence coverage with increasing genome sequence coverage (Supplementary Data 5). As expected, the consensus 16S rRNA gene sequences all clustered together with Human-SFB-ML on a phylogenetic tree (Fig. 6a). The trend of increasing 16S rRNA gene sequence and genome sequence coverage was also present within the bioprojects from Tanzania, Niger, and Malawi that included multiple SFB-positive samples (Supplementary Fig. 23f). Combining the metagenomic reads of SFB-positive samples within each bioproject again increased the coverage while maintaining an ANI of over 99.4% (Supplementary Fig. 23c–e(ii)). In addition, the Congo cohort included 16S rRNA gene amplicon data of the same samples (Supplementary Data 4), and a comparison between the SFB 16S rRNA read number or relative abundance of the 16S rRNA gene amplicon data and Human-SFB-ML-1 genome coverage, obtained through metagenomic read mapping, showed the expected correlation (Supplementary Fig. 25a(ii, iii)). As seen for Zimbabwe, the SFB genomic sequences obtained, through metagenomic read mapping, from individuals of other African countries also point towards the presence of the unique Human-SFB-ML factors, such as the agmatinase, glutathione peroxidase, alkanesulfonate and osmoprotectant transporters, and the starch/glycogen utilization system, in Human-SFB-refML genomes throughout Africa (Supplementary Data 2), These data further substantiate, on the genome level, the presence of the Human-SFB-refML lineage in the gut microbiota of humans across geographically distant locations in Africa.

### Identification of the Human-SFB-SE genome in metagenomic datasets

We then extended the metagenomic analysis to the other SFB genome of human origin. The Human-SFB-SE genome was identified in children participating in the Finnish HELMI cohort[74], an individual from the USA[75], and a 24-month-old child from a New Zealand of European descent (Supplementary Fig. 26a–c)[76]. Read mapping of the New Zealand and USA sample led to a 71% and 54% Human-SFB-SE genome coverage at a 100% and 99.9% ANI (Supplementary Fig. 26a, b), respectively, as well as 100% nucleotide identity for both covering the >1520 bp Human-SFB-SE 16S rRNA gene sequence (Fig. 6a, Supplementary Data 5). The Finnish HELMI cohort had previously been analyzed using 16S rRNA gene amplicon sequencing[77] and for five of the SFB-positive samples the metagenomic data was available (Supplementary Data 4, Fig. 26c). This allowed a comparison between SFB detection based on 16S rRNA gene amplicon and metagenomic data. Read mapping of the sample with the highest SFB relative abundance from a Finnish 24-month-old child led up to a 51% Human-SFB-SE genome coverage at a 99.7% ANI (Supplementary Fig. 26c (v)) as well as 100% nucleotide identity covering the 1495 bp Human-SFB-SE 16S rRNA gene sequence (Supplementary Data 5, Fig. 6a). Pooling of the metagenomic reads from the five samples increased the genome coverage from 51% to 65% without affecting the ANI and with only a maximum 3% coverage of other SFB genomes (Fig. 6e). For the five samples, a high correlation was found between the 16S rRNA gene amplicon-based SFB relative abundance and coverage of the Human-SFB-SE genome by WGS read mapping (R-squared = 0.9754, p-value = 0.0016) (Fig. 6f and Supplementary Fig. 26d) as well as the SFB relative abundance calculated based on WGS read mapping frequency (R-squared = 0.9675, p-value = 0.0025) (Supplementary Fig. 26e). These results validate our approaches of identifying SFB in both 16S rRNA gene amplicon and metagenomic datasets. The percent coverage of the Human-SFB-SE 16S rRNA gene sequence and genome were also correlated for Human-SFB-SE (R-squared = 0.9494, p-value = 0.0049), as seen for Human-SFB-refML in Zimbabwe (Fig. 6d), while no correlation was obtained with the Pig-SFB-JP 16S rRNA gene sequence (Fig. 6g). Furthermore, the frequency of mapped metagenomic reads correlated well with genome coverage (Supplementary Fig. 26f). These data identify the Human-SFB-refSE lineage, at the genome level, in New Zealand, the USA, and Finland.

### Identification of the Turkey-SFB-US genome in a metagenomic dataset of paleofecal samples

Going beyond the human SFB genomes, we did not identify the mouse or rat SFB genome in human samples, but we did identify the Turkey-SFB-US genome (Supplementary Table 2) in a ~1500-year-old human paleofecal sample recovered from a cave in Southwestern USA[78]. Mapping of the metagenomic sequence reads resulted in a 70% coverage of the Turkey-SFB-US genome with an ANI of 98.6%, while other SFB genomes showed only up to 5% coverage using identical mapping parameters (Fig. 6h). Coverage was again largely evenly spaced across the genome. In addition, the percent coverage of the Human-SFB-US paleofecal 16S rRNA gene sequence and Turkey-SFB-US genome was close to the linear regression lines obtained for Human-SFB-refML in Zimbabwe and Human-SFB-refSE in Finland (Fig. 6i), pointing towards a limitation of sequencing depth rather than only partial genome identification. These results support the presence of an SFB with similarity to the SFB found today in turkeys in an individual living around 1500 years ago in North America.

### The Human-SFB-ML lineage is distributed globally in a non-random pattern

The 16S rRNA gene amplicon and metagenomic data analysis provides a first global picture of human SFB lineage distribution (Fig. 5e, Supplementary Fig. 21, Supplementary Data 7). Africa, as a continent, was strikingly homogenous in SFB lineage distribution. Based on the 16S rRNA gene analysis, all fourteen countries south of the Sahara had at least 93% of their SFB-positive samples with a SFB reference designation of Human-SFB-refML. Overall, 98.7% of the 396 SFB-positive samples were of the Human-SFB-refML lineage, while eleven countries, with a total of 284 samples, showed a 100% Human-SFB-refML sample designation. Similarly, the Human-SFB-refML lineage made up 100% of

the 53 SFB-positive samples of the four SFB-positive South American countries. This contrasts with Europe, where Human-SFB-refML constituted only 2 of the 100 (2.0%) SFB-positive samples across 10 countries, and Asia, where, with the exception of Bangladesh, Human-SFB-refML made up only 2 of 292 (0.7%) SFB-positive samples across the remaining 9 countries. In addition, the Human-SFB-refML lineage was not represented in the 143 SFB-positive samples spanning Canada, the USA, and Australia. The Human-SFB-refML lineage thereby showed an uneven distribution across the world.

For now, when focusing on countries outside of Africa with a relatively large (14 or more) number of SFB-positive samples (BD, CN, IN, KR, AU, AT, FI, IT, NO, CA, US, PE, VE), the majority of the samples within one country harbored SFB of the same lineage (Fig. 5e, Supplementary Data 7). An exception was China, which, while being strikingly devoid of the Human-SFB-refML lineage, with a dataset of 21 separate projects and 205 SFB-positive samples, showed a clear mixed sample population of Human-SFB-refChicken and Human-SFB-refMouse at 32.7% and 59.5%, respectively, with 76% of bioprojects displaying a mixed SFB lineage designation (Fig. 5e, Supplementary Data 4). While the low SFB lineage diversity in many countries may be a result of a limited number of bioprojects analyzed per country, thereby potentially reflecting only a small area of the country, the data nevertheless support the existence of regional variations in the distribution of the different SFB lineages.

### Human SFB lineages co-exist within a single individual

The existence of multiple SFB lineages within the human population also raised the question of whether SFB lineages could co-exist within one individual. To address this question, we categorized all reads in the V3-V4-containing 16S rRNA gene amplicon SRAs based on their percent nucleotide identity to the fifteen reference SFB 16S rRNA gene sequences available (Supplementary Fig. 3, Supplementary Fig. 22b). This identified the co-existence of multiple SFB lineages in China (Fig. 5f, Supplementary Table 7). In line with the Human-SFB-refMouse and Human-SFB-refChicken being particularly abundant lineages in China, 14 SRAs from separate individuals of 5 bioprojects harbored both of these SFB lineages. In two separate bioprojects, an SRA contained also the Human-SFB-refRat lineage, with one SRA containing all three Human-SFB-refChicken, refMouse, and refRat lineages (Supplementary Table 7) with a robust 100% nucleotide sequence identity to the V3-V4 region (405 bp). The 16S rRNA reads covering the relatively long V3-V4 16S rRNA gene region showed a clear clustering with the reference lineages on a maximum likelihood phylogenetic tree as they were all at least 99% identical to their respective reference lineage (Supplementary Fig. 27). Notably, both the Human-SFB-refMouse and Human-SFB-refChicken lineage could be dominant within an individual (Fig. 5f). In addition, co-occurrence of the Human-SFB-refML and Human-SFB-refSE lineages was identified in metagenomic RNA sequencing data of a fecal sample from Niger (Fig. 5g) and from Ethiopia (Supplementary Fig. 27c), first using PebbleScout[73], and then verified using metagenomic paired read mapping, BLAST analysis using the NCBI nt database, and phylogenetic positioning of the consensus sequence (Supplementary Data 5). For the SFB lineage co-colonization samples, 0% coverage was obtained when using the next nearest SFB 16S rRNA gene sequence of the pig (Supplementary Fig. 27d, e). These data reveal the potential for co-occurrence of SFB within a single individual for different combinations of five SFB lineages.

### Human SFB are at a greater relative abundance in intestinal biopsies than in feces

We next aimed to characterize the colonization dynamics of human SFB. Focusing on the location of human SFB colonization, the SFB prevalence was found to be higher in intestinal biopsies as compared to feces in studies of inflammatory bowel disease in China (Fig. 7a (i),

Supplementary Fig. 28a) and the USA (Fig. 7b (i)). In the latter study on pediatric Crohn's disease, the SFB prevalence was not significantly different for biopsies taken from the ileum, cecum, and rectum biopsies (Fig. 7b (i), but a high SFB relative abundance of over 0.1% was identified only in ileal biopsies (Fig. 7b (ii)). Ileal biopsies were also SFB-positive, with one sample at 0.24% SFB relative abundance, in a separate study from the USA (Supplementary Fig. 28b). Conversely, SFB was found in fecal samples, but in none of the duodenal or gastric chyme samples, in 5.6 and 2.7% of young children (2–5 years of age) living in the Central African Republic (Fig. 7c)[79] or Madagascar (Supplementary Fig. 28c)[79], respectively. These data are consistent with SFB being a colonizer of the ileal mucosal surface and having a general tropism for the mucosal surface beyond the ileum. In addition, these data are consistent with the human SFB lineages being minor members of the gut microbiota whose relative abundance is significantly higher in intestinal biopsies than fecal samples.

### SFB are low abundant but prevalent members of the gut microbiota in humans

As SFB are minor members of the gut microbiota, the assessment of their prevalence is highly dependent on the sequencing depth. Standard amplicon sequencing depth gives ~50–100,000 reads per sample for a detection limit of 0.002–0.001%. The limitation this imposes on SFB detection is evident in a rare ultra-deep sequencing study with ~2 million rRNA reads per sample, or down to 0.00005% in read frequency detection capability (Fig. 7d, e)[80]. In this early seminal study of the gut microbiota across westernized and non-westernized communities, we identified SFB 16S rRNA reads in fecal samples of 23.2% of Amazonians living in Venezuela and in 45.6% of individuals living in rural communities in Malawi (Fig. 7d (i), e (i)), with a median SFB relative abundance of 0.00008% and 0.00026%, respectively. At a regular sequencing depth of 0.001%, no individuals would have been identified in Venezuela. In Malawi, the SFB prevalence would have been reduced from 45.6 to 9.7% (Fig. 7d (i/ii), e (i/ii)). The increased detection power of the ultra-deep sequencing was, at an 8-fold increase, notably striking for the older (>10 years of age) population (Supplementary Fig. 28d). Many individuals, particularly in Venezuela, were SFB-positive with only a single 16S rRNA read and therefore at the detection limit (Fig. 7d). This indicates that SFB prevalence is likely still substantially underestimated even under ultra-deep sequencing conditions. The data also point towards potential geographic differences in SFB detectability, either due to the detection limit or a lack of SFB presence, as SFB was not detected in this study in the US cohort of individuals residing in St. Louis, Philadelphia, and Boulder in the USA (Supplementary Fig. 28f).

The human SFB lineages were detectable in the feces of individuals across a wide age range, including at an advanced age, for Human-SFB-refML in the Amazonian communities in South America (Venezuela) and rural Africa (Malawi), for Human-SFB-refChicken in urban Northern Europe (Norway), and for Human-SFB-refMouse, as well as Human-SFB-refChicken and Human-SFB-refRat, in China (Fig. 7d–g (ii)). At regular sequencing depths in the range of ~25,000–250,000 reads, with a detection limit of ~0.004–0.0004%, SFB had a prevalence of 0.2–100% in individual bioprojects and a high prevalence in a range of countries worldwide (Supplementary Data 4). For example, SFB was found in the feces of children at a prevalence of: 5.6% in the Central African Republic in a study of malnutrition (Fig. 7c (i)), 11.1% in Tanzanian Pemba Island in a study of chronic whipworm infection, 8.6% in Cameroon in a study of age and diet, and 100% in Canada in a study of preterm infants fed with either formula or human milk fortifier (Fig. 8a (i–iii)). In the Canadian preterm infants, SFB was identified with a median read number of six reads per sample, suggesting that a maximal median SFB relative abundance was reached.

SFB was also detected in feces of children and adults at a prevalence of 48.6% in China in a study of inflammatory depression

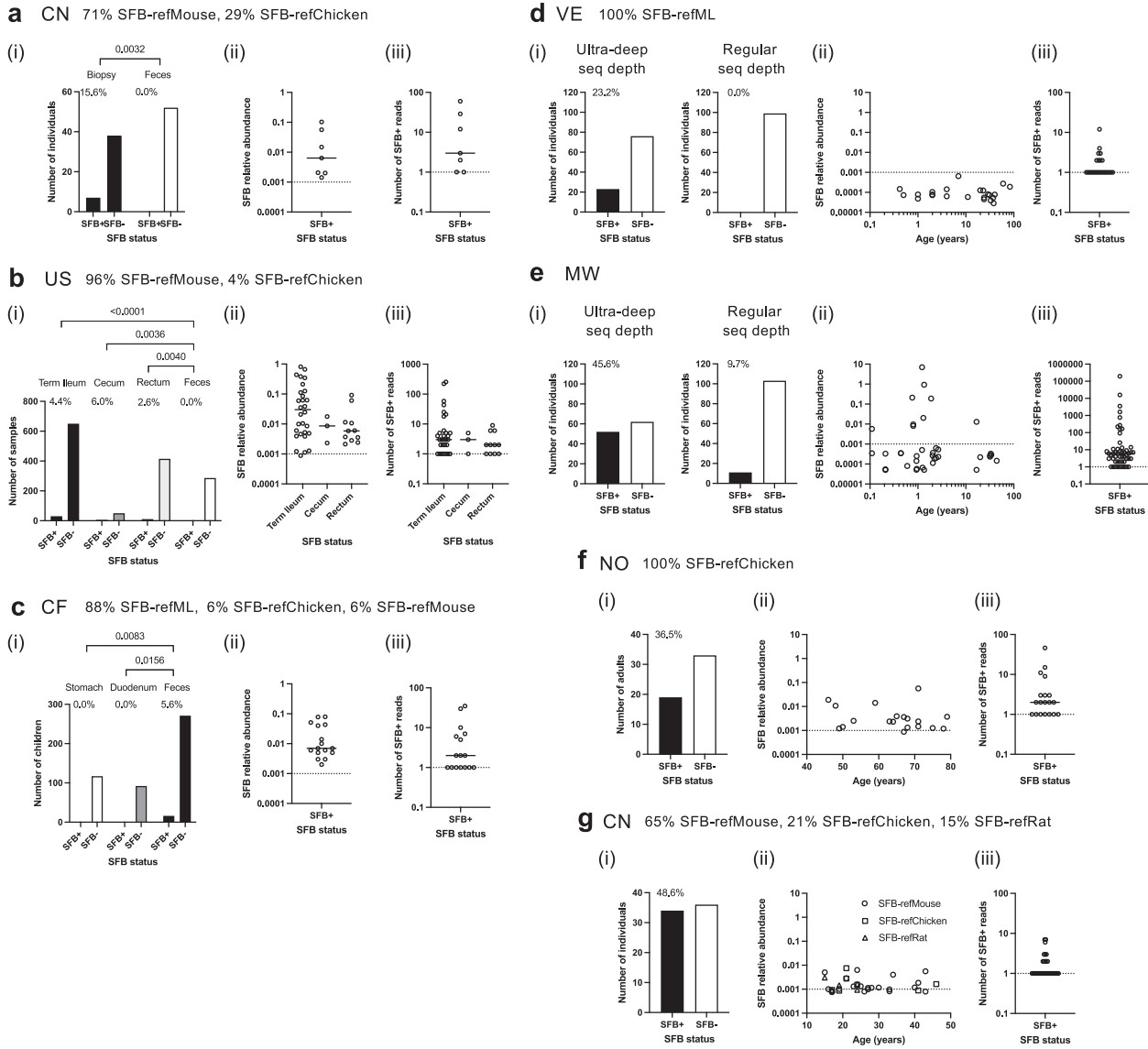

**Fig. 7 | Human SFB colonization location and relative abundance across different ages based on 16S rRNA gene amplicon analysis.** Analysis of **a**–**g**(i) SFB prevalence based on SFB 16S rRNA read identification and including the percentage of SFB-positive individuals, or for (**b**) the percentage of SFB-positive samples, **a**–**g**(ii) SFB relative abundance, and **a**–**g**(iii) SFB 16S rRNA read numbers in various bioprojects (PRJEB/PRJNA). **a** PRJNA800628, a study of Crohn's disease in China. Number of SFB-positive individuals: biopsy *n* = 7/45, feces *n* = 0/52. **b** PRJNA237362, a study of inflammatory bowel disease in the USA. Number of SFB-positive samples: terminal ileum *n* = 30/680, cecum *n* = 3/53, rectum *n* = 11/426, feces *n* = 0/286. **c** PRJEB48119, a study of malnutrition in children 2–5 years old in the Central African Republic. Number of SFB-positive children: stomach *n* = 0/117, duodenum *n* = 0/92, feces *n* = 16/287. **d**, **e** PRJEB3079, fecal samples from a study of microbiota diversity

in Venezuela, Malawi and the USA. Analysis of samples from **d** Venezuela and **e** Malawi, highlighting in **d**, **e**(i) the effect of ultra-deep sequencing on SFB prevalence calculation. Regular sequencing (seq) depth calculation is based on a relative abundance of 1 in 100,000 and higher. Number of SFB-positive individuals: Venezuela *n* = 23/99, Malawi *n* = 52/114. **f** PRJNA517050, fecal samples from a study of chronic alcohol consumption in Norway. Number of SFB-positive individuals: *n* = 19/52. **g** PRJNA1081663, fecal samples from a study of inflammatory depression in China. Number of SFB-positive individuals: *n* = 34/70. **a**–**d**, **f**, **g** Proportion of samples of a given SFB lineage is included, when possible. **a**–**c**(i) *P*-values were obtained with the two-sided Fisher's exact test. A dotted line is included at **a**–**g**(ii) a relative abundance of 0.001 and **a**–**g**(iii) a read number of 1 for reference purposes. **a**–**c**(ii)/**a**–**g**(iii) Crossbars indicate the median value of the datasets.

(Fig. 7g (i)) and 43.1% in India in a study of anemia, in feces of adults at a prevalence of 45.7% in China in a study of systemic lupus erythematosus and depression, 9.2% in South Korea in a study of depression and anxiety (Fig. 8a (iv–vi)), 36.5% in Norway in a study of chronic alcohol overconsumption (Fig. 7f), and, in colonic mucosal samples, in 14.6% of children in the USA in a study of inflammatory bowel disease and 63.2% of adults in Australia in a study of colorectal cancer (Fig. 8b (i, ii)).

The median SFB relative abundance in fecal samples was generally low in adults, between ~0.0001% and 0.006%, with a median 16S rRNA read number of only one or two reads per sample and thus near the detection limit (Fig. 7d–f, Fig. 8a (iv–vi)). The SFB relative abundance

also generally did not reach above 0.1%, with an exception of a study on elderberry supplementation in Austria (Supplementary Fig. 28d). The median SFB relative abundances in humans are lower than what can be found in mice, where they were closer to 0.01% or higher, depending on the mouse chow[81]. However, depending on the mouse vendor, SFB can also be at a very low relative abundance[6], and SFB are also known to naturally exhibit a wide range of colonization levels even within apparently identical groups of mice[24]. Taken together, these data reveal that SFB of various lineages can be found as a prevalent, albeit overall minor, member of the gut microbiota in humans in many countries worldwide.

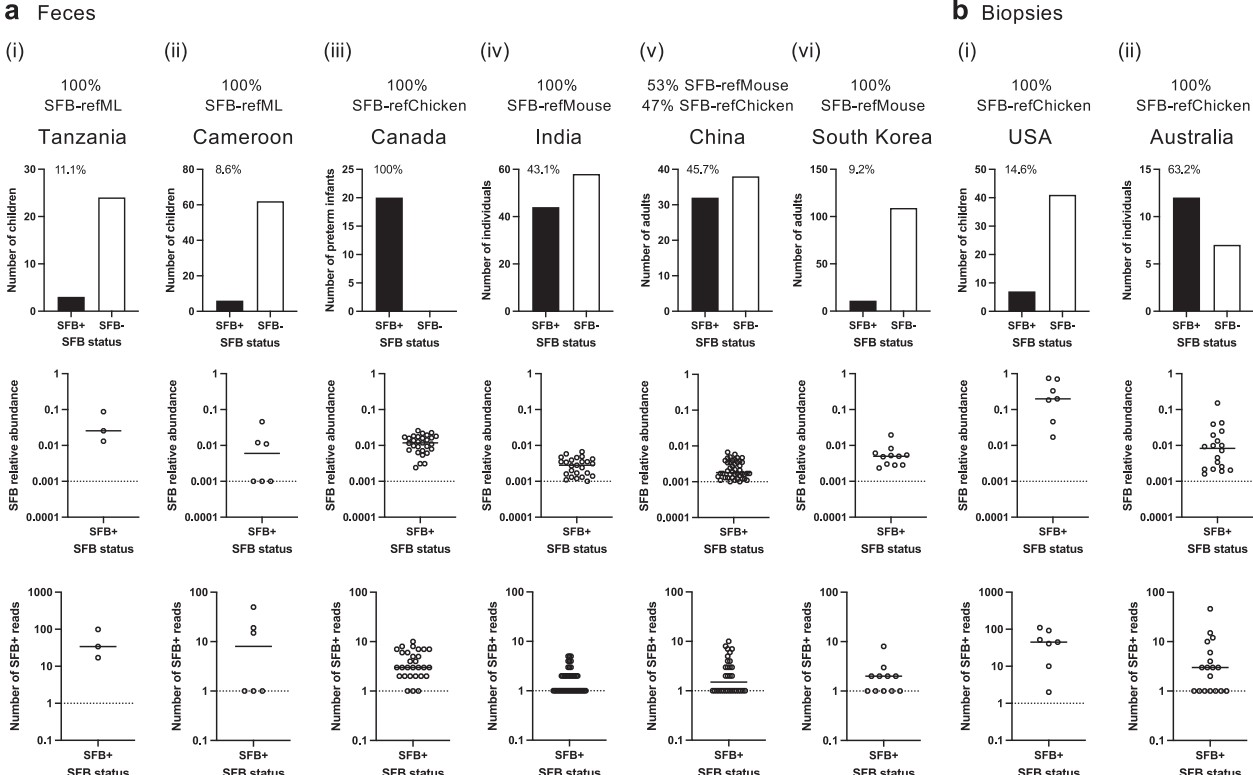

**Fig. 8 | Human SFB prevalence and relative abundance based on 16S rRNA gene amplicon analysis.** Examples of SFB prevalence in various countries based on SFB identification in 16S rRNA gene amplicon datasets from **a** fecal samples or **b** colonic biopsies. Included is the percentage of SFB-positive individuals and, per sample, the SFB relative abundance and 16S rRNA read number in studies of: **a**(i) chronic whipworm infection (TZ, PRJNA629760, number of SFB-positive children $n = 3/27$), **a**(ii) age and diet (CM, PRJNA1048169, number of SFB-positive children $n = 6/70$), **a**(iii) premature infants (CA PRJNA506262, number of children $n = 20$, number of SFB-positive fecal samples $n = 29$), **a**(iv) anemia (IN, PRJNA786061, number of SFB-positive individuals $n = 44/102$), **a**(v) chronic lupus erythematosus and depression (CN, PRJNA883162, number of SFB-positive individuals $n = 32/70$), **a**(vi) probiotics and mental health (KR, PRJNA678145, number of SFB-positive individuals $n = 11/120$), **b**(i) inflammatory bowel disease (US, PRJNA284397, number of SFB-positive children $n = 7/48$), and **b**(ii) colorectal cancer (AU, PRJNA487683, number of SFB-positive individuals $n = 12/19$, number of SFB-positive samples $n = 18$).

**a, b** Proportion of samples of a given SFB lineage is included. Crossbars indicate the median value of the datasets. A dotted line is included at a relative abundance of 0.001% and a read number of 1 for reference purposes.

## Human SFB colonization peaks within the first five years of life

In mice, rapid SFB colonization starts at the time of weaning at around 3 weeks of age, peaks at about 4.5 weeks of age, and then decreases again to low levels within another week[23]. In longitudinal cohorts in Malawi, similar SFB colonization dynamics were identified in fecal samples in two independent studies of children of up to 18 (Supplementary Fig. 28g) and 30 (Fig. 9a) months of age[82,83]. In the more comprehensive study (Fig. 9a (i)), SFB prevalence was zero at 1 month, low at 0.4% by 6 months, increased significantly to 10.8% by 12 months, peaked at 13.3% by 18 months, and significantly decreased to 2.6% by 30 months. Overall, 30.4% of children sampled at the 12, 18, and 30-month time points were SFB positive, with a median SFB relative abundance of 0.01%, a median SFB 16S read number of six, and a median age of 18 months (Fig. 9a (ii–vi)). The SFB relative abundance and read number are similar to what was observed in the Canadian babies (Fig. 8a (iii)), who were all colonized by Human-SFB-refChicken lineage rather than the Human-SFB-refML lineage found in Malawi. These SFB colonization dynamics are also supported by longitudinal studies in geographically distant regions. A similar trend of increasing SFB prevalence and SFB relative abundance was identified in fecal samples in children up to 2 years of age in Peru[84] (Fig. 9b), harboring the Human-SFB-refML lineage, and in children up to 2.5 years of age in Finland, harboring the Human-SFB-refSE lineage (Fig. 9c)[77]. In Peru, where the median SFB relative abundance was 0.008% and median age of the children at the time of SFB detection was 21 months (Fig. 9b (iv, vi)), the weaning status of the children was recorded at each sampling.

SFB detection corresponded to the period when children were only partially breastfed and fully weaned (Fig. 9b (vii)), linking SFB colonization specifically to the late weaning period in humans. In Finland, the median SFB relative abundance was 0.003% and the median age of the children at the time of SFB detection was again 18 months (Fig. 9c (iv, vi)), while SFB detection in infants was restricted to those born by vaginal birth with no difference in SFB prevalence in infants to mothers who received intrapartum antibiotics (Fig. 9c (vii)). SFB prevalence in Finnish infants was overall relatively low at 1.1%, but significantly higher than in parental samples (Fig. 9c (ii)). In agreement, the Human-SFB-refSE genome could only be identified at 24 months, and not at the sampling time at birth or at 12 months, in the longitudinal study in New Zealand (Supplementary Fig. 26a, Supplementary Data 5)[76].

In a separate longitudinal study, with an even greater temporal detail, where children were sampled on a monthly basis from birth up to two years of age, or for Bangladesh up to five years of age[85], SFB-positive fecal samples were detected at a median age of 16, 20 and 19 months in India, Peru and South Africa (Fig. 10a (i)), respectively. No SFB 16S rRNA gene sequences were found in Brazil, while in Bangladesh, the median age of SFB appearance was significantly older at 41 months (Fig. 10a (i))[85]. Notably, the weaning period has been described to be particularly long in Bangladesh[85]. The mean maximal fecal SFB relative abundance in SFB-positive children was again approximately 0.01% (Fig. 10a (ii)) and, despite a generally low detection capability of ~0.003% and a limited sampling period of two years, SFB prevalence was high at 22.7% for Peru, 14.3% for India, and 57.1% for

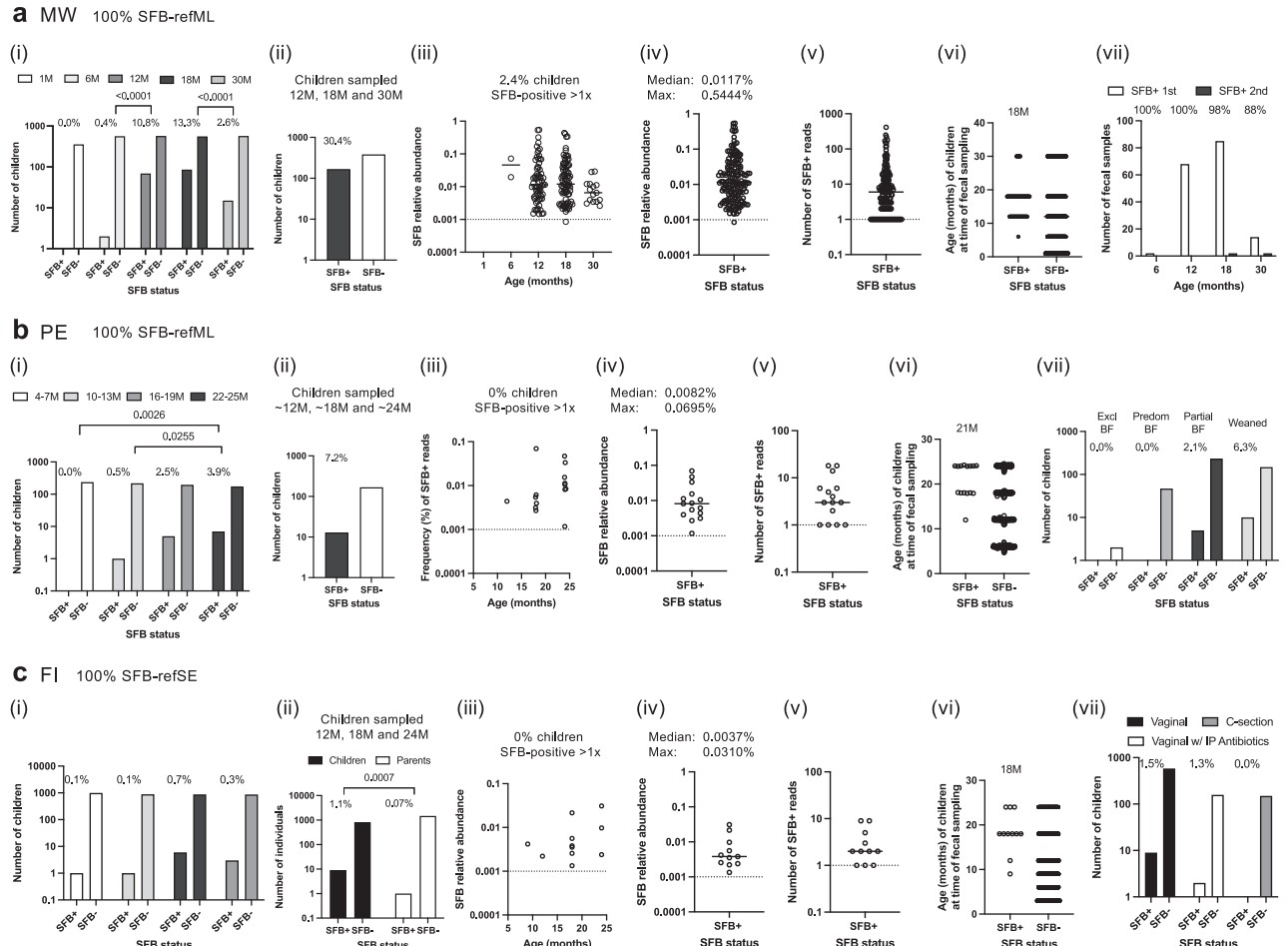

**Fig. 9 | Time-course of Human SFB colonization dynamics during early childhood based on 16S rRNA gene amplicon analysis.** a–c(i-vi) Analysis of fecal samples for a–c(i/ii) SFB prevalence a–c(i) per time point, or a–c(ii) overall (including percent SFB-positive individuals); a–c(iii/iv) SFB relative abundance as compared to a–c(iii) the child's age or a–c(iv) independent of age and including the median and maximum SFB relative abundances; a–c(v) SFB 16S rRNA read numbers; a–c(vi) Age of child when fecal samples were SFB-positive or negative, including the median age. **a** PRJEB29433, a longitudinal study of infant microbiota and their association with health in Malawi. Number of SFB-positive children: 1 M $n = 0/355$, 6 M $n = 2/566$, 12 M $n = 69/640$, 18 M $n = 86/645$, 30 M $n = 15/586$; number of children SFB-positive and sampled at 12 M, 18 M and 30 M $n = 174/552$; number of SFB-positive samples: 1 M $n = 0$, 6 M $n = 2$, 12 M $n = 69$, 18 M $n = 87$, 30 M $n = 15$. **a**(vii) Number of SFB-positive fecal samples at different time points with samples separated by the number of times (1st or 2nd) a child was SFB-positive, including the percent of SFB-positive samples belonging to a child's 1st positive fecal sample. **b** PRJEB28159, a longitudinal birth cohort study in children with a high burden of diarrhea and stunting in Peru; **b**(vii) SFB prevalence with respect to breast feeding (BF) status. Excl: exclusive, Predom: predominant. Number of SFB-positive

children: 4–7 M $n = 0/238$, 10–13 M $n = 1/219$, 16–19 M $n = 5/201$, 22–25 M $n = 7/181$; number of SFB-positive children sampled at ~12 M, ~18 M and ~24 M $n = 13/181$; number of SFB-positive children sampled at exclusive BF $n = 0/2$, predominantly BF $n = 0/47$, partial BF $n = 5/239$, weaned $n = 10/158$. SFB-positive samples $n = 15$. **c** PRJEB55243, a longitudinal study on microbiota variation during early development in Finland. Number of SFB-positive children: 1–9 M $n = 1/996$, 12 M $n = 1/888$, 18 M $n = 6/885$, 24 M $n = 3/877$; number of SFB-positive children sampled at 12 M, 18 M, and 24 M $n = 9/830$; number of SFB-positive adults $n = 1/1469$; number of SFB-positive children in **c**(iii-vi) $n = 11$. **c**(vii) SFB prevalence in fecal samples of children born by vaginal delivery or C-section, including percent SFB-positive children. Number of SFB-positive children with vaginal birth $n = 9/592$, C-section $n = 2/160$, and vaginal with intrapartum (IP) antibiotics $n = 0/150$. **a–c** Country of sample origin is indicated by its 2-letter country code. Proportion of samples of a given SFB lineage is included. **a, b**(i) P-values were obtained with the two-sided Fisher's exact test. Crossbars indicate the median value of the datasets. A dotted line is included at **a–c**(iii, iv), a relative abundance of 0.001%, and **a–c**(v), **a** read number of 1 for reference purposes.

South Africa due to the high sampling frequency (Fig. 10a (v)). SFB prevalence was 21.8% for Bangladesh, but 0% when only considering the first two years (Fig. 10a (i/iv)). A lack of SFB detection in Brazil may therefore also be potentially due to the limited 2-year sampling period. These data reveal an SFB colonization peak in humans at the end of the weaning period and within the first one to two, but up to 5 years of life.

### Human SFB have a short but potent colonization peak
All longitudinal studies suggest a short colonization peak of SFB in humans. In the studies in Malawi, Peru, and Finland that sampled the fecal microbiota at ~6-month intervals using a standard sequencing

depth (~50,000 reads/sample), no child was positive at more than one time point in the Peruvian (Fig. 9b), Finnish (Fig. 9c), and smaller Malawian cohort (Supplementary Fig. 28g). In the larger Malawian study that included 168 SFB-positive children sampled at the three most relevant time points (12, 18, and 30 months), 97.6% of the children were SFB-positive at only one time point (Fig. 9a (iii, vii)). This general trend was conserved for the cohorts sampled monthly in India, Bangladesh, Peru, and South Africa. When children were SFB-positive more than once, the maximum number of SFB-positive time points was three, and the samples clustered together in time (Fig. 10a (v), Supplementary Fig. 28h (i–iii)). Together, these data reveal a colonization peak of human SFB with a duration of approximately one month.

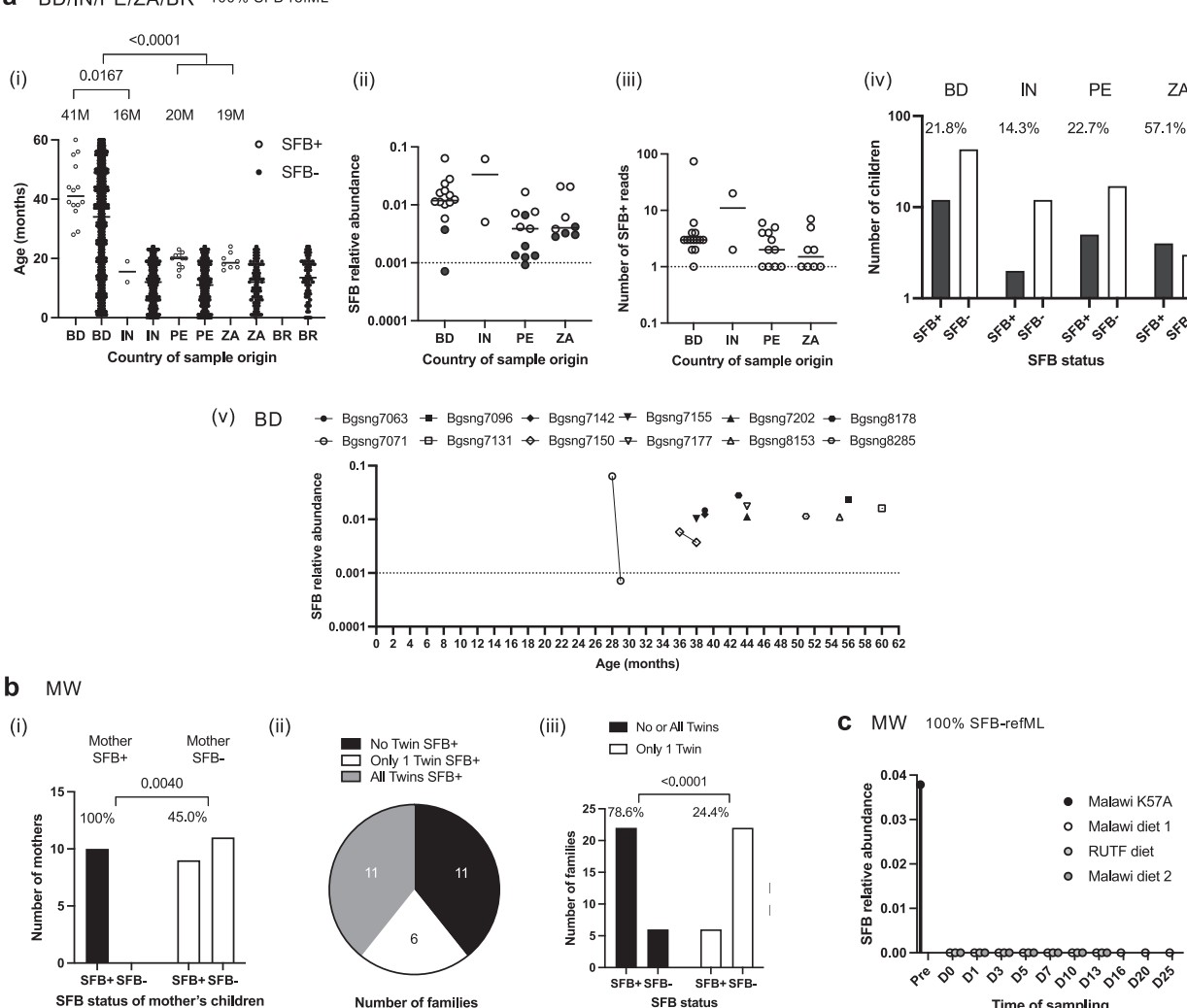

**Fig. 10 | Human-SFB-refML colonization dynamics, transmission, and species-specificity based on 16S rRNA gene amplicon analysis. a** PRJEB27068, fecal samples from a study of healthy and impaired human gut microbiota development with monthly fecal sampling of infants. Number of SFB-positive children: Bangladesh (BD) $n = 12/53$; India (IN) $n = 2/14$, Peru (PE) $n = 5/22$, South Africa (ZA): $n = 4/7$, and Brazil (BR) $n = 0/7$. Number of SFB-positive samples: BD $n = 14/2441$, IN $n = 2/329$, PE $n = 11/477$, ZA $n = 8/140$, and BR $n = 0/124$. **a**(i) Age, including median age in months (M), at which SFB-positive and negative fecal samples were sampled. **a**(ii) SFB relative abundance, with open circle indicating the fecal sample with the highest SFB 16S rRNA relative abundance per individual and the filled circle the additional SFB-positive fecal samples; **a**(iii) Number of SFB 16S rRNA reads in all fecal samples; **a**(iv) SFB prevalence per country; **a**(v) SFB relative abundance in fecal samples compared to the age of the child for the twelve SFB-positive children in the Bangladesh cohort. Fecal samples from a single individual are connected. *P*-values for **a**(i) were obtained with the two-tailed Mann–Whitney exact test. **b** PRJEB3079,

fecal samples from a study on microbiota diversity including twins and mothers in Malawi; **b**(i) Prevalence of SFB-positive and negative mothers with no twin SFB-positive versus at least one SFB-positive twin and including the percent of mothers with at least one SFB-positive twin (SFB-positive/negative mothers $n = 10/20$); **b**(ii) Number and **b**(iii) prevalence of families with twin groups concordant and discordant for their SFB status including in **b**(iii) the percent of SFB-positive mother/twin groups ($n = 28$ families). *P*-value for **b**(i) was obtained with the two-sided Fisher's exact test. **c** PRJEB3324, a study on Kwashiorkor discordant twins in Malawi. Fecal material from SFB-positive donor K57A (pre) was transplanted into mice put on three different diets, and mouse feces was sampled over a 25-day (D) period. Country of sample origin is indicated by its 2-letter country code. **a**–**c**(iii-vi) Crossbars indicate the median value of the datasets. A dotted line is included at **a**(ii/v) a relative abundance of 0.001% and **a**(iii) a read number of 1 for reference purposes.

The colonization peak of human SFB is also potent. In Bangladesh, the child with an SFB relative abundance of 0.054% was positive for SFB the following month at a relative abundance of 0.0007%, or a 77-fold decrease, due to an unusually high detection limit of 0.0002% for this particular sample (Fig. 10a (v)). Similarly, in the Malawian study, several children with fecal samples containing SFB at the relatively high relative abundance of ~0.5% were no longer SFB-positive 6 months later (Fig. 9a (iii)). Given a detection limit of around 0.002% in this study, the decrease in SFB frequency in a 6-month period is therefore at least 270-fold. However, the median SFB relative abundance in feces outside of the early colonization peak in the Malawian population is closer to

0.0001% (Fig. 7e (ii)), and an SFB relative abundance of 0.01–0.1%, and up to 0.54%, is commonly detected in Malawian children at the colonization peak (Fig. 9a (iii)). This suggests that the SFB relative abundance is generally 100–1000, but up to 5400 times greater at the short-lived maximum colonization peak than outside it. Together, these data reveal a potent colonization peak in children of several orders of magnitude as compared to SFB levels outside the peak.

## Human SFB transmission and species-specificity
Bioinformatic analysis furthermore provided clues to possible factors influencing human SFB transmission. In one of the seminal studies in

Malawi[80], data collection included twins and fecal analysis of mothers. Due to the unusual ultra-deep sequencing depth (Fig. 7e), 10 of the 30 mothers that had at least one twin sampled were SFB-positive. SFB-positive mothers were significantly more likely to have at least one twin that was SFB-positive compared to SFB-negative mothers (Fig. 10b (i)). In addition, only 21% of the 28 families that included at least two twins were discordant for SFB detection, and twins were significantly more likely to be either both positive or negative for SFB (Fig. 10b (ii, iii)). These data support a mode of SFB transmission from mother to child.

In animals, SFB colonization is species-specific, most likely due to a lack of attachment of SFB to the intestinal epithelium of non-cognate host species[7,22,86]. In a study where fecal material originating from Malawian children was transplanted into germfree mice put on three different diets and followed over more than a 3-week period[87], one of the donor feces (K57A) had a relatively high SFB-human-refML relative abundance (0.04%), but no SFB 16S rRNA reads were identified in any of the fecal samples tested in the transplanted mice (Fig. 10c). These data are consistent with a species-specific colonization potential of Human-SFB-refML.

## Discussion

SFB are gut commensals found in a range of vertebrate species, but their description remains limited. Here, we characterized the human SFB species Human-SFB-ML that we named *Anisomitus miae* and linked its characteristic SFB morphology to its phylogenetic positioning within the SFB clade, firmly establishing the existence in humans of a SFB species with a morphology and genome composition similar to SFB from rodents. The Human-SFB-ML is a previously undescribed SFB species; its full-length 16S rRNA gene sequence shares less than 97% nucleotide identity with previously published 16S rRNA gene sequences, and its whole-genome ANI is 79% or less to all other SFB genomes published. This confirms the existence of multiple SFB species in the gut microbiota of the human population.

Our 16S rRNA gene amplicon analysis indicates the existence of at least six SFB lineages and thereby a striking diversity in SFB in the human population. Human-SFB-refML is the dominant human SFB lineage in all investigated African countries except Egypt. This is based on the minimum 99.9% identical full-length (~1200 bp or more) SFB 16S rRNA gene sequence obtained from fecal samples from Mali, The Gambia, Kenya, Tanzania, Zimbabwe, Ethiopia, Niger, Uganda, Madagascar, Malawi, and the Republic of the Congo as well as the 100% sequence identity in the V1-V2, V4 and/or V3-V4 16S rRNA gene amplicon datasets across Africa. The Human-SFB-refML lineage may be an ancestral SFB lineage, as the other SFB lineages were not only strikingly scarce in African countries south of the Sahara, but we also identified the Human-SFB-ML genome sequence in the gut microbiota of traditional cultures such as the Hadza hunter-gatherers[72] (Supplementary Fig. 23b, Supplementary Data 2). Human-SFB-refML as the ancestral lineage is also in line with the evolutionary history of humans, their migration patterns, as well as the distribution of another host-associated gut bacterium, *Helicobacter pylori*, that was traced to African ancestral origins with repeated out-of-Africa expansions across the world[88]. Our identification of the Human-SFB-refML lineage in South America[80,84,89], including in the indigenous Tsimane horticulturalists of the Amazon in Bolivia, children from the Amazonian lowlands in Peru, and individuals from the Amazonas Guahibo Amerindians in Venezuela, is also intriguing (Supplementary Data 4). It is in contrast to the Human-SFB-refML lineage being largely absent in the SFB-positive individuals thus far identified living in Europe, Asia, Oceania, and North America, with an exception of Bangladesh (Fig. 5e, Supplementary Data 4). While more extensive 16S rRNA gene sequence and genomic data is necessary to obtain a more detailed understanding of the Human SFB lineage composition, diversity, and distribution patterns, the current data stimulates intriguing hypotheses. As such, we hypothesize that the Human-SFB-refML lineage may have become replaced, or outcompeted, by SFB from other lineages in individuals outside of Africa. In addition, we hypothesize that the presence of the Human-SFB-refML lineage in South America may be a result of the maintenance of the Human-SFB-refML lineage in the South American population after human migration from Africa. Future investigations, including greater 16S rRNA gene sequence and genomic sequence data, should shed further light on these hypotheses.

For SFB of animal origin, the phylogenetic tree shows some clustering consistent with the phylogenetic relatedness of the host. For example, the 16S rRNA gene sequences originating from different varieties of ray-finned fish (trout, nibea, mackerel, and pinfish; Class Actinopterygii) clustered together in a distinct clade, as did SFB sequences from a Hound dog and a sealion (Order Carnivora), a pig and a dolphin (Order Artiodactyla), and those of mice and rats (Order Rodentia, Family Muridae). As the clustering of dog and sealion SFB, as well as pig and dolphin SFB, is not easily explained through sharing of the same environment, the data is supportive of a potential co-evolution with its host. However, multiple datapoints are also consistent with the transfer of SFB between animals and between animals and humans. For one, we identified the Turkey-SFB-US genome, at a 99% ANI and 68% genome coverage, in a human paleofecal sample originating from Arizona (Fig. 6h). These data provide evidence for the presence of a similar SFB species in humans as found today in turkeys. Sharing of a potentially closely related SFB between humans and animals is furthermore supported by the high nucleotide sequence identity of 99–100% for the refChicken, refMouse, and refRat SFB lineages identified in some samples in the V3-V4 16S rRNA gene region, and 98% and 100% nucleotide identity for refChicken and refMouse lineages identified for samples in the V1-V2 16S rRNA gene region. While more sequencing data may decrease their relatedness to the reference SFB from animal origin, in animals, sharing of a relatively closely related, or potentially the same SFB species, between different host species is also supported by the greater than 99% nucleotide identity shared between the 1366 bp of Gorilla-SFB-US 16S rRNA gene sequence and mouse SFB and 970 bp Macaque-SFB-JP 16S rRNA gene sequence and rat SFB (Supplementary Fig. 3, Supplementary Fig. 6, Supplementary Data 4)[50]. We therefore hypothesize that SFB may have both co-evolved with its host and, at times, crossed over into different hosts, potentially through sharing of the same environment. More 16S rRNA gene and genome sequence data should strengthen or refute these hypotheses in the future.

Our data show that SFB are gut commensals in humans worldwide. SFB was detected in humans from forty-four countries and at a high prevalence in a number of studies even in feces at a regular sequencing depth. In intestinal biopsies, where SFB could be at a particularly high relative abundance, SFB was identified in 15% of adults in the USA and a striking 63% of adults in Australia. Notably, current SFB prevalence calculations are likely to underestimate the true SFB prevalence as it is dependent on the sequencing depth. For one, ultra-deep sequencing revealed that regular sequencing depths, particularly in fecal samples, only captures the 'tip of the iceberg' in terms of SFB prevalence as the SFB relative abundance outside of the colonization peak during childhood appears to lie below the detection limit reached by regular sequencing depth. The sequencing depth limitation was also apparent in longitudinal studies of children where SFB became detectable only for a short time interval. At the same time, the data does point towards potential regional differences in SFB frequency and/or prevalence. For example, SFB was undetectable in ultradeep-sequenced fecal samples from mostly adult individuals in three metropolitan areas in the USA (Supplementary Fig. 28f). However, SFB was detected in intestinal biopsies of children (Fig. 7b (i), Fig. 8b (i), Supplementary Fig. 28b) as well as in fecal samples from adults in other bioprojects from the USA (Supplementary Data 4). These results are consistent with the detection of SFB in American children using qPCR analysis with SFB-specific primers[38], a methodology capable of

detecting bacteria at a relative abundance of 0.00001–0.000001% in typical fecal samples of $10^{11}$–$10^{12}$ bacteria/gram[90]. Overall, our data show that SFB are present in individuals in many countries worldwide, often at a high prevalence but very low relative abundance, while prevalence calculations underestimate SFB prevalence due to the limited depth of sequencing.

Colonization dynamics of human SFB were similar to those described for mouse SFB. All the Human-SFB lineages could be found at a low relative abundance in feces. A strong (-0.01% in SFB relative abundance) but short colonization peak in children between 1 and 2, but up to 5 years of age was found for the Human-SFB-refML lineage in Malawi, Peru, South Africa, Bangladesh, and India (Fig. 9a (iii), Fig. 9b (iii), Fig. 10a (v), Supplementary Fig. 28g). A similar trend was found for Human-SFB-SE in Finland (Fig. 9c (iii)). Meanwhile, the Human-SFB-refChicken lineage was present at the median maximal SFB relative abundance in premature infants in Canada (Fig. 8a (iii)), supporting a similar colonization peak. This shows consistency in colonization dynamics for these three human SFB lineages and is in agreement with a greater detectability of SFB in children in their first 3 years of life using qPCR analysis in China[36,38], where the Human-SFB-refMouse and Human-SFB-refChicken lineages are particularly abundant. The colonization peak of human SFB showed some geographic variation that may be due to differences in societal breastfeeding habits[85] or other factors generally known to influence the gut microbiota composition. These factors may also influence SFB colonization levels outside the colonization peak, as could potential differences in the fitness of SFB from varying lineages. However, co-colonization of the Human-SFB-refMouse and Human-SFB-refChicken lineages in a single individual, where either of the two lineages could be dominant in individuals from the same region (Fig. 5f), argues against a difference in colonization potential.

In mice, SFB are strong inducers of the weaning reaction and, in the presence of bacterial metabolites such as short-chain fatty acids, SFB colonization leads to a healthy immunological imprinting, while in the absence of these metabolites, SFB promotes an inflammatory phenotype, highlighting an important context-dependent host response to SFB colonization[9]. Here, we link the colonization burst of the Human-SFB-refML lineage during early child development specifically to the late weaning period (Fig. 9b (iii, vii)).

Diet is an important contributor to SFB colonization in mice[15,91] and chickens[92] and may have an impact on SFB colonization in humans[40]. Our bioinformatic analysis suggests that feeding preterm babies with formula or human milk fortifier may favor a rapid expansion of SFB (Fig. 9a (iii)). Dietary interventions may therefore disrupt the normal timing of the colonization peak and may also be capitalized on to modulate SFB levels, as previously shown in mice[15]. Notably, disruption of the normal SFB colonization peak at weaning may be medically relevant given the context-dependent immune activation of SFB observed in mice[9].

The outcome of potential nutritional interventions in humans may be influenced by the SFB species present in an individual, given the unique repertoire of sugar transporters found in the broad phylogenetic groups of turkey SFB and rodent SFB and the two human SFB species. The Human-SFB-ML genome is, for now, the only SFB genome to encode all the necessary enzymes required for the breakdown of glycogen and starch and for the import of degradative sugars such as $\alpha(1,4)$-linked maltooligosaccharides. Human-SFB-ML may thus degrade starch found in the extracellular environment, or potentially extracellular glycogen, as has been demonstrated for the vaginal microbiota[93]. Notably, maltodextrin, a starch derivative consisting of branched, $\alpha(1-4)$ and $\alpha(1-6)$-linked D-glucose polysaccharides, is a common additive in processed foods, including infant formula, serving as fillers, thickeners, texturizer and coating agents[94]. The increase in maltodextrin in the American diet[95] and a mucosal microbiome enriched for maltodextrin metabolism[96] in CD and patients with ileal

CD, but not patients with colonic CD and non-IBD controls, is therefore noteworthy. Alternatively, Human-SFB-ML may be able to access glycogen stores of the host cell cytosol through its intimate attachment to the host.

The morphological characterization and bioinformatic analysis of the colonization dynamics suggest that the life cycle of human SFB is similar to that of rodent SFB. Human-SFB-ML filaments displayed the morphologically smooth and bulbous filament phenotype, the pointed tip that is a defining feature of SFB and mediates attachment to epithelial cells, the apparent ability to develop into IOs within the filament, and the capacity to form spores (Fig. 1d, f, h). The ability of human SFB to intimately attach to the host is supported by the visualization of filamentous bacteria attached to the ileal epithelium in Chinese children[45]. Notably, while 16S rRNA gene amplicon analysis did not lead to a bacterial candidate[45], we identified a 550 bp V3-V4 rRNA gene read (SRR4272058.195935.1) with 100% identity to Mouse-SFB-NL in the publicly available 16S rRNA gene amplicon data of this study, providing support that these filamentous bacteria are SFB of the Human-SFB-refMouse lineage. As would be expected, SFB was detected at a higher prevalence in intestinal biopsies compared to feces, reflecting a greater relative abundance of SFB in biopsies and suggesting tissue tropism. In addition, transmission of SFB to the offspring through the mother is supported by a significantly greater chance of children being SFB positive when SFB was in sufficiently high frequency to be detectable in the feces of their mother, as well as a significantly greater chance of twins being either both SFB positive or negative as compared to being discordant (Fig. 10b).

In mice, SFB can protect the host by promoting colonization resistance against a wide array of pathogens in the gut[3,97] and more recently also in the lung[12,13,98] while also exacerbating disease severity in models including autoimmune encephalitis[18], obesity-induced liver damage[19], autoimmune arthritis[17,20], depression[42], Crohn's disease[15], and autism[16]. Our data identifies SFB as a prevalent group of gut microbes in humans with similar colonization dynamics as seen in mice, providing support for a likely medical relevance of SFB colonization in humans. While the immunostimulatory potential of SFB is complex and its immune activation can be modulated by microbiota metabolites[9], and thus may depend on microbiota composition, the future analysis of the SFB relative abundance and prevalence in terms of disease association should provide further insights into the potential role of SFB in disease in humans.

This study identified a new species of SFB in humans and identifies SFB as a group of commensals present in the gut microbiota of humans worldwide. The study is limited by the low availability of sequencing data, not only sufficient sequencing depth to identify SFB outside of its colonization peak, leading to SFB identification by only a small number of reads in a limited number of individuals, but also sufficient sequencing length to assign SFB reads from various lineages to specific SFB species. Thus, while the current data, based on the 16S rRNA gene V3-V4 region, points towards the existence of at least six SFB lineages in the human population, a more in-depth analysis, with more substantial 16S rRNA gene sequence and genomic data, should reveal the extent of different SFB species and strains in the human population. Notably, the lineage designation may also change and expand when full length 16S rRNA gene sequences become available, given the high sequence variability of the SFB 16S rRNA gene V1-V2 region seen within the reference SFB 16S rRNA gene sequences and the variable placement of the HumanSkin-SFB-US 16S rRNA gene sequence on the phylogenetic tree when considering only the V3-V4 region or the full length sequence (Fig. 1, Supplementary Fig. 5a, b, Supplementary Data 4). Our current understanding of the distribution of SFB lineages in the human population, as well as the contribution of SFB to health and disease, will also undoubtedly evolve as more sequence data is analyzed and becomes available. Further work will be necessary to obtain a better understanding of the fitness and immunostimulatory

potential of the different SFB lineages in the human population. However, given all the similarities to mouse SFB we identified for the human SFB lineages, as well as their global distribution, SFB are likely to be important and medically relevant gut commensals in humans worldwide.

## Methods

### Morphological analysis of human and mouse fecal samples

Fecal samples from Mali, The Gambia, and Kenya from the study by Pop et al.[47] were obtained by a Material Transfer Agreement between Institut Pasteur, Paris, France, and the Center for Vaccine Development, Bamako, Mali, The Medical Research Council Unit, Serrekunda, The Gambia, and KEMRI, Kisumu, Kenya. Ethical approval for the fecal sample analysis with participant consent was obtained from the Institutional Review Board under the following Federal Wide Assurance numbers: The Gambia, Medical Research Council Labs FWA 00006873, Kenya Medical Research Institute FWA 00002066, University of Mali Faculty of Medicine, Pharmacy, and Dentistry FWA 00001769. No additional ethical approval was required for the current work. No participant compensation was given for the current work. Sex of the human donors was not a consideration. Frozen fecal samples kept at −80 °C were thawed on ice, and a part of each sample was diluted in PBS, spotted on a slide, and Gram-stained using the Gram-Hueckner kit (RAL Diagnostics, 361520). For fluorescent in situ hybridization (FISH) analysis, fecal samples were diluted and fixed in 4% paraformaldehyde in PBS overnight at 4 °C. Samples were spun down and washed with PBS before their storage in 50% ethanol at 4 °C for up to 5 days. Fixed fecal samples were washed once with PBS, spotted onto glass slides, and allowed to air-dry. The spotted regions were encircled with a fatty pen, permeabilized with 40 U/ml mutanolysin (Sigma, M9901-5KU) in PBS for 30 min at 37 °C, washed with PBS, and incubated in hybridization buffer (20 mM Tris-HCl [pH8], 0.9 M NaCl, 0.01% SDS) at 50 °C for 20 min. The hybridization buffer was replaced with a hybridization buffer containing 50 nM of an SFB-specific 16S Cy3-labeled probe (5′-GGG TAC TTA TTG CGT TTG CGA CGG CAC-3′)[54] and an A488-labeled all bacteria EUB338 probe (5′-GCT GCC TCC CGT AGG AGT-3′)[99], which were synthesized by Invitrogen, and incubated overnight at 50 °C in a dark, humid chamber. After a wash with the hybridization buffer for 15 min at 48 °C, slides were briefly washed with water, stained for 3 min with 0.125 µg/ml 4′,6-diamidino-2-phenylindole (DAPI), washed again with water, and air-dried before coverslips were mounted with ProLong Gold (Thermo Fisher P36930). Images were taken on an Olympus IX81 fluorescence microscope equipped with a CCD camera and processed using ImageJ (colors were reversed for better visualization). For scanning electron microscopy (SEM) analysis, part of the fecal samples were diluted in PBS, centrifuged at 8000 × g for 3 min, resuspended in 0.1 M sodium cacodylate buffer (pH 7.2) containing 2.5% glutaraldehyde, and fixed overnight at 4 °C. Samples were then washed in 0.1 M sodium cacodylate buffer and suctioned onto 13 mm polycarbonate filters with 0.1 µm pores (VCTP01300, Millipore). Samples were post-fixed in the dark with 1% osmium tetroxide at room temperature (RT) for 1 h, dehydrated in a series of graded ethanol baths, dried using critical-point drying (CPD300, Leica), coated with gold/palladium, and observed with a JEOL 6700 F scanning electron microscope. To grow Mouse-SFB-NL in mice, experiments were performed in accordance with French and European regulations on the protection of the animals used for scientific purposes (Directive 2010/63 of the European Parliament and French decree of February 1, 2013). The mouse experiment (dap210054) was approved by the Institut Pasteur ethical committee for animal experimentation (CETEA, registry number #89) and authorized by the Ministère de l'Enseignement Supérieur et de la Recherche. Mouse-SFB-NL was recovered from monocolonized C57BL/6 mice and purified from feces and intestinal and cecal content using Nycodenz gradients[52] before being stained with the Gram-Hueckner kit

(RAL Diagnostics, 361520) and imaged with an Olympus IX81 microscope. The sex of the mice was not considered, as the mice were not used for experimentation but as a means to propagate SFB.

### Metagenomic chromosome conformation capture (Meta3C) libraries generation and sequencing

Frozen fecal samples were thawed, and 500 mg of each sample was resuspended in 200 mL of crosslinking solution of 1× PBS with 5% formaldehyde (Sigma Aldrich−Formaldehyde 36.5% in Methanol 15%) and incubated for 1 h at RT under gentle agitation. Formaldehyde was quenched with 40 mL of 2.5 M glycine for 20 min at RT under gentle agitation. Samples were recovered by centrifugation (4500 × g for 10 min), washed with 10 mL PBS, re-centrifuged, and aliquots of 100 mg were stored at −80 °C until use. One aliquot (100 mg) of fecal matter for each experimental sample was resuspended in 8 mL 1× Tris-EDTA pH7,4 supplemented with antiprotease (mini tablets−Roche), transferred into four Precellys tubes (2 mL−VK05 supplemented with 100 mL of VK01 glass beads) and disrupted (6700 rpm−20 s ON/30 s OFF−six cycles). Lysates were recovered, pooled, and 40 µL of Ready-to-lyse lysozyme (Epitech) was added prior to a 1 h incubation at 37 °C under gentle agitation. Sodium dodecyl sulfate (SDS) at 10% was added to a final concentration of 0.5%, and the lysate was incubated 10 min at RT. For each library, 4 mL of lysate was transferred to a tube containing the digestion reaction solution (2 mL NEB1 10X buffer, 2 mL Triton 10%, 4000 U HpaII or MluCI, H2O, final volume of 16 mL). Digestion was allowed to proceed for 3 h at 37 °C under gentle agitation. Tubes were centrifuged for 20 min at 4 °C and 16,000 × g, and the supernatants discarded. Pellets were resuspended and pooled in a final volume of 500 µL of water. The 500 µL were then transferred to a tube containing the ligation reaction (160 µL NEB ligation buffer 10×, 16 µL ATP 100 mM, 16 µL BSA 10 mg/mL, 500 U T4 DNA ligase, final volume 1.1 mL). Ligations were processed for 4 h at 16 °C; 20 mL EDTA 0.5 M, 80 mL SDS 10%, and 2 mg proteinase K were added to each reaction and incubated overnight at 65 °C to digest proteins. DNA was extracted using phenol-chloroform and precipitated with 2.5 vol 100% ethanol. Pellets were suspended in a final volume of 130 mL TE 1X supplemented with RNAse, incubated 1 h at 37 °C, and stored at −20 °C until use. Libraries were then processed for sequencing as described[100–102]. Proximity ligation libraries were sequenced using paired-end (PE) Illumina sequencing (2 × 65 bp, NextSeq500 apparatus)[102].

### Meta3C data processing, binning procedure, and genome scaffolding

Reads from the two libraries were filtered and trimmed using cutadapt[103]. Quality was controlled with FastQC (Andrews, 2010; http://www.bioinformatics.babraham.ac.uk). The retained PE reads were used independently to perform a draft assembly and the binning procedure using the metaTOR pipeline and default parameters (https://github.com/koszullab/metaTOR)[100,102]. The different MAGs obtained were then blasted against the mouse SFB genomes as reference to identify those corresponding to human SFB. The identified MAGs were then used as a template to scaffold the genomes using the GRAAL algorithm[100]. A discrete cluster of sequences, comprising 984 contigs for the S340 (Human-SFB-ML-1) and 526 contigs for S195 (Human-SFB-ML-2), was obtained. The S340 draft genome sequence (Human-SFB-ML-1) underwent more stringent refinement, resulting in 119 unique contigs.

### RNA bait-mediated SFB genome capture and Illumina and Pac-bio sequencing

A set of overlapping 99,922 120 bp-long RNA baits were designed and generated by ArborBiosciences based on the genome sequence of eight SFB (with the omission of small and large subunit ribosomal RNA genes islands) including five mouse SFB (SFB-YIT NCBI RefSeq

assembly #GCF_000284435, SFB-Japan NCBI RefSeq assembly #GCF_0002270205, SFB-NL NCBI RefSeq assembly #GCF_000709435, SFB-NYU NCBI RefSeq assembly #GCF_000225365, and SFB-SU NCBI RefSeq assembly #GCF_000252785), one rat SFB (NCBI RefSeq assembly #GCF_000283555), one turkey SFB (NCBI RefSeq assembly #GCF_001655775) and the draft human Mali SFB S340 sequence generated by chromosome conformation capture. Sequences were first RepeatMasked for low-complexity repeats only (soft-masked), and strings of 10 or fewer N's were replaced with T's. Baits were designed with 2x tiling density (28,076 for human and 55,565 for non-human). For the non-human baits, those exhibiting 90% identity across 75% overlap were clustered and collapsed to one sequence. Baits were BLASTed against the human hg38 genome, and 6 baits with one or more blast hits were removed. Stringency was furthermore increased by removing 352 baits with >50% soft-masking. To overcome bait complementarity issues, all baits were mapped to the human draft genome sequence, and any that mapped in the reverse orientation were reverse-complemented.

The genomic DNA from the human Mali fecal samples was extracted using the PowerFecal DNA kit (Qiagen 51804) and amplified in quadruplicate using the whole-genome amplification Midi Repli-G kit (Qiagen 150023). Amplified samples were pooled and sent to Dalhousie University Integrated Microbial Resource for full-length 16S rRNA gene sequencing using the Sequel sequencer and to ArborBiosciences for bait capture, using the Mybaits manual version 3, and myReads library construction and 125 bp paired-end Illumina sequencing using the HiSeq2500 sequencer. For Pacbio metagenomic sequencing, whole genome amplification (WGA) products were purified using Ampure XP beads (Beckman A63880), quantified, and quality controlled using the Bioanalyzer DNA 12000 kit. 2 μg of purified WGA products were used for pre-capture library preparation using the Kapa HyperPrep kit (Roche). Samples were end-repaired, and linear barcoded adapters for Pacific Biosciences sequencing were ligated. After ligation, samples were purified and amplified using a Pacific Biosciences universal primer. Purified and amplified pre-capture libraries were quantified and qualified using the Bioanalyzer DNA 12,000 Kit (Agilent 5067-1508). Capture of the samples was made by hybridizing 2 μg of each sample for 40 h, with the custom biotinylated RNA baits. After hybridization, the RNA/library complex was captured with streptavidin beads. After capture, all samples were amplified and quality controlled with the Bioanalyzer DNA 12000 kit (Agilent 5067-1508). Hairpin adapters were added to the captured samples using the STMRTbell express Template prep kit 2.0 (PacBio 100-938-900). Single-strand overhangs were removed, and DNA damage repair and End repair were performed. The extremities of the DNA fragments were a-tailed, and the hairpin overhang adapters ligated. Final libraries were quantified, qualified, and multiplexed in a single pool to be sequenced on a Sequel I sequencer.

## Data pre-processing

Reads of likely human origin were identified and removed, as recommended by Human Microbiome Project[104] with BMTagger v3.101[105] using a standard human genome reference (GRCh37.p5[106]) to both protect subjects' privacy and focus exclusively on the microbial community in each sample. Stringent quality control using Trimmomatic v0.36[107], in which the Illumina adapters were excised, reads were trimmed using a 4 bp sliding window with an average quality score threshold of Q15, and reads containing any ambiguous bases were removed. Sequenced ribosomal RNA was removed in silico by aligning all reads using Bowtie v1[108] against the SILVA PARC ribosomal-subunit sequence database[109]. Both read pairs were filtered out if at least one read mapped to either the human genome or the rRNA database. Raw reads with low-quality bases as determined by base calling (phred quality score of 20) were trimmed from the end of the sequence, and only sequences longer than 75 bp were retained.

## Metagenome assembly and genome analyses

De novo assembly of Illumina reads was performed using metaSPAdes (v3.11) with default settings[110,111]. De novo assembly of Pacbio reads was performed using canu (v1.8) with default settings[112]. Mauve (Multiple Alignment of Conserved Genomic Sequence With Rearrangements) and its contig reorder tool were applied[113,114]. Unalignable contigs were searched through the NT database and shown to also include SFB sequences with the greatest similarity to the Mouse-SFB-NL genome, the mouse SFB strain studied in the lab. The absence of Mouse-SFB-NL genome sequences in the Meta3C experiment confirms its introduction during whole genome amplification of the human Malian fecal samples. Due to the sequence divergence between the mouse and human SFB genomes, this contamination did not interfere in the human SFB genome assembly. The alignable and reordered contigs of the human SFB genomes were scaffolded using a validated improvement pipeline[115] to extend the length and enhance the quality of the resulting assemblies. This pipeline incorporated an iterative assembly improvement step to scaffold the contigs using SSPACE v2.1.1[116] and fill in sequence gaps using GapFiller[117]. PacBio long-read assembly was used to guide scaffolding.

Genes were called on the resulting contigs using prokka (v1.14.6)[118] to predict coding sequences (CDSs) with default settings. Eight SFB genomes were used in the analyses, including six downloaded from GenBank: Rat-SFB-YIT (AP012210), Mouse-SFB-NL (CP008713), Mouse-SFB-JP (AP012202), Mouse-SFB-YIT (AP012209), Turkey-SFB-US (UMNCA01), and Human-SFB-SE (ERZ1468256). The pangenome was constructed using Anvi'o v6.2 workflow[119,120]. Briefly, this workflow uses BLASTP to compute ANI identity between all pairs of genes, uses the Markov Cluster Algorithm (MCL)[121] to generate homologous gene clusters (HGCs), and aligns amino acid sequences using MUSCLE[122] for each gene cluster. Homologous gene clusters (HGCs) were identified in this set of genomes based on all-versus-all sequence similarity. Each gene was assigned to the core or accessory gene pool according to the hierarchical clustering of the gene clusters. Functional annotation of each secreted protein was performed employing the eggNOG database v5.0[123] using eggNOG-mapper v2[124], and the results were imported into the Anvi'o contig database. Further functional annotation includes PFAMs based on a hidden Markov model (HMM) search using Pfamvers34.0[125]. Protein-coding genes were also annotated with their metabolic functionally categories using KEGG (Kyoto Encyclopedia of Genes and Genomes)[126]. GhostKOALA annotation tool was used to assign KEGG Identifiers[127]. KEGG orthologs, enzymes, functional pathways, and modules were inferred using KEGG mapper to KEGG BRITE hierarchy information[126,127]. The presence and absence of genomic features from SFB from different hosts were analyzed using the basic BLAST Score Ratio (BSR) approach with default settings[58]. To globally identify differentially present genomic features, a large-scale BLAST Score Ratio (LS-BSR) method was used[128]. A comparison of average BSR values across each genomic region was used to identify genomic features that are more prevalent in specific lineages that are host-specific or universally present. Circular maps of the Human-SFB-ML-1 and ML-2 genomes were generated using proksee[129] following gene annotation of the pre-assembled contigs with prokka. Annotations for the presence of mobile genetic elements (MGEs) were performed using the mobileOG-db database v1.1.3[130] and integrated into the graphical maps.

## Protein expression and purification of MdxE

The mdxE gene without the protein signal sequence was synthesized with Escherichia coli codon optimization and cloned in the plasmid pET151/D-TOPO (GeneArt, Thermo Fisher). Protein expression in the BL21Star E. coli strain was performed at 30 °C in Luria-Bertani broth with an induction of 4 h and 0.1 mM Isopropyl β-d-1-thiogalactopyranoside (IPTG). MdxE was purified by affinity chromatography followed by size-exclusion using a Cobalt 5 mL HiTrap TALON crude column (Cytiva) and a Superdex 200 Increase 10/300 GL

column (Cytiva), respectively. Protein purification was performed using an AKTA-FPLC purification system following the instructions of the manufacturer. Protein-containing fractions were analyzed by sodium dodecyl sulfate–polyacrylamide gel electrophoresis (SDS-PAGE) and western blot using an anti-histidine antibody (A7058, Sigma Aldrich).

## MdxE-sugar binding assay

The interaction of the purified MdxE with potential sugar substrates was determined by surface plasmon resonance using a Biacore T200 instrument (Cytiva) equilibrated at 25 °C in 1× Phosphate-buffered saline (PBS, pH7.4) supplemented with 0.01% Tween20 and 100 μM EDTA. Between 2700 and 2800 resonance units (RU) of MdxE were covalently immobilized on a CM5 Sensor Chip (Cytiva), and an empty flow cell without immobilized protein was used as a reference. The sugar substrates tested including: glucose (Sigma, ref. G8270), trehalose (Sigma, PHR1344), isomaltose (Sigma, 05389), maltose (Sigma, 1059120025), maltotriose (Sigma, M8378), maltotetraose (Sigma, SMB01322), maltopentaose (Sigma, SMB01321), maltohexaose (Sigma, M9153), maltoheptaose (Sigma, M7753), α-cyclodextrin (Sigma, C4642), γ-cyclodextrin (Sigma, G0150000) and sucrose (Sigma, S7903). Each substrate was injected in the flow cell containing the immobilized MdxE (experimental flow cell) and in the reference cell at a flow rate of 50 μl/min. Association was monitored for 300 s and dissociation for 480 s. To assess the specific MdxE-sugar interaction signal, the SPR responses obtained in the reference cell were subtracted to the responses detected in the experimental flow cell. SPR responses are given in resonance units (RU; 1 RU corresponding to the binding of ≈ 1 pg/mm$^2$ of sugar). To determine the MdxE-sugar binding affinity (KD), the following concentrations were used: (i) 3-fold serial dilutions from 22 to 0.8 mM for glucose; (ii) 3-fold serial dilutions from 800 to 3 μM for maltose; (iii) 3-fold serial dilutions from 1500 to 6 nM for maltotriose, maltotetraose, maltopentaose, maltohexaose and maltoheptaose; (iv) 3-fold serial dilutions from 250 to 1 μM for α-cyclodextrin; and (v) 3-fold dilutions from 25 to 0.1 μM for γ-cyclodextrin. KD were calculated by fitting the concentration-dependence of the steady-state SPR responses using the Biacore T200 Evaluation Software version 3.1.

## Phylogenetic analysis

The evolutionary relatedness between the eight SFB genomes was assessed on single-copy core genes (SCGs); 668 gene clusters were filtered using Anvi'o v6.2[131] and cleaned by removing nucleotide positions with gap characters in over 50% of the sequences using trimAl[132]. Phylogenomic analysis and construction of a maximum likelihood tree based on SCGs was performed using the CLC Genomics Workbench software (v23.0.5) with the 'WAG' general matrix model (Whelan and Goldman) and 1000 bootstrap replicates. The average amino acid identity (AAI) and average nucleotide identity matrix of the SFB genomes were calculated with EzAAI (v1.2.1, https://github.com/endixk/ezaai) and FastANI (v1.34, https://github.com/ParBLiSS/FastANI), respectively. The dendrogram using the AAI matrix was constructed with IQ-TREE2 software[133].

For the 16S rRNA gene phylogeny, we first identified 16S rRNA gene sequences in the NCBI database, using the V1-V4 16S rRNA gene sequence region of Mouse-SFB-NL, that fell within the SFB clade. We retained representative 16S rRNA gene sequences from various hosts. When multiple sequences were present from an individual animal (e.g. pinfish, mackerel, shorebird and sealion) we selected the most abundant sequence (Human-SFB-SE contig141_89555, HumanSkin-SFB-US JF168221, Pig-SFB-JP_AB822980, Chicken-SFB-BE_PV993571.1, Chicken-SFB-UK_X80834.1, Chicken-SFB-CN_DQ342328.1, Chicken-SFB-CN_DQ342328.1, Shorebird-SFB-US_KC478326.1, Mouse-SFB-NL_CP008713.1, Mouse-SFB-JP_AP012202.1, Mouse-SFB-YIT_AP012209.1, Rat-SFB-NL_X87244.1, Rat-SFB-JP_D86302.1, Rat-SFB-YIT_AP012210.1,

Macaque-SFB-JP_D86303.1, Gorilla-SFB-US_EU474247.1, Dog-SFB-US_DQ113757.1, Sealion-SFB-US_JQ207968.1, Dolphin-SFB-US_JQ202596.1, Mackerel-SFB-US_JQ191772.1, Pinfish-SFB-US_KJ197471.1, Trout-SFB-FR_AY007720.1 and Nibea-SFB-CN_KX431301.1.). The 16S rRNA gene sequences were aligned with sequences of four closely related *Clostridium* species (*Clostridium acetobutylicum* (NR_074511), *Clostridium saccharobutylicum* (NR_122051), *Clostridium butyricum* (CP040626), and *Clostridium tetani* (X74770)) as an outgroup. Similar results were found when a higher number of outgroup species were used (Supplementary Fig. 4). For unrooted and rooted maximum-likelihood trees, the dominant sequence of each bioproject or SRA was selected. In the absence of a dominant sequence, the sequence with the highest nucleotide identity to the SFB 16S rRNA reference gene was used. Nucleotide alignment was performed using the MAFFT multiple sequence alignment program[134]. The evolutionary history was inferred using the maximum likelihood method with the predicted best Bayesian information criterion (BIC) model, general time reversible (GTR), with parameters + FO using the IQ-TREE2. To compare the phylogenetic relationships among the SFB 16S rRNA gene hypervariable regions, full length MAFFT aligned sequences were trimmed to their respective variable regions.

Phylogenetic analysis was performed for the protein sequence of the SFB nitroreductases and the RNA polymerase subunit beta (RpoB) and beta' (RpoC) of SFB and three closely related *Clostridium* strains, *C. acetobutylicum* (CP002660), *C. saccharobutylicum* (CP006721), and *C. butyricum* (CP033249), taken as an outgroup. Alignment and maximum-likelihood phylogenetic trees were constructed using a "WAG" model in the CLC Genomics Workbench, and branch supports were assessed by generating 100 bootstrap replicates. iTOL (https://itol.embl.de) was used for the visualization of all maximum likelihood trees and whole-genome AAI and ANI.

## SFB identification in human 16S rRNA gene amplicon datasets

Analysis was performed only on bioprojects with a minimum read length of 200 bp but covering either the V1 or V4 16S rRNA gene variable regions. An exception is the inclusion of PRJEB3324 and the ultra-deep sequencing bioproject PRJEB3079, which include smaller reads. Raw FASTQ files and metadata for SFB-positive bioprojects were downloaded either from NCBI Sequence Read Archive (SRA) using sra-tools v3.0.8 or from the European Nucleotide Archive (ENA). Quality control was performed with FASTQC and FASTP (v0.23.4)[135]. The resulting cleaned reads were merged using the default parameters of FLASH2 v 2.2.11[136] and BLASTed locally using NCBI BLAST v2.14.1+ against our database containing full-length 16S rRNA gene sequence from four SFB i.e., Human-SFB-ML, Human-SFB-SE, Chicken-SFB-BE and Mouse-SFB-NL with a parameter of generating an output with reads showing above 94% nucleotide identity and 90% query coverage to our reference sequences for the V4 and V3-V4 variable 16S rRNA gene reads and 91% nucleotide identity and 90% query coverage for the V1-V2 amplicon projects. All sequence reads had SFB as the closest hit in the NCBI non-redundant (nr) sequence database, disregarding hits to uncultured or uncharacterized bacteria and the annotated genome sequence of *Peptoclostridium difficile* BN1097. This genome contained a more than 4000 bp contig (contig93) of the 5S, 16S, and 23S rRNA genes that were 100% identical to the Mouse-SFB-NL rRNA island (Supplementary Fig. 17). This genome assembly from metagenomic data thereby appears to include the incorrect assignment of SFB genomic sequences. The relative abundance of SFB 16S rRNA gene amplicon reads for each SRA was calculated based on the total number of reads per SRA obtained after read cleaning and read merging.

For a bioproject-based analysis, reads were pooled, the unique SFB-positive sequences for each bioproject were identified, and their abundance calculated using the R packages Biostrings v2.68.1 and SeqinR v4.2.30. The dominant sequence was identified, aligned with the full-length SFB reference 16S rRNA gene sequences, and trimmed

to either the V1-V2, V4, or V3-V4 region before Maximum-likelihood phylogenetic analysis (Fig. 5c).

For the SRA-based analysis, the dominant 16S rRNA read per SRA was identified and assigned to one of the 15 SFB reference 16S rRNA gene sequences with the highest nucleotide identity (Supplementary Fig. 16, Supplementary Data 4). This reference SFB read of each SRA was then analyzed using a Maximum-likelihood phylogenetic analysis with the addition of SFB 16S rRNA gene sequences for which the full or nearly full, length 16S rRNA gene sequences were available (Supplementary Fig. 2). Correct SFB lineage designation based on nucleotide sequence identity was verified with the phylogenetic analysis. For eight SRAs, SFB lineage designation was based solely on phylogenetic positioning due to equal nucleotide identity to more than one reference SFB 16S rRNA gene sequence (Supplementary Data 10). Countries were colored based on the detection of SFB presence (Fig. 5b) or based on the SFB lineage with the highest sample number (Supplementary Fig. 21) using R (v4.0.2).

For bioproject PRJEB3079, which had smaller 16S rRNA read lengths, the phylogenetic positioning for the dominant read of all SRAs from Malawi were all within the SFB clade, and for all SRAs from Venezuela were closest to Human-SFB-ML. For Malawi, the majority of 16S rRNA reads, and all those above 100 bp, clustered with Human-SFB-ML, while the phylogenetic positioning for some SRAs below 95 bp showed less reproducible clustering. The Venezuela samples, but not those from Malawi, were therefore included in the more detailed SFB lineage analysis (Supplementary Data 7).

To identify co-existence of SFB lineages within a single SRA, the percent nucleotide identity of SFB reads from all V3-V4-based 16S rRNA gene amplicon SRAs and 15 SFB 16S rRNA gene sequences comprising at least the V1 to V4 16S region was determined (Supplementary Fig. 22). For SRAs with multiple SFB lineages, the 16S rRNA gene read was verified by phylogenetic analysis (Supplementary Fig. 27).

### SFB identification in human 16S multiplex and RNA metagenomic datasets

Raw FASTQ files of SRAs identified to be potentially SFB-positive using PebbleScout[73] were downloaded using CLC genomics workbench v23.0.5. All reads of each SRA were mapped to the full-length 16S rRNA gene sequence of the reference SFB with stringent mapping parameters of 100% nucleotide identity across 100% of the read length. Single-end reads were excluded for further analysis. The consensus sequence was generated from the paired-end reads by the CLC software. BLASTn analysis of the consensus sequence was performed against the NCBI nt database, and SRAs were considered positive for SFB if SFB appeared as the top hit and clustered with the SFB clade in a maximum likelihood phylogenetic tree.

### SFB genome and 16S rRNA gene sequence identification in human metagenomic datasets

FASTQ files of all SRAs from a bioproject, or only those SRAs of a bioproject identified by PebbleScout[43] to be potentially SFB-positive based on 16S rRNA gene sequence analysis, were downloaded using CLC genomics workbench v23.0.5 NCBI database interface. Quality filtering was performed by trimming of sequences with a low-quality score (below 0.05), removing the terminal 7 nucleotides from the 5′ end, and discarding sequences with lengths below 25 nucleotides. The resulting quality-filtered reads from each SRA were mapped against the reference SFB genome sequences from various host origins with a percent nucleotide sequence identity and coverage cutoffs of 95%. Reference genomes available only in multiple contigs were concatemerized into a single contig. Mapped reads were scaffolded into consensus sequences with N ambiguity symbols and aligned against the reference SFB genomes to calculate the average nucleotide identity and coverage. To visualize mapping, heatmaps were generated to represent the quality scores across the genomic positions. The quality score (QS) of each base in the consensus sequence is a measure of the cumulative sequence read quality, which reflects the confidence in the nucleotide correctness at each base position, for all the reads that cover this particular base.

Data corresponding to positional and quality scores from mapping to different reference SFB genomes generated after mapping in CLC genomics were processed using R (v4.0.2) with the ggplot2 package for plotting and the patchwork package for combining multiple plots. Individual plots were created for each dataset and include the reference genome and coverage statistics.

To obtain a 16S rRNA gene sequence consensus, metagenomic reads were mapped to the full-length 16S rRNA gene sequence of the relevant reference SFB at a 100% sequence length and identity. Only paired-end reads were included in the CLC-based analysis. The read mapping stringency was lowered to 99% nucleotide identity when coverage was uneven, as observed in the samples from Zimbabwe, Niger, Tanzania, and the paleofecal sample. For the extended analysis, single-end reads that map to the extreme 5′ or 3′ of the 16S rRNA gene sequence were included to extend the sequence coverage. In this case, the percent sequence length identity was maintained from the CLC-based analysis for the single-end reads, except for the paleofecal sample, where the % identity was lowered to 98% to obtain full coverage. The consensus sequence was generated by the CLC software and was further verified by BLAST analysis to the NCBI nt database and phylogenetic positioning within the SFB clade.

### Statistical analysis, reproducibility, and data representation

Statistical analysis was performed with GraphPad Prism 9.10 (Graph-Pad Software Inc., USA). Upset plots displaying the number of shared and unique gene clusters among the SFB genomes from varying hosts were generated in R v4.0.2 using the package UpSetR v1.4. Maps showing the SFB presence and distribution across the globe were generated using R with packages rnaturalearth v0.3.4, countrycode v1.5, ggrepel v0.9.3, and tidyverse v2.0. Plot showing overall similarity between the reference genome Mouse-SFB-NL (CP008713) and *Peptoclostridium difficile* genome BN1097 with its contig 93 was plotted using SimiPlot v1.1 (https://gitlab.pasteur.fr/GIPhy/SimiPlot). The two-sided Fisher's exact test for contingency analysis and the two-tailed Mann–Whitney exact test for sample analysis were performed using Prism v10.6.1. Representative images for Fig. 1c and Supplementary Fig. 8, Fig. 1d, f and Supplementary Fig. 9a, Fig. 2g and Supplementary Fig. 9b, Fig. 1h and Supplementary Fig. 10a, Fig. 1i and Supplementary Fig. 10b are from three independent experiments. Representative images for Fig. 1a, b and Supplementary Fig. 10d, Fig. 1c, e and Supplementary Fig. 10c, and Fig. 1d are from two independent experiments.

### Reporting summary

Further information on research design is available in the Nature Portfolio Reporting Summary linked to this article.

## Data availability

The raw sequencing files and the metagenome-assembled genomes (MAGs) for Human-SFB-ML-1 and ML-2 generated in this study have been deposited in the National Center for Biotechnology Information (NCBI) database under bioproject PRJNA1106451 and includes the genome assemblies with accession numbers JBRACM000000000 (Human-SFB-ML-1) and JBRACM000000001 (Human-SFB-ML-2). The 16S rRNA gene sequences are made available at NCBI under the GenBank accession numbers PX000509 (Human-SFB-ML-1), PX000510 (Human-SFB-ML-2), PX000511 (Human-SFB-KE), and PX000512 (Human-SFB-GM). Additional SFB 16S rRNA gene sequences used in this study: Human-SFB-SE [contig141_89555], HumanSkin-SFB-US [JF168221.1], Pig-SFB-JP [AB822980.1], Chicken-SFB-BE [PV993571.1],

Chicken-SFB-UK [X80834.1], Chicken-SFB-CN [DQ342328.1], Chicken-SFB-CN [DQ342328.1], Shorebird-SFB-US [KC478326.1], Mouse-SFB-NL [CP008713.1], Mouse-SFB-JP [AP012202.1], Mouse-SFB-YIT [AP012209.1], Rat-SFB-NL [X87244.1], Rat-SFB-JP [D86302.1], Rat-SFB-YIT [AP012210.1], Macaque-SFB-JP [D86303.1], Gorilla-SFB-US [EU474247.1], Dog-SFB-US [DQ113757.1], Sealion-SFB-US [JQ207968.1], Dolphin-SFB-US [JQ202596.1], Mackerel-SFB-US [JQ191772.1], Pinfish-SFB-US [KJ197471.1], Trout-SFB-FR [AY007720.1] and Nibea-SFB-CN [KX431301.1]. Four *Clostridium* species used as outgroups for phylogenetic analysis: *C. acetobutylicum* [NR_074511], *C. saccharobutylicum* [NR_122051], *C. butyricum* [CP040626], and *C. tetani* [X74770.1]. SFB genome sequences used for the comparative analysis are: Rat-SFB-YIT [AP012210], Mouse-SFB-NL [CP008713], Mouse-SFB-JP [AP012202], Mouse-SFB-YIT [AP012209], Turkey-SFB-US [UMNCA01], and Human-SFB-SE [ERZ1468256]. The Turkey-SFB-US 16S rRNA gene sequence in Supplementary Figs. 3 and 7 was obtained from the assembled genome sequence. Source data are provided for the analysis of new data generated in this study and presented in Figs. 1–6 and Supplementary Figs. 13, 15, 23, 25 and 26. All bioprojects analyzed in this study are listed in Supplementary Data 4 and 5. Source data are provided with this paper.

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

## Acknowledgements

We thank Patrick England and the team at the Plate-Forme de Biophysique Moleculaire at Institut Pasteur for their technical help. We thank Jules Rousseau for his Master internship work. We thank Romain Koszul (R.K.), Eduardo Rocha, Benoit Chassaing, Stephane Descorps Declere, Simonetta Gribaldo and Michael D. Wilson for helpful discussions, Julien Guglielmini for help with the FATA conjugation system identification, and William Daniau for informatics assistance. P.S. also thanks Abeer Hoque, Jehanne Carillon, Nandi Simpson, Nicole Scherg, Pascale Vonaesch, Mirjam Schnupf and Darius Teter for their support. The lab of P.S. is supported by INSERM, CNRS and Université Paris Cité and this work was funded through the Bill and Melinda Gates Grand Challenge grant (OPP1141322), the ANR JCJC grant MICROEDUCATION, the ERC Consolidator Grant NICHEADAPT (86222) and the Bettencourt Foundation Coups d'élan prize to P.S. J.R. was supported by the Chaire Blaise Pascal, awarded by the Foundation of the Ecole Normale Supérieure; N.C.B. was supported by the ERC Advanced Grant IMMUNOBIOTA (339407) and A.R.C and A.D. were supported by PhD fellowships from BioSPC, Université Paris Cité. M.M. and L.B. benefited from funding from the European Research Council under the Horizon 2020 Program (grant agreement 260822) to R.K while A.S. and W.M.dV and the HELMi cohort were supported by the Business Finland Grant 329/31/2015 and the Academy of Finland Grants 1325103 and 325103. The Biomics Platform, C2RT, Institut Pasteur, Paris, France, is supported by France Génomique (ANR-10-INBS-09) and IBISA. Open Access enabled by the ERC CoG NICHEADAPT (866222). Lastly, we would like to thank all the investigators whose studies we analyzed for making their data publicly available.

## Author contributions

Bioinformatic identification of SFB-positive Mali, Kenya, and The Gambia samples by P.Sc., M.B., N.C-B., and J.R. and acquisition from B.T. and S.O.S (Mali), R.O. and J.B.O (Kenya), and J.H. and M.A. (The Gambia) with the help of J.R. and P.J.Sa. Gram stain and FISH analysis by P.Sc. SEM by P.Sc. and M.M.N. Chromosome conformation capture and analysis by L.B., B.M., M.Ma., J.R., and S.K. Bait capture and PacBio sequencing by J.P.d.F. and P.Sc. with data processing by S.K, M.Mo., and T.C. Genome assembly and bioinformatic processing by B.M. and S.K. and analysis by P.Sc. and S.K. MdxE expression, purification, and characterization by A.R.C. Global bioinformatic SFB identification by P.Sc., A.R.C., S.K., and A.D. Bioinformatic processing of 16S and metagenomic datasets by S.K., A.R.C., A.D., and A.L. with help for pipeline design from J.G. and data analysis by S.K. Unpublished metagenomic data for I484_MGI_24m provided by A.S. and W.M.dV. Bioproject metadata analysis by P.Sc. with assistance from A.R.C, S.K., and A.D. Writing of the manuscript by P.Sc. P.Sc. is the lead author.

## Competing interests

The authors declare no competing interests.

## Additional information

[1]Laboratory of Host-Microbiota Interaction, INSERM UMR-S1151, CNRS UMR-S8253, Institut Necker Enfants Malades,  Université Paris Cité, Paris, France. [2]Institute for Genome Sciences and Department of Microbiology and Immunology, University of Maryland School of Medicine, Baltimore, MD, USA. [3]Institut Pasteur, Laboratory of Spatial Regulations of Genomes, Paris, France. [4]Institut Pasteur, Plate-forme Technologique Biomics,  Université Paris Cité, Paris, France. [5]Bioinformatics and Biostatistics Hub, Université Paris Cité, Paris, France. [6]Institut Imagine, Laboratory of Intestinal Immunity, INSERM UMR1163, Paris, France. [7]UtechS Ultrastructural BioImaging UBI, Centre de Recherche et de Ressources Technologies (C2RT), Institut Pasteur, Paris, France. [8]University of Lausanne, Lausanne, Switzerland. [9]Kenya Medical Research Institute (KEMRI)-US Centers for Disease Control and Prevention Research Collaboration, Kisumu, Kenya. [10]Medical Research Council Unit, Serrekunda, The Gambia. [11]Center for Vaccine Development, Bamako, Mali. [12]Human Microbiome Research Program, Faculty of Medicine, University of Helsinki, Helsinki, Finland. [13]Laboratory of Microbiology, Wageningen University, Wageningen, The Netherlands. [14]Collège de France, Paris, France and Institut Pasteur, Unité de Pathogénie Microbienne Moléculaire, INSERM U1202, Paris, France. [15]These authors contributed equally: Shashi Kiran, Ana Raquel Cruz, Pamela Schnupf. ✉e-mail: pamela.schnupf@inserm.fr

