## [Transparent Peer Review file · Nature Communications]

Segmented filamentous bacteria are worldwide human gut commensals

Corresponding Author: Dr Pamela Schnupf

Version 0:

Reviewer comments:

Reviewer #1

(Remarks to the Author)

The authors continually discuss the number of reads in both the figures and text. When referring to 16S datasets this should be changed to relative abundances and normalisation should be conducted (and stated) for the metagenomic data.

The authors have been used all the potential metadata available to them, but the paper is currently far too dense to make sense of. The authors need to condense the text into few points, which can be expanded within a Supplementary Results section.

By creating a supplementary results the authors can condense many of the results into single impactful results. For example, the authors could compress Figure 7 into a single panel detailing the percentage of SFB positive samples at each time point across different countries. This would provide the most critical point of the time-specific colonisation, and remove a lot of the 'fog' which will exhaust the reader. Currently this is the major issue with this work. It is a lot of work and presented in an unclear and exhausting manner which makes a reader want to skip large parts because it becomes very repetitive.

Major edits

- The results from the core genome are confusing. You state that it consists of 782 genes. Yet, you then state the Human-SFB-ML genomes are most similar, sharing 411 genes with each other. This is lower than the core genes. Are you referring to additional accessory genes, of which 411 match between the two strains? If so that must be clarified as currently these two statements contradict each other.

- This again becomes an issue when discussing 'ln 239' where you state " Human-SFB-ML shares uniquely only 10 genes with all". This is not true based on your core gene analysis. If you are referring to the accessory genome, you need to state this. If this is what you mean by 'uniquely' shared genes, this would refer to the idea they are only shared between these two strains, not that they are additional to the core genes.

- The authors refer to '6 main lineages' of SFB in humans, but this is all based on 16S rRNA gene identity. I think this must be clarified throughout the text, that 'based on 16S rRNA gene sequence identity, at least 6 lineages can be identified in humans'.

- In Figure 5 multiple panels report 'QS' but in the text it is referred to as ANI analysis. This is another figure which is very full, but lacks clarity. I am unsure what 'QS' is and expected it to be regions of ANI similarity but have no confirmation that this assumption is true. In the legend you refer to it as Quality Score, but this doesn't provide any more clarity on what determines quality i.e. quality of ANI or alignment?

- Figure 5j = I am quite concerned by the analysis of $n = 1$. This provides no insight, yet regression analysis is being done on this single point. This should be removed as I see no justification for this to be done.

- Throughout the paper the number of samples in each analysis is not provided in the figure. The authors should state this for each analysis conducted, both in the text and in the figures directly so readers can appreciate if the authors have analysed just a single sample or thousands. Without this information it is impossible to understand how prevalent SFB truly is. In many

instances I believe the authors would be better off reporting the percentage of positive samples as well. For example, in Figure 7eiii the number of samples are meaningless without knowing how many samples were analysed and its the proportion which is most interesting.

- The Anvio analysis has a few issues. The 'core' group of proteins arnt 100% present, so cannot be deemed core. Please also add the number of genes under each grouping. The colours are also hard to distinguish. The figure is also very busy. You include additional tracks on homogeneity, but these are not referred to in the text. Please remove or discuss if they are interesting. You have also included ANI analysis again within this figure which is redundant with 'b'. This makes the figure even more busy and adds nothing which isnt already present. Either remove 'b' or the ANI analysis from 'e'.

Minor edits

-In 58 = outsized is an odd phrase to use, maybe consider 'great' or 'substantial'

-In 87 = 'in humans' to 'within humans' to avoid repetition

-In 116 = 'gene sequences were identical and therefore constitute the same SFB species.' this is not true. Many species are unable to be differentiated based on their 16S rRNA gene sequence.

-Figure 1b = 16S rRNA gene identity cannot go over 100 so limit the y axis.

-In 158 = you state the higher SFB reads correlated to higher number of filaments. Firstly, were the 16S reads normalised? It is typical to report relative abundance, so the terming of 'reads' is concerning'. Secondly, please report this correlation if it exists.

- Figure 2b / Figure S9e, limit y xis to 100.

- Figure 2 g = You use different colours here than you have used for the different host species throughout the rest of the figure. This is very confusing without a new legend. Please clarify this figure so it makes sense with the rest of the work.

- Figure 4a = limit y axis to 100

- Do not use '% coverage' in the text, % should be written out as percentage unless it is providing a percentage value.

Reviewer #2

(Remarks to the Author)

In this study, Kiran et al. identify and characterize a novel human-associated segmented filamentous bacteria (SFB) species, "Human-SFB-ML", from fecal samples of African cohorts, and then provide support for 4 major and 2 minor lineages worldwide. The authors provide genomic and morphological data in support of the novelty and relatedness of this species, map its global distribution, and investigate colonization dynamics, demonstrating peak prevalence around weaning. The authors highlight that early-life colonization, as in animal models, could have immunological significance in humans as well. They note that murine SFB are potent inducers of gut immunity, and by analogy suggest that Human-SFB-ML may contribute to mucosal immune development in humans. This work is significant as it confirms that SFB, known immunomodulators in animal models, are present globally in humans. Establishing human-specific SFB lineages opens essential avenues to study their potential role in human mucosal immunity and disease.

A major strength of this study is the breadth of methods used to validate Human-SFB-ML. The authors perform phylogenetic analysis of 16S rRNA sequences and deep metagenomic assembly (16S rRNA analysis and long-read and Meta3C metagenomics), coupled with morphological techniques (Gram staining, SFB-specific FISH, and SEM)—to confirm the filamentous morphology in human fecal samples) in support of the existence of a novel SFB in humans. The focus on metabolic pathway analysis (starch/glycogen utilization) provides functional insights beyond taxonomic classification. The functional studies such as binding capacity of recombinant ABC transporter substrate-binding protein are much appreciated.

The weaknesses of this study include assertions about the specificity and uniqueness of cell morphology without much comparative analysis to other microbial cells with and without alleged SFB status, and limited evidence linking the SFB sequences to the morphotypes.

Specific Comments:

1. History is replete with examples of mistaken or inappropriate classification and taxonomic assignment, especially of microbial life, based on morphology. In short, morphological features are usually non-specific and of polyphyletic origin. The authors make a number of statements about "highly unique" morphological features of SFB, e.g., tip structure. It may well be unusual, but "unique" is a high bar, and frankly, there are very few careful morphological surveys of microbial communities.

Figure 1ij is said to “[confirm] the highly unique and SFB-specific tip structure”, but the basis for this claim is unclear.

It would help to provide a side-by-side comparison of Mali S340 and Mali S195 with analogous images of reference SFBs in order to support claims of morphological similarity, and then images of closely related bacterial tip morphologies or termini of non-SFB filamentous organisms to support claims of distinctness. Regardless, I would refrain from claims of uniqueness without clear caveats and descriptions of the comparison set. And I would avoid assumptions about phylogenetic relatedness based on morphology.

2. Regarding phylogenetic coherence, Fig 1A seems to include only sequences that have been labelled as associated with SFB, based on non-explicit criteria, as well as a discrete Clostridia outgroup. Are there 16S rRNA gene sequences found in databases that are more closely related to some of these “SFB” sequences but not annotated as associated with SFB? If so, they should be added to this analysis.

3. A critical aspect of this story is the establishment of a specific and sensitive association (co-localization) between the typical filamentous (proposed SFB) bacterial morphotypes and a proposed SFB-specific 16S rRNA gene sequence.

a. There are relatively few SFB DNA PCR-negative samples (eight). It would be helpful to have more negative controls, including bacteria with the most closely related 16S rRNA sequences, e.g., Clostridia.

b. It would also be helpful to provide the correlation plot for SFB 16S rRNA gene reads in a sample versus number of visualized filaments (line 158).

c. Fig 1H (FISH) is quite important for the claims made in this study—but it is the only one (?). It would be important to provide additional FISH images, from multiple samples and hosts.

4. Lines 798-806. The human disease-related phenotypes associated with SFB carriage, such as pathogen colonization resistance and exacerbation of auto-immune disease and IBD, are confusing, given the high prevalence of SFB carriage in African populations and the relative rarity of these host phenotypes in Africa. It would be helpful for the authors to address this apparent inconsistency.

5. It would be helpful for the authors to provide explicit clarification of bacterial nomenclature for the novel SFB species described in this study, in line with the International Code of Nomenclature of Prokaryotes (“Candidatus” or SeqCode designation). This would align the findings in this manuscript more clearly with contemporary taxonomic standards.

Minor:

6. The authors should be consistent in treating SFB as plural or single (e.g., see line 53 versus 60-61).

7. Figs 5-7: These Figures are too (unnecessarily) complex, crowded, busy. I would present fewer data in each Figure and focus on data most important for making the major points.

8. Similarly, the Results section in general is too long with too many minor facts and details. In particular, the sections entitled “Identification of the Human-SFB-XX genome in metagenomic datasets” could each be shortened significantly without undermining the key findings.

9. Did Meta3C data processing include the use of an error model so as to identify and filter out likely spurious ligation contacts?

10. Lines 606, 630: What does “potent” mean, with respect to “potent colonization peak”?

Version 1:

Reviewer comments:

Reviewer #1

(Remarks to the Author)

The authors have addressed all my major issues and I think it will be a great publication.

Major edits:

- The inclusion of the formal description of a species is fantastic, however the protologue provided does not follow the standard formatting. A separate protologue is needed for both the genus, and the species description. A nomenclatural type has also not been proposed in the protologue, I believe the genome/MAG created from the mentioned BioSample must be deposited as well as the sample. I also see in the SeqCode system there are a few errors so this must still be resolved.

Minor edits:

- In 182: "IO formation" is mentioned but not explained as to what it means.

Reviewer #2

(Remarks to the Author)

In this study, Kiran et al. identify and characterize a novel human-associated segmented filamentous bacteria (SFB) species, "Human-SFB-ML", from fecal samples of African cohorts, and then provide support for 4 major and 2 minor lineages worldwide. The revised manuscript is an improvement over the original and the responses to reviewers are constructive. Among my original critiques and the associated responses by the authors, a few deserve mention here, along with some additional ones.

1. The discussion and defense of the term, "unique" in referring to the structure of SFB is appreciated. Perhaps my discomfort with the use of this term arises from the fact that we have sampled and examined visually only a tiny fraction of microbial diversity. "Unique" needs to be understood as referring to only this tiny fraction, in which case, it might be more appropriate to use, "unique, so far", or "unique among microbes characterized so far". If the intended meaning is to refer to all microbes on the planet, then "potentially unique" would be more appropriate.

2. Fig S4 and the new phylogenetic analysis of SFB sequences together with the "49 closest bacterial species" is appreciated. However, the authors state (line 129) that these 49 species were identified using the "EzBioCloud" database. This is a commercial/proprietary database and according to the company's website (<https://help.ezbiocloud.net/ezbiocloud-16s-database/>), this database is dated "July 10, 2017". For both these reasons, this is not an appropriate reference/resource on which to conduct the analysis presented in this paper. Similarly, use of a non-aligned sequence database like NCBI may not necessarily provide the "most closely related SSU rRNA sequences. Use of a state-of-the-art, expert-curated rRNA sequence database, like Silva (<https://www.arb-silva.de/>) would be helpful. As an aside, in the title of Fig S4 "trees" should be "tree".

3. Lines 168-171: The authors now state, "Long filaments (normally >30 µm) with the thin and-smooth to thick-and-bulbous filament transition...were more readily identified in fecal samples that, based on 16S rRNA gene amplicon analysis, had a higher SFB relative abundance." It would be helpful to provide the data that support this finding, rather than just stating this finding.

4. The authors state in their response to a critique of FISH, "It is clear from the FISH analysis that our staining is very specific as the positive bacteria are in a dense aggregation of bacteria". This statement is not justified unless they happen to know that the surrounding bacteria have very similar (closely related) 16S rRNA sequences. All we know from this photo is that there are some bacteria that do not stain with the SFB probe. This is not a convincing argument for specificity. To evaluate specificity, the authors should obtain some of the Clostridia shown in Fig S4, or the most closely related bacteria to SFB that are available in culture and test these bacteria under the same conditions as used for the SFB staining.

Version 2:

Reviewer comments:

Reviewer #2

(Remarks to the Author)

The authors have adequately responded to my comments. I especially appreciate the use of a closely related Clostridium species as a negative control for FISH. I would have preferred some explanation or rationale for the use of "unique" in describing SFB morphology, rather than a curt dismissal of my comment, but this is not a major issue.

Point-by-point response to reviewers' comments for the manuscript NCOMMS-25-35880-T entitled "Segmented filamentous bacteria are worldwide human gut commensals"

Text in black – reviewer comments

Text in blue – author response

Text in green – changes in the manuscript

We thank the reviewers for their careful reading of the manuscript and the helpful comments and suggestions that have greatly improved the manuscript. We have addressed all of the reviewer's concerns through the addition of more supportive data and changes to the text and figures. Please find below a detailed description of the modifications in the point-by-point response.

Reviewer #1 (Remarks to the Author):

The authors continually discuss the number of reads in both the figures and text. When referring to 16S datasets this should be changed to relative abundances and normalisation should be conducted (and stated) for the metagenomic data.

At the reviewer's request, we have now changed the nomenclature from SFB read frequency to SFB relative abundance within the text and figures.

We still refer to 16S rRNA gene reads in the manuscript as we want to be transparent about the number of positive hits we obtain per sample. Given the low number of positive reads, we included in the analysis both the frequency (i.e. relative abundance) and the read number per sample.

As the total number of reads (i.e. sequencing depth) can vary from one study to the next, the relative abundance of SFB does not give the reader the information on how close we are to the detection limit. This parameter is important because, as we are often at the limit of detection, it suggests that SFB prevalence may be even greater than what we currently calculate. This point is developed in Fig. 6d and 6e where ultradeep sequencing revealed that SFB prevalence can increase in the adult population by 8-fold just by increasing the sequencing depth from 100,000 to 2 million reads.

At the reviewer's request, we also include normalization of metagenomic data. Specifically, for the Finnish metagenomic bioproject PRJEB51728, for which we also have the 16S rRNA gene amplicon data, we had compare the SFB relative abundance based on 16S rRNA gene amplicon analysis and the percent coverage of the Human-SFB-SE genome sequence, shown in Fig. 5f. We now include, in Fig. S26d, the correlation obtained when the metagenomic read data of all five samples analyzed are adjusted to 10 million reads/sample. The normalization does not have an impact on the conclusions and we therefore prefer to leave the unadjusted data in the main Fig. as it shows the actual percent Human-SFB-SE genome coverage obtained per sample, making it easier to understand and compare to the other data presented.

For the analysis conducted on only the metagenomic data, as we are interested in comparing the total coverage of the SFB genomes or 16S rRNA gene sequences by metagenomic reads, we do not include any normalization.

The authors have been used all the potential metadata available to them, but the paper is currently far too dense to make sense of. The authors need to condense the text into few points, which can be expanded within a Supplementary Results section.

We agree with the reviewer that the manuscript is dense. We have worked to shorten the text of the manuscript, simplified some figures (Fig. 5), and moved data into the supplementary section. We have also divided Fig. 6 and 7 into two separate figures (Fig. 6 and 7, and Fig. 8 and 9, respectively) to make the amount of data in the figures less overwhelming.

Indeed, we struggled for a long time in trying to see how best to present the data and whether breaking it up into two manuscripts: 1. Identification and characterization of a new SFB species; and 2. Analysis of SFB presence worldwide and its colonization dynamics, would be best. However, each time, presentation of the full story, despite its length, seemed more appropriate.

We have noted that previous reviewers of the manuscript have had difficulties understanding some of the analysis due to a lack of sufficient detail. While the detail may make the manuscript quite dense, it does allow us to address the comments and concerns from other readers. We have structured the manuscript so that the end of each paragraph of the results section summarizes the findings, allowing the reader to skip the details if wished.

Importantly, the online format of Nature Communications and its less restrictive word count provides the rare possibility to publish a longer, more in-depth, and detailed manuscript. This makes it a particularly good journal for this manuscript that covers a broad range of topics and allows the inclusion of sufficient detail so that all readers, who invariably focus on different parts of the paper, can fully understand all the analysis presented.

We therefore ask the reviewer for lenience in the length and details provided in the manuscript. We believe the data is best presented together in one manuscript as there is a natural flow from the identification of the new species to its identification in publicly available databases, leading to the finding of the distribution of the various lineages. The manuscript thereby provides a comprehensive understanding of the distribution of SFB lineages in humans, including genomic data to support the 16S rRNA gene amplicon data, and makes it very clear that humans harbor multiple SFB lineages, thereby also making sense of the already published literature and clearing up confusing findings such as, for example, the difficulty in finding Human-SFB-SE in publicly available datasets.

Notably, the search for SFB in publicly available databases has been attempted by a number of investigators in the field but due to the low SFB relative abundance our work is the first to provide an outline of SFB distribution and prevalence worldwide. The large amount of data we were able to obtain on SFB in humans may start to fatigue the reader but, when put in the context of how hard it has been for the field to obtain an understanding of the presence of SFB in humans, it is a major achievement and this manuscript, through its details, allows the reader to understand why the field has struggled to identify SFB. This includes, for example:

1. The difficulty in identifying SFB reads -> due to the only ~1-month long colonization peak and the steady state SFB levels largely being below the detection limit of the regular sequencing depth.
2. The difficulty in finding Human-SFB-SE in publicly available datasets -> because this SFB lineage appears to be a dominant lineage predominantly only in Northern Europe.
3. The difficulty in obtaining convincing data at very low frequency -> most of our data relating to colonization dynamics and life cycle are from the Human-SFB-refML lineage. The identification of Human-SFB-ML, for the first time, allowed the discovery of much of the data and due to its full characterization greatly supports the low frequency data.

By creating a supplementary results the authors can condense many of the results into single impactful results. For example, the authors could compress Fig. 7 into a single panel detailing the percentage of SFB positive samples at each time point across different countries. This would provide the most critical point of the time-specific colonisation, and remove a lot of the 'fog' which will exhaust the reader. Currently this is the major issue with this work. It is a lot of work and presented in an unclear and exhausting manner which makes a reader want to skip large parts because it becomes very repetitive.

We thank the reviewer for their recognition of the large amount of work that went into the study. We understand the reviewer's concerns about the length and the repetitiveness.

We have been cognizant of the large amount of data and data analysis that is part of the manuscript. To only show the most relevant data in the main figures, the manuscript contains 28 supplementary figures and 15 supplementary tables.

We have included a detailed analysis of the most relevant bioprojects in the main figures because we feel that it is, indeed, important to demonstrate the high consistency of the results across different studies and particularly across studies from different geographic regions and with different SFB lineages, as much as this was possible.

For Fig. 7 a-c (now Fig. 8 a-c) for example, (a/b) are with the Human-SFB-refML lineage in (a) Malawi and (b) Peru, while (c) is for the Human-SFB-refSE lineage in Finland.

Notably, we have otherwise placed data from the same Human-SFB lineage in the same country but from a different study in the Supplemental data. For example:

1. Fig. 7a (now Fig. 9a) is a time course of Human-SFB-refML in Malawi and we have placed another similar but smaller study time course study in Malawi in Fig. S28g to avoid unnecessary repetition for data that shows similar results.
2. Fig. 6a shows the higher prevalence of SFB (Human-SFB-refMouse/refChicken) in intestinal mucosal samples as compared to feces in a study from China, while we included a similar finding from a different study in China in Fig. S28a.

We have also placed repetitive findings, even when originating from different countries, in the supplementary data. For example:

1. Fig. 6c shows the detection of SFB (predominantly Human-SFB-refML) in fecal but not stomach and duodenal samples in Central African Republic while similar results were obtained for Madagascar, which is shown in Fig. 28c.

Regarding specifically Fig. 7 (now Fig. 8), within the different panels of Fig. 7a-c (i-vii), we structured the data representation so that the equivalent analysis lines up horizontally for a direct comparison of the data in these three studies across geographic regions and Human-SFB lineages for ease of understanding.

We believe that each panel is important as each panel was included to show a specific result:

- (i) Prevalence based on child age group, highlighting the significant increase in SFB prevalence in around 12 to 18M of age .
- (ii) Overall prevalence in the study, highlighting the SFB prevalence likely present within the country at large.
- (iii) SFB relative abundance per age group, highlighting the age of maximal SFB relative abundance for comparison across countries
- (iv) Overall SFB relative abundance, highlighting the median and max values for comparison across countries.
- (v) Number of SFB reads per sample, highlighting that many samples were at the detection limit and thus prevalence is likely higher than what is currently calculated.
- (vi) Comparison of age of the individuals SFB positive and negative, highlight the median age of SFB detection for comparison across countries.
- (vii) A particularity of each study:
 - a. the number of children testing positive for SFB at more than one time point;
 - b. the SFB prevalence in children as a function of their weaning status; and
 - c. SFB prevalence in children as a function of the birth mode.

While we understand that this is a lot of data analysis, it makes for a thorough analysis and serves to show consistency across studies from different countries and Human-SFB lineages.

For Fig. 7d (now Fig. 9a), we have now contracted the original Fig.7d (i) and (ii) into one figure (now Fig. 9(i)). We previously already had only included the time course analysis for Bangladesh (Fig. 9a (v)), while placing those of India, Peru and South Africa in a supplementary figure (now Fig. S28) to avoid repetition. Within Fig. 9a, each panel similarly is included to convey an important point:

- (i) SFB status as a function of age, to highlight the massive data that is part of this seminal study and now includes median age of SFB-positive children with statistical analysis, highlighting the median age of SFB detection and the significant deviation for Bangladesh.
- (ii) SFB relative abundance per country, highlighting that for children with more than one positive sample (as sampling occurred monthly), the maximum SFB relative abundance of a child is, at around 0.01 relative abundance, consistent across countries.
- (iii) Number of SFB reads per sample, highlighting that many samples were at the detection limit for the low frequency samples.
- (iv) Overall SFB prevalence in children in each country, highlighting that even in a country like India where only 2 children, with one sample each, were positive, the overall relative abundance is high (14%).
- (v) SFB relative abundance compared to age for children in Bangladesh, to give an example and highlight that for individuals that were positive more than once, these samples tightly clustered together in time. This, together with the data we put in the supplementary figure,

allows us to conclude that the SFB colonization peak is only approximately one month long, a quite striking result thanks to the extensive data available in this seminal study.

We have been able to extract a large amount of information from these rigorously executed and influential papers and we provide a comprehensive analysis. We believe it is important to develop each major point and to include some repetition as much of the data consists of very low values (both in relative abundance and read number). The consistency in the data across multiple studies therefore significantly strengthens the work and we feel that if we were to move the data to the supplementary data files, the absence of the data in the main figures may negatively impact the ability of the reader to understand and trust the underlying data from which we draw our conclusions.

Major edits

- The results from the core genome are confusing. You state that it consists of 782 genes. Yet, you then state the Human-SFB-ML genomes are most similar, sharing 411 genes with each other. This is lower than the core genes. Are you referring to additional accessory genes, of which 411 match between the two strains? If so that must be clarified as currently these two statements contradict each other.

Indeed, we refer to similarities in SFB genomes outside of the 782 core genome genes. In the Upset plot of Fig. 2c, the first category is the core genome and the second column shows that 411 genes are shared exclusively between the two Human-SFB-ML genomes. We have now changed the text to increase clarity:

262 Outside the core genome, the Human-SFB-ML genomes are most similar to each other, exclusively sharing 411 gene clusters with each other when all SFB genomes are considered (Fig. 2c), or 607 gene clusters when compared only to the Human-SFB-SE genome (Fig. 2d).

- This again becomes an issue when discussing 'In 239' where you state " Human-SFB-ML shares uniquely only 10 genes with all". This is not true based on your core gene analysis. If you are referring to the accessory genome, you need to state this. If this is what you mean by 'uniquely' shared genes, this would refer to the idea they are only shared between these two strains, not that they are additional to the core genes.

The reviewer is correct, the 10 genes are only shared between Human-SFB-ML and all rodent SFB. We have changed the text to increase clarity:

277 Within the accessory genes, Human-SFB-ML shares a number of genes with only a subset of the other SFB species; for example Human-SFB-ML shares uniquely only 10 genes with all rodent SFB, 30 genes with Turkey-SFB-US, 38 genes with both Human-SFB-SE and Turkey-SFB-US, and 22 genes only with Human-SFB-SE (Fig. 2c).

- The authors refer to '6 main lineages' of SFB in humans, but this is all based on 16S rRNA gene identity. I think this must be clarified throughout the text, that 'based on 16S rRNA gene sequence identity, at least 6 lineages can be identified in humans'.

To ensure clarity on this point, we have made changes to the results section and the discussion. For example, we added it in this sentence in the section on "Humans harbor four main SFB lineages in their gut microbiota":

389 Together, these findings support the existence, based on the V3-V4 16S rRNA gene region, of at least six human SFB lineages with the main lineages being those with the highest similarity to Human-SFB-ML, Human-SFB-SE, mouse SFB and chicken SFB.

And in the Discussion:

701 Our 16S rRNA gene amplicon analysis indicates the existence of at least six SFB lineages and thereby a striking diversity in SFB in the human population.

In addition, to reiterate the point that, in the future, lineage designations based on full length 16S rRNA gene sequences may be different, given that the V1-V2 16S rRNA gene regions are highly variable, we now include the following sentence:

856 Notably, the lineage designation may also change and expand when full length 16S rRNA gene sequences become available, given the high sequence variability of the SFB 16S rRNA gene V1-V2 region seen within the reference SFB 16S rRNA gene sequences and the variable placement of the

HumanSkin-SFB-US 16S rRNA gene sequence on the phylogenetic tree when considering only the V3-V4 region or the full length sequence (Fig. 1, Fig. S5a/b, Table S1).

- In Fig. 5 multiple panels report 'QS' but in the text it is referred to as ANI analysis. This is another Fig. which is very full, but lacks clarity. I am unsure what 'QS' is and expected it to be regions of ANI similarity but have no confirmation that this assumption is true. In the legend you refer to it as Quality Score, but this doesn't provide any more clarity on what determines quality i.e. quality of ANI or alignment?

We have now indicated in the Material and Methods that:

The quality score (QS) of each base in the consensus sequence is a measure of the cumulative sequence read quality, which reflects the confidence in the nucleotide correctness at each base position, for all the reads that cover this particular base.

- Fig. 5j = I am quite concerned by the analysis of $n = 1$. This provides no insight, yet regression analysis is being done on this single point. This should be removed as I see no justification for this to be done.

We apologize for the lack of clarity. In Fig. 5j (now Fig. 5i), we plot, for the human SFB paleofecal sample in Fig. 5i (now Fig. 5h) the percent 16S rRNA gene coverage as compared to the percent genome coverage. We include the regression lines of the similar analysis obtained for the Zimbabwean (Fig. 5e (i), now Fig. 5d) and Finnish human samples (Fig. 5f (i), now Fig. 5g) that had enough samples to form a regression line. This was done in order to demonstrate that the data of the single paleofecal sample falls at the expected position close to these regression lines obtained for other SFB genomes using multiple samples, thereby increasing confidence in the analysis.

We have changed the Fig. legend to increase clarity:

Comparison between the percent coverage of the Turkey-SFB-US and 1491 bp Human-SFB-US paleofecal 16S rRNA gene sequence obtained through metagenomic read mapping. Includes the simple linear regression lines obtained for the equivalent analysis with the Zimbabwe (ZW) and Finland (FI) samples in (d) and (g) for comparison purposes.

- Throughout the paper the number of samples in each analysis is not provided in the figure. The authors should state this for each analysis conducted, both in the text and in the figures directly so readers can appreciate if the authors have analysed just a single sample or thousands. Without this information it is impossible to understand how prevalent SFB truly is. In many instances I believe the authors would be better off reporting the percentage of positive samples as well. For example, in Fig. 7eiii the number of samples are meaningless without knowing how many samples were analysed and its the proportion which is most interesting.

We thank the reviewer for pointing out the labeling of Fig. 7e (iii) (now Fig. 9b (iii)). This figure is based on the families with twin groups analyzed in (ii) and thus we have changed the axis title from "Number of samples" to "Number of families". Notably, for Fig. 7e (now Fig. 9b), we have changed the analysis to include only twins, rather than all children associated with a mother, as we cannot be sure the other children (i.e. siblings) within a family are from the same mother of the twins. We have changed the text accordingly. The conclusions have not changed.

We have tried to make the data as transparent as possible. We include in each figure the prevalence calculation in our analysis which includes the number of SFB positive and negative samples analyzed and the percent of positive samples. Most projects had only one sample per individual, making SFB prevalence of samples and individuals the same. However, when multiple samples were present, as in the time course analysis, we provide additional analysis to show SFB status at the different time points (for example Fig. 7 (now 8) a-c(i)) including the % of samples for each time point that are positive.

Our analysis also includes a detailed description of each project in the xlm file of Table S10 with the total number of samples (SRA) analyzed, the number of SFB+ samples and the SFB prevalence, i.e. percent positive individuals, when possible. Particularities, including multiple samples per individuals, are listed in a separate column.

- The Anvio analysis has a few issues. The 'core' group of proteins aren't 100% present, so cannot be deemed core. Please also add the number of genes under each grouping. The colours are also hard to distinguish. The Fig. is also very busy. You include additional tracks on homogeneity, but these are not

referred to in the text. Please remove or discuss if they are interesting. You have also included ANI analysis again within this Fig. which is redundant with 'b'. This makes the Fig. even more busy and adds nothing which isn't already present. Either remove 'b' or the ANI analysis from 'e'.

We thank the reviewer for their suggestions to improve the figure. We have made several modifications to the Anvi'o analysis at the reviewer's request.

1. We have replaced the ANI analysis with a number of genome features: Completeness, Number of CDS, Total length, Number of contigs, GC content, and Singletons. The exact values are also available now in Table S2.
2. We have fine-tuned the Core gene clusters; as only the mouse and rat genomes are complete, we have included in the core gene clusters those that are found in these and all others, with the exception of Human-SFB-SE as this is only a draft genome sequence with an 85.91% completeness score by CheckM (version 1.1.0) and thus is expected to also contain these gene clusters once a more complete genome sequence becomes available. Notably, we have updated the CheckM score based on the latest available version (1.1.0) and now included this data in Table S2.
3. We have simplified the Anvi'o figure to only include "core", "phylo-specific" and "cross-phylo specific", removing "strain-specific" as this distinction could only be made for mouse SFB for which the genomes are closed and for which three genomes are available. We now list the number of gene clusters in all the categories in the main text.
4. We have modified the colors of the groupings to increase contrast between them and make them easier to distinguish.
5. We have replaced the geometric and functional homogeneity analysis with a combined homogeneity analysis to decrease how busy the Fig. is and we define this parameter in the main text.

Minor edits

-ln 58 = outsized is an odd phrase to use, maybe consider 'great' or 'substantial'

We have replaced outsized with "substantial".

-ln 87 = 'in humans' to 'within humans' to avoid repetition

As we would like to keep the wording consistent, we have changed the wording in the previous sentence from "in human" to "in human fecal or intestinal samples" to avoid repetition.

-ln116 = 'gene sequences were identical and therefore constitute the same SFB species.' this is not true.

While SFB is not known to have any extra chromosomal DNA, and our chromosome conformation capture experiments support this for the Human-SFB-ML species, we have deleted this sentence from the text.

-Fig. 1b = 16S rRNA gene identity cannot go over 100 so limit the y axis.

We have changed the figure at the reviewer's request, as well as other figures that showed percentage above 100%..

-ln 158 = you state the higher SFB reads correlated to higher number of filaments. Firstly, were the 16S reads normalised? It is typical to report relative abundance, so the terming of 'reads' is concerning'. Secondly, please report this correlation if it exists.

We thank the reviewer for pointing this out; we should have written "higher SFB relative abundance" instead of "SFB reads" in this instance although SFB reads is also correct but it does not provide the additional context.

Obtaining a true correlation proved difficult as bacteria could cluster together and could not be properly counted. Instead of referring to a correlation, we now state that it was easier to identify SFB when the SFB relative abundance was also higher.

We have changed the text to:

168 Long filaments (normally >30 µm) with the thin-and-smooth to thick-and-bulbous filament transition, a characteristic for the differentiation of filament segments into unicellular intracellular offsprings¹⁻³, were more readily identified in fecal samples that, based on 16S rRNA gene amplicon analysis, had a higher SFB relative abundance. SFB-like filaments were never observed in the eight human fecal samples that were SFB-negative by 16S rRNA gene amplicon analysis.

- Fig. 2b / Fig. S9e, limit y axis to 100.

We have changed the figures at the reviewer's request. We also changed the figure legend on the accompanying AAI graph on Fig. S9 (now Fig. S12d).

- Fig. 2g = You use different colours here than you have used for the different host species throughout the rest of the figure. This is very confusing without a new legend. Please clarify this figure so it makes sense with the rest of the work.

In Fig. 2g we now changed the color coding. Genes present in black are part of the SFB core genome; those in purple are only found in the Human-SFB-ML, Human-SFB-SE and Tukey-SFB-US genomes; and those in yellow-green are only in the Human-SFB-ML and Turkey-SFB-US genomes. For consistency, we changed as well the folate pathway of Fig. S13d.

- Fig. 4a = limit y axis to 100

We have changed the figure at the reviewer's request.

- Do not use '% coverage' in the text, % should be written out as percentage unless it is providing a percentage value.

We now spell out % in the text and figure legends of the manuscript.

Reviewer #2 (Remarks to the Author):

In this study, Kiran et al. identify and characterize a novel human-associated segmented filamentous bacteria (SFB) species, "Human-SFB-ML", from fecal samples of African cohorts, and then provide support for 4 major and 2 minor lineages worldwide. The authors provide genomic and morphological data in support of the novelty and relatedness of this species, map its global distribution, and investigate colonization dynamics, demonstrating peak prevalence around weaning. The authors highlight that early-life colonization, as in animal models, could have immunological significance in humans as well. They note that murine SFB are potent inducers of gut immunity, and by analogy suggest that Human-SFB-ML may contribute to mucosal immune development in humans. This work is significant as it confirms that SFB, known immunomodulators in animal models, are present globally in humans. Establishing human-specific SFB lineages opens essential avenues to study their potential role in human mucosal immunity and disease.

A major strength of this study is the breadth of methods used to validate Human-SFB-ML. The authors perform phylogenetic analysis of 16S rRNA sequences and deep metagenomic assembly (16S rRNA analysis and long-read and Meta3C metagenomics), coupled with morphological techniques (Gram staining, SFB-specific FISH, and SEM)—to confirm the filamentous morphology in human fecal samples) in support of the existence of a novel SFB in humans. The focus on metabolic pathway analysis (starch/glycogen utilization) provides functional insights beyond taxonomic classification. The functional studies such as binding capacity of recombinant ABC transporter substrate-binding protein are much appreciated.

We thank the reviewer for their appreciation of the work.

The weaknesses of this study include assertions about the specificity and uniqueness of cell morphology without much comparative analysis to other microbial cells with and without alleged SFB status, and limited evidence linking the SFB sequences to the morphotypes. We have addressed the reviewer's concerns in detail in the following sections.

Specific Comments:

1. History is replete with examples of mistaken or inappropriate classification and taxonomic assignment, especially of microbial life, based on morphology. In short, morphological features are usually non-specific and of polyphyletic origin. The authors make a number of statements about “highly unique” morphological features of SFB, e.g., tip structure. It may well be unusual, but “unique” is a high bar, and frankly, there are very few careful morphological surveys of microbial communities. Fig. 1ij is said to “[confirm] the highly unique and SFB-specific tip structure”, but the basis for this claim is unclear.

We agree with the reviewer that many mistakes have been made in the literature. At the reviewer’s request, we have eliminated the word “unique” in the manuscript when referring to the tip structure.

We also agree that filamentation and spore formation within filaments is not unique to SFB. Particularly, spore-formation in filamentous bacteria has been described in termites for example ⁴. However, the tip structure that is present in SFB from mice and other vertebrate hosts is, for now, and to our knowledge, only found in bacteria of the SFB clade, that is, bacteria that adhere to the ileal epithelial surface and form filaments that show the other defining characteristic of a smooth to bulbous transition.

SFB has always been identified by its unique morphology and intestinal attachment. For the reviewer, we provide here an overview of SFB in animal species showing a similar morphology and host attachment (by the tip structure, when available).

We now include references to the literature describing SFB attachment via the pointy tip:

198 Scanning electron microscopy (SEM) analysis validated the qualitative difference in SFB filaments of the two Mali samples, the characteristically segmented morphology of the filaments (Fig. 1l-n), and the characteristic SFB tip structure that in SFB from other vertebrate hosts is involved in host attachment (Fig. 1l/m) ^{2,3,5-7}.

In addition, the smooth to bulbous transition has, specifically, been highlighted in a number of papers, please see below (non-exhaustive).

Through morphological examination, SFB has been identified in quite a wide range of animals.

On the other hand, the phylogenetic positioning for 16S rRNA gene sequences obtained from the rat, turkey, chicken, palm ruff, trout, vervet monkey, and pig is consistent with these bacteria being in the SFB clade. However, the link between the phenotype and the 16S rRNA gene sequence is not always made (palm ruff, pig), making our comprehensive analysis using multiple techniques particularly valuable.

No full length 16S rRNA gene sequence is yet available for SFB in cows, rabbits, vervet monkey and horses but our own analysis has identified SFB in these animals based on the V3-V4 16S rRNA gene region homology (unpublished).

SFB remains understudied but all the data that is currently available in the literature is consistent with the tip structure, as well as the smooth to bulbous transition, being present in all SFB of the SFB clade and for the 16S rRNA gene sequences available for the SFB clade to correspond to epithelial-adhering bacteria with a smooth to bulbous transition.

It would help to provide a side-by-side comparison of Mali S340 and Mali S195 with analogous images of reference SFBs in order to support claims of morphological similarity, and then images of closely related bacterial tip morphologies or termini of non-SFB filamentous organisms to support claims of distinctness. Regardless, I would refrain from claims of uniqueness without clear caveats and descriptions of the comparison set. And I would avoid assumptions about phylogenetic relatedness based on morphology.

For ease of comparison, we now provide an example of Mouse-SFB-NL filament morphology including the smooth and thin morphology at the tip, the bulbous morphology at the filament end, as well as the generation of spores with a round morphology (Fig. 1c), contrasting with the more rectangular spore shape of Human-SFB-ML (ex. Fig. 1d (iv), Fig. 1f, Fig. 1h). We furthermore provide additional Gram stain images of Mouse-SFB-NL in a new supplementary figure (Fig. S8). All images were obtained from fecal samples of mice monocolonized with Mouse-SFB-NL and thus show the heterogeneity and breadth of morphologies associated with SFB.

In addition, to address the reviewer's request for a morphological comparison to a related bacterium without the tip structure or the thin to bulbous morphological transition, we give here the example of *Clostridium saudiense* and *filamentum*, which are filamentous and host-associated *Clostridium* bacteria without the characteristic SFB tip structure and which fall outside of the SFB clade, from our recent collaborative work of Makki *et al.* 2025 BioRxiv (10.1101/2025.02.28.640726).

Figure 1: Morphological characterization of *C. saudiense* JCC and *C. filamentum* ETTB1, ETTB2 and ETTB3 A. Gram staining of overnight cultures grown in LYBHI broth. B. Scanning electron microscopy showing the morphology of *C. saudiense* JCC and *C. filamentum* isolates. For acquiring SEM pictures, electron high tension (EHT) of 2.00 KV and magnifications of 7.00 KX (upper panel B) and 25.00 KX (bottom panel B) were used. Scale bars represent 5 µm length for panel A and 2 µm length for panel B.

Figure 2: Phylogenetic and genomic analyses for *C. filamentum* ETTB1, ETTB2 and ETTB3 and comparisons with *C. saudiense* JCC A. Phylogenetic tree based on genome sequences showing the relationship of the *C. filamentum* ETTB1, ETTB2 and ETTB3 with other *Clostridium* species.

SFB has always been identified by its morphology and the phylogenetic positioning has been confirmed in all SFB for which the work has been done to connect the morphology and phylogeny. The limitation is that the data available for SFB is still scarce. Our analysis is arguably the most extensive and comprehensive for any SFB species. Our analysis is highly consistent with the literature, however, linking the SFB-characteristic morphology with the SFB 16S rRNA gene sequence.

The tip structure is a defining morphological feature of SFB; indeed, we consider the tip structure to be the most defining feature of SFB as it is present in both unicellular SFB and filamentous SFB. The tip is used for host cell adhesion and, to our knowledge, no other bacteria, either host associated or not, have thus far been identified in vertebrates or invertebrates other than what we consider SFB. This makes our clear demonstration of the presence of the tip structure in Human-SFB-ML particularly exciting and convincing.

The Mali S340 and S195 are morphologically particularly different to each other as the S195 sample, originating from a child with diarrhea, appears to be in a degraded state (Fig. 1i). This is one reason we also included the analysis of the Kenya and The Gambia fecal samples (Fig. 1d/f/g) as these show morphologies of filaments similar to what is seen for S340 (Fig. 1h).

Notably, we previously characterized the morphology of SFB in mice in detail as we succeeded to grow SFB in an *in vitro* SFB-host cell co-culturing system (Schnupf *et al.* 2015 Nature)¹. This work confirmed the life cycle progression of SFB previously deduced from electron microscopy analysis of intestinal sections in the 1970s. We provide here an overview for the reviewer to show the similarities between Mouse-SFB-NL and Human-SFB-ML (addition of zoom images and annotation for Fig. 3c, reproduction of the schematic of the SFB life cycle included in the manuscript Schnupf *et al.* 2020¹ as Fig. 2a).

Schnupf *et al.* 2015 Nature – modified Figure 3c with Figure 2a

Our manuscript also included SEM images of the Mouse-SFB-NL morphology, which we reproduce here with annotation for the reviewer.

All morphological examinations of bacterial species to date in the literature, aside from SFB, are, to our knowledge, negative for the tip structure and the tip structure thus remains a unique structure of SFB.

2. Regarding phylogenetic coherence, Fig 1A seems to include only sequences that have been labelled as associated with SFB, based on non-explicit criteria, as well as a discrete Clostridia outgroup. Are there 16S rRNA gene sequences found in databases that are more closely related to some of these “SFB” sequences but not annotated as associated with SFB? If so, they should be added to this analysis.

The reviewer is correct, Fig. 15a shows the four outgroup bacteria (all Clostridia) and 16S rRNA gene reference sequences available on NCBI that all fall within the SFB clade.

The outgroup species were chosen to represent clades of the most closely related bacterial species to the SFB clade.

We now include in a new supplementary figure (Fig. S4) the SFB 16S rRNA gene reference sequences in a phylogenetic tree with the next 49 closest bacterial species (as well as an additional more distant outgroup species). We include only four of the *Clostridium* species in the main text due to space limitations.

We also include in the manuscript the following text:

126 As expected, the African human SFB 16S rRNA gene sequences fell within the SFB clade when using as an outgroup four *Clostridium* species representing different *Clostridium* clades (Fig. 1a) or the 49 closest bacterial species to the SFB clade in the EzBioCloud 16S database (Fig. S4).

The 16S rRNA gene reference sequences were chosen based on 2 criteria: 1. the length of the 16S rRNA gene sequence and 2. the host species. For sequences that fell within the same group (i.e. mouse, rat, chicken), the gene sequences that are largely full length were chosen. For other hosts (pig, pinfish, shorebird, sealion and mackerel), a representative 16S rRNA gene sequence was chosen because although multiple sequences were present in the NCBI nucleotide database, they originated from the same animal. Only 16S rRNA gene sequences that contained the V1-V4 region were considered.

The additional sequences, which we had not included in our tree, are derived from mouse, chicken, or rat and fall within the refMouse, refChicken, and refRat lineages close to the reference SFB sequences we included. The reference sequences for refMouse/Chicken/Rat were chosen as they were available at full length.

We now include in a new supplementary figure (Fig. S2) the phylogenetic tree of all the 16S rRNA gene sequences, covering at least the 16S V1-V4 region, present in the NCBI nucleotide database, that share at least a 92% nucleotide identity to the V1-V4 16S rRNA sequences of the Mouse-SFB-NL 16S rRNA gene sequence, which was used as the query.

In the manuscript, we have now changed the text to make it clearer that these 16S rRNA gene reference sequences fall in the SFB clade and that, based on this criterion, we refer to them as SFB rRNA gene reference sequences:

118 To identify SFB 16S rRNA gene reference genes, 16S rRNA gene sequences that covered at least the V1-V4 region and showed at least a 92% nucleotide identity to the Mouse-SFB-NL 16S rRNA gene were identified in the NCBI nucleotide database using the V1-V4 region of the Mouse-SFB-NL 16S rRNA gene sequence as the query. All 16S rRNA gene sequences identified fell within the SFB clade on a maximum likelihood phylogenetic tree (Fig. S2). The SFB reference sequences retained from various hosts covered the V1-V9 variable regions, unless not available, in which case representative sequences covering the V1-V4 variable regions were retained. These reference sequences of the SFB clade are referred to as “host”-SFB-“geographic origin of host” (Fig. S3).

The inclusion of 16S rRNA gene sequences that fall into this clade but are not formally linked to a bacterium is common in the SFB field (Pamp *et al.* 2012 Genome Research; Prakash *et al.* 2011 Cell Host Microbe, Ericsson *et al.* 2014 Comparative Medicine)⁸⁻¹⁰. This is because when a 16S rRNA gene sequence of the SFB clade has been linked to a specific morphology, the bacterium has the SFB-characteristic morphology. As discussed above, all currently available data supports the conserved morphology of the bacterial species within this clade.

In addition, while the 16S rRNA gene sequence data has been linked to the bacterium’s morphology in only a limited number of species (notably mouse, rat, and chicken), bacteria with the SFB morphology are present in many host species, as outlined in our responses above and in the review by Ericsson *et al.* 2014¹⁰ (rhesus macaques, crab-eating macaque, vervet monkeys, african gorillas, south african claw-footed toads, carp, rainbow trout, wood mice, guinea pigs, rabbits, horses, cattle, pigs, cats, turkeys, jackdaws, and magpies). The identification of V1-V4 16S rRNA gene sequences within the SFB clade in numerous animals is therefore consistent with the presence of SFB in a wider range of animals based on morphological examination.

Of note, for a direct comparison between fully equivalent 16S rRNA gene sequence regions of our SFB 16S rRNA gene reference sequences, we now include the percent 16S rRNA gene sequence nucleotide identity when considering only the V1-V4 region in Table S1, as our reference 16S rRNA gene sequences all have the V1-V4 16S rRNA gene sequence region, but not all have the V5-V9 region.

3. A critical aspect of this story is the establishment of a specific and sensitive association (co-localization) between the typical filamentous (proposed SFB) bacterial morphotypes and a proposed SFB-specific 16S rRNA gene sequence.

We agree with the reviewer that this a particularly important point.

a. There are relatively few SFB DNA PCR-negative samples (eight). It would be helpful to have more negative controls, including bacteria with the most closely related 16S rRNA sequences, e.g., *Clostridia*.

The fecal samples we have analyzed are complex microbiotas that all contain *Clostridia* species. We have now looked at 14 more SFB-negative human fecal samples from another study and the SFB-characteristic morphology, particular phenotype of the smooth to bulbous transition, were not present.

b. It would also be helpful to provide the correlation plot for SFB 16S rRNA gene reads in a sample versus number of visualized filaments (line 158).

Obtaining a true correlation proved difficult as bacteria could cluster together and could not be properly counted. Instead of referring to a correlation, we now state that it was easier to identify SFB when the SFB relative abundance was also higher.

We have changed the text to:

168 Long filaments (normally >30 μm) with the thin-and-smooth to thick-and-bulbous filament transition, a characteristic for the differentiation of filament segments into unicellular intracellular offsprings¹⁻³, were more readily identified in fecal samples that, based on 16S rRNA gene amplicon analysis, had a higher SFB relative abundance. SFB-like filaments were never observed in the eight human fecal samples that were SFB-negative by 16S rRNA gene amplicon analysis.

While we understand the reviewer's concerns, the SFB phenotype in its differentiated state is very distinctive and only fecal samples with a high SFB relative abundance include filaments that are consistent with the SFB phenotype despite this being somewhat difficult to quantitate.

Notably, the Mali S195 sample had 5.6% SFB relative abundance and had a high proportion of filaments. We now include additional gram stains (Fig. S10b (iii-v)) to better show that the filaments, which we confirmed by SEM to have a pointed tip characteristic of SFB, were one of the dominant bacteria in the sample. In addition, we highlight the presence of apparent spores in some of the filaments (Fig. S10b (iii/v)). However, the Human-SFB-ML-2 genome was still challenging to obtain due to high host contamination of this diarrheal fecal sample.

c. Fig 1H (FISH) is quite important for the claims made in this study—but it is the only one (?). It would be important to provide additional FISH images, from multiple samples and hosts.

We have now provided an additional FISH image (Fig. 1k) of sample S340 that shows staining of a filament, a likely filament remnant (SEM of a likely SFB filament remnant now also provided in Fig. S10d), and, importantly, it also includes staining of two small bacteria showing an irregular shape reminiscent of teardrop-shaped SFB IOs.

It is clear from the FISH analysis that our staining is very specific as the positive bacteria are in a dense aggregation of bacteria (Fig. 1j, Fig.1k (i), Fig.S10c). In addition, those that are labeled support the claim that these bacteria have an SFB-like morphology. The bright field image in Fig. 1j (i) shows that the SFB probe-labeled filament has, as expected, the characteristic bulbous morphology. Similarly, the SFB-specific FISH staining of the new image included as Fig. 1k points towards a difference in filament thickness along the filament and for unicellular bacteria to have the characteristic IO-shape.

As we obtained the fecal samples from a study conducted over 10 years ago, we were only able to get a small amount, precluding further analysis of other samples.

However, our fluorescent *in situ* hybridization data is also consistent with the literature for SFB. For example, for SFB in chickens (Liao *et al.* 2012 FEMS Microbiology Ecology)¹¹, where the investigators, using the same SFB-specific probe, found, as we also do (Fig. 1k (ii)), both teardrop-shaped unicellular

bacteria and unevenly labeled filaments with the FISH staining. We highlight their finding below with a more in-depth analysis of the published figure; we add **zooms**, **annotation** and **green arrows** to highlight the apparent tip morphology of the likely unicellular SFB and point out the uneven FISH staining revealing thin and thick sections of the filament.

Our data is therefore consistent with the literature of fluorescent *in situ* hybridization of SFB from other vertebrate hosts in showing an uneven FISH staining for filaments and a pointed phenotype for unicellular bacteria with the same SFB-specific probe.

We now also include measurements of filament thickness for the smooth filament portions that clearly have an SFB-like tip structure in SEM. The thickness ranges from 0.7 to 1.0 μm , identical to what is described for murine SFB^{2,3} (Fig 1), providing further support.

Finally, we have identified SFB in the 16S rRNA gene sequence data in the Chinese study that showed filamentous bacteria attached to the ileal epithelium but had not identified SFB in their 16S rRNA gene amplicon data (Chen *et al.* 2020)¹². While it is only one read, potentially explaining the lack of detection, it is 100% identical to mouse SFB in the V3-V4 region. This is in line with our other data that shows that the Human-SFB-refMouse lineage is the dominant SFB lineage in China. We suspect that the low relative abundance of SFB is likely due to the inclusion of fecal material during the procedure to obtain the ileal biopsy. We discuss this finding in the discussion:

827 Notably, we identified a 550 bp V3-V4 rRNA gene read (SRR4272058.195935.1) with 100% identity to Mouse-SFB-NL in the 16S rRNA gene amplicon data of this study, providing support that these filamentous bacteria are SFB of the Human-SFB-refMouse lineage.

4. Lines 798-806. The human disease-related phenotypes associated with SFB carriage, such as pathogen colonization resistance and exacerbation of auto-immune disease and IBD, are confusing, given the high prevalence of SFB carriage in African populations and the relative rarity of these host phenotypes in Africa. It would be helpful for the authors to address this apparent inconsistency.

We believe that in western cultures, the microbiota composition can be dysbiotic for various potential reasons (for example ultraprocessed food), thereby promoting the potentially negative effects of the immunostimulatory potential of SFB.

In terms of colonization resistance, we believe that infectious agents would be more detrimental in the African population in the absence of SFB.

We now include the following statement:

842 While the immunostimulatory potential of SFB is complex and its immune activation can be modulated by microbiota metabolites¹³, and thus may depend on microbiota composition, the future

analysis of the SFB relative abundance and prevalence in terms of disease association should provide further insights into the potential role of SFB in disease in humans.

5. It would be helpful for the authors to provide explicit clarification of bacterial nomenclature for the novel SFB species described in this study, in line with the International Code of Nomenclature of Prokaryotes (“Candidatus” or SeqCode designation). This would align the findings in this manuscript more clearly with contemporary taxonomic standards.

We thank the reviewer for this suggestion.

We have registered the Human-SFB-ML genomes in SeqCode and propose the name *Anisomitus miae* for the Human-SFB-ML genomes (seqco.de/r:zlg3gn05). As the genomes have been validated to fulfill all the requirements. The genus name *Anisomitus* has been approved as a valid genus name. This should establish *Anisomitus miae* as the nomenclature type strain for the SFB clade after confirmation of the species name and publication of the manuscript.

We include the following description in the manuscript:

239 The Human-SFB-ML genomes were registered as a new genus and species under the Code of Nomenclature of Prokaryotes Described from Sequence Data (SeqCode) ¹⁴ and named *Anisomitus miae* (seqco.de/r:zlg3gn05). While SFB have long been called Candidatus *Arthromitus* in the literature ¹⁵, *Arthromitus* also describes non-SFB bacteria in termites ⁴. The name Candidatus *Dwaynsavagella* has been used more recently ¹⁶ but, in keeping with a more descriptive name, we chose the genus name *Anisomitus* proposed, in 1925, by Pierre-Paul Grassé to describe a SFB-like organism attached to the intestinal epithelium of the domestic duck and to distinguish it from *Arthromitus*-like organisms found in the gut of termites ¹⁷.

Description of *Anisomitus miae* gen. nov. sp. nov.

Anisomitus miae (*A.ni.so.mi'tus*. **Gr. masc. adj.** *anisos*, uneven; **Gr. masc. n.** *mitos*, thread; **N.L. masc. n.** *Anisomitus*, referring to uneven filamentous form; *mi'ae*. **L. gen. fem. n.** *miae*, of Mia, daughter). This bacterial species is Gram variable and spore-forming, hybridizes with the 16S rRNA-targeted oligonucleotide probe 5'-GGG TAC TTA TTG CGT TTG CGA CGG CAC-3' ¹⁸, and has a 16S rRNA gene sequence that clusters within the monophyletic group in the Clostridiaceae that includes SFB from hosts such as the mouse, rat, chicken, and human. The genome is ~1.6 to 1.7 Mb in size with a GC content of ~30.6%. This species includes all bacteria with genomes that show ≥95% average nucleotide identity to the type genome of the MAG ID Human-SFB-ML-1 from the biosample SAMN41134099 available at NCBI.

The description by Pierre-Paul Grassé:

(<https://www.parasite-journal.org/articles/parasite/pdf/1925/04/parasite1925034p343.pdf>) is to our, and the author's, knowledge the first description of SFB and came almost 50 years prior to the first description of SFB in rodents by Dwayne Savage, a researcher who was nominated by Thompson *et al.* (2012)⁴ to give his name to the genus of bacteria in the SFB clade (*Savagella/Dwaynesavagella*). We propose that in the future an SFB species found in rodents should be dedicated to Dwayne Savage.

Minor:

6. The authors should be consistent in treating SFB as plural or single (e.g., see line 53 versus 60-61).

We have gone through the manuscript with attention to this particular comment. We refer to SFB as plural or single depending on the context. If we only speak about SFB in mice (where there is, for now only one SFB species we are referring to) or the SFB clade, or one specific SFB lineage, we use singular; however, when we refer to SFB lineages or SFB in different host species, we use plural.

7. Figs 5-7: These Figures are too (unnecessarily) complex, crowded, busy. I would present fewer data in each figure and focus on data most important for making the major points.

We understand the reviewer's concern and agree that the manuscript is long and dense.

We have addressed the reviewer's concerns by moving some of the data into the supplementary figures: previous Fig. 5d, the pooled SRA metagenomic read mapping, is now Fig. 23a; previous Fig. 5f (ii), the correlation between the frequency of SFB reads in the metagenomic data and the percent

Human-SFB-ML2 genome coverage, is now Fig. S23f; and previous Fig. 5g (ii), the correlation of the relative abundance based on either WGS read mapping of 16S rRNA gene amplicon analysis, is now Fig. S23e.

In addition, we have split Figures 6 and 7 into two figures (now Fig. 6 and 7; and Fig. 8 and 9, respectively) to make the figures less overwhelming.

We ask the reviewer for lenience in this instance, however. In Fig. 7 (now Fig. 8) for example, each panel is included for a specific reason. As mentioned in the response to reviewer #1, this work is often based on very low numbers and frequencies. For this reason, it is important to be repetitive to show that the results are highly consistent. This can only be done when all the data is presented and by showing the data for different projects, in different countries, for different SFB lineages. We have been able to extract a large amount of data from seminal papers in the field and to do the data analysis justice, we believe it is important to present a comprehensive analysis.

In addition, different readers of the manuscript have also asked us to elaborate on different sections of the manuscript depending on the reader's particular interest or expertise. We therefore chose to elaborate on the different sections while being aware that they include much detail that not all readers may be interested in.

We struggled for a long time in trying to break it up into two papers but while the manuscript is now long, the first part of the paper, i.e. the morphological and genomic identification of a new SFB species in humans, is important to validate the second part of the paper, i.e. the bioinformatic analysis of the SFB presence globally. We therefore found the paper too difficult to separate into two and we ask the reviewer for lenience.

Notably, we think Nature Communications is particularly well suited for the work due to their more relaxed length limits as an online journal.

8. Similarly, the Results section in general is too long with too many minor facts and details. In particular, the sections entitled "Identification of the Human-SFB-XX genome in metagenomic datasets" could each be shortened significantly without undermining the key findings.

We have worked to simplify the text and particularly the section on the Human-SFB-XX genome in metagenomic datasets section. Again, we do believe that repetition is important to show consistency in the data despite often low read number, relative abundance, or genome coverage. In addition, the data for Human-SFB-ML and Human-SFB-SE provides the background for the identification of the Human-SFB-US that is closely related to SFB found in turkeys today. This is a particularly novel finding as well and provides supporting data for different hosts sharing similar, or the same, SFB species.

9. Did Meta3C data processing include the use of an error model so as to identify and filter out likely spurious ligation contacts?

Meta3C data were analysed using the dedicated pipeline MetaTOR (Baudry *et al.*, 2019; Marbouty *et al.*, 2021)^{19,20}. The pipeline uses a normalization procedure based on the HiC coverage of the interacting contigs that take into account different bias identified for this type of data (length of the contigs, abundance of the contigs, and number of restriction sites). The pipeline does not use an error model that identifies and filters out likely spurious ligation contacts as this type of filtering is solely based on a threshold of interactions values and could remove true and informative events. We generally prefer to control the procedure and the correctness of the binning by visualizing and manually inspecting contact map as previously shown (Marbouty *et al.*, 2017; Baudry *et al.*, 2019; Marbouty *et al.*, 2021)¹⁹⁻²¹. The contact map presented in Fig. S8 (now Fig. S11) clearly shows that the data were of good quality and that the MAG corresponding to the SFB genome does not interact strongly with other MAGs from the same sample.

10. Lines 606, 630: What does "potent" mean, with respect to "potent colonization peak"?

With "potent" we mean a strong or vigorous colonization peak. In the manuscript we analyze this colonization peak and show that during this peak, the SFB 16S rRNA read frequencies in feces increases by 100 to 5000 times.

Additional changes to manuscript during revision:

Table S4: addition of BSR scores for SFB genome sequences derived from metagenomic read mapping from Tanzania, Niger, Malawi, Republic of the Congo, Uganda, Kenya and Madagascar.

Table S13: an additional Sub-Total was added for clarity (Row 53)

Fig. S9 (now Fig. S12): removal of (c) whole-genome ANI dendrogram due to repetition as the more rigorous Maximum-Likelihood tree of core genome ANI was already included in Fig. 2a.

Fig. 6g: graphs for PRJNA1081663 included to also demonstrate the presence of Human-SFB-refMouse and refRat across a wide age range.

Corrections of minor mistakes noticed during revision:

Fig. 4b: Homogenization of country labels

Table S11: Line 27: Uganda replaced by Niger and Ethiopia for countries for which nucleotide identity was lowered to 99% - no impact on any analysis or conclusions

Table S13: D52, 1 instead of 104 – a copying mistake - no impact on any analysis or conclusions

Table S13:

D56, 1004 instead of 1106 – a subsequent addition mistake - no impact on any analysis

C53: 42 instead of 43; no impact on any analysis

C57: 44 instead of 45: changes number of SFB-positive countries from 44 to 45 also in text but does not change the world maps showing SFB-positive countries

Fig. S11a: arrow highlighting SFB MAG was adjusted

Fig. S24 b:

Addition of the name of the second SRA used (ERR2367738), and change “9” to “8” month-old infants (previous rounding error)

Fig. 28g (i): inclusion of 18M in statistical analysis

References:

1. Schnupf, P. *et al.* Growth and host interaction of mouse segmented filamentous bacteria in vitro. *Nature* **520**, 99–103 (2015).
2. Chase, D. G. & Erlandsen, S. L. Evidence for a complex life cycle and endospore formation in the attached, filamentous, segmented bacterium from murine ileum. *J Bacteriol* **127**, 572–83 (1976).
3. Ferguson, D. J. P. & Birch-Andersen, A. Electron microscopy of a filamentous, segmented bacterium attached to the small intestine of mice from a laboratory animal colony in Denmark. *Acta Pathol. Microbiol. Scand. Sect. B Microbiol.* **87B**, 247–252 (1979).
4. Thompson, C. L., Vier, R., Mikaelyan, A., Wienemann, T. & Brune, A. “Candidatus Arthromitus” revised: segmented filamentous bacteria in arthropod guts are members of Lachnospiraceae. *Environ Microbiol* **14**, 1454–65 (2012).
5. Smith, T. M. Segmented filamentous bacteria in the bovine small intestine. *J. Comp. Pathol.* **117**, 185–190 (1997).
6. Sanford, S. E. Light and Electron Microscopic Observations of a Segmented Filamentous Bacterium Attached to the Mucosa of the Terminal Ileum of Pigs. *J. Vet. Diagn. Investig.* **3**, 328–333 (1991).
7. Shini, S., Aland, R. C. & Bryden, W. L. Avian intestinal ultrastructure changes provide insight into the pathogenesis of enteric diseases and probiotic mode of action. *Sci Rep* **11**, 167 (2021).
8. Pamp, S. J., Harrington, E. D., Quake, S. R., Relman, D. A. & Blainey, P. C. Single-cell sequencing provides clues about the host interactions of segmented filamentous bacteria (SFB). *Genome Res* **22**, 1107–19 (2012).

9. Prakash, T. *et al.* Complete genome sequences of rat and mouse segmented filamentous bacteria, a potent inducer of th17 cell differentiation. *Cell Host Microbe* **10**, 273–84 (2011).
10. Ericsson, A. C., Hagan, C. E., Davis, D. J. & Franklin, C. L. Segmented filamentous bacteria: commensal microbes with potential effects on research. *Comp. Med.* **64**, 90–8 (2014).
11. Liao, N. *et al.* Colonization and distribution of segmented filamentous bacteria (SFB) in chicken gastrointestinal tract and their relationship with host immunity. *FEMS Microbiol Ecol* **81**, 395–406 (2012).
12. Chen, B. *et al.* Adhesive bacteria in the terminal ileum of children correlates with increasing Th17 cell activation. *Front. Pharmacol.* **11**, 588560 (2020).
13. Nabhani, Z. A. *et al.* A weaning reaction to microbiota is required for resistance to immunopathologies in the adult. *Immunity* **50**, 1276–1288.e5 (2019).
14. Hedlund, B. P. *et al.* SeqCode: a nomenclatural code for prokaryotes described from sequence data. *Nat. Microbiol.* **7**, 1702–1708 (2022).
15. Snel, J. *et al.* Comparison of 16S rRNA sequences of segmented filamentous bacteria isolated from mice, rats, and chickens and proposal of “*Candidatus Arthromitus*.” *Int. J. Syst. Evol. Microbiol.* **45**, 780–782 (1995).
16. Gilroy, R. *et al.* Extensive microbial diversity within the chicken gut microbiome revealed by metagenomics and culture. *PeerJ* **9**, e10941 (2021).
17. Grassé, P.-P. *Anisomitus denisi* n. g., n. sp. schizophyte de l’intestin du canard domestique. *Ann. Parasitol. Hum. Comparée* **3**, 343–348 (1925).
18. Urdaci, M. C., Regnault, B. & Grimont, P. A. D. Identification by in situ hybridization of segmented filamentous bacteria in the intestine of diarrheic rainbow trout (*Oncorhynchus mykiss*). *Res. Microbiol.* **152**, 67–73 (2001).
19. Baudry, L., Foutel-Rodier, T., Thierry, A., Koszul, R. & Marbouty, M. MetaTOR: A Computational Pipeline to Recover High-Quality Metagenomic Bins From Mammalian Gut Proximity-Ligation (meta3C) Libraries. *Front Genet* **10**, 753 (2019).
20. Marbouty, M., Thierry, A., Millot, G. A. & Koszul, R. MetaHiC phage-bacteria infection network reveals active cycling phages of the healthy human gut. *Elife* **10**, (2021).
21. Marbouty, M. & Koszul, R. Generation and Analysis of Chromosomal Contact Maps of Bacteria. *Methods Mol Biol* **1624**, 75–84 (2017).

Point-by-point response to reviewers' comments for the second revision of the manuscript NCOMMS-25-35880A entitled "Segmented filamentous bacteria are worldwide human gut commensals"

Text in black – reviewer comments

Text in orange – author response

Text in green – changes in the manuscript

REVIEWER COMMENTS

Reviewer #1 (Remarks to the Author):

The authors have addressed all my major issues and I think it will be a great publication.

We thank the reviewer for his/her kind comments.

Major edits:

- The inclusion of the formal description of a species is fantastic, however the protologue provided does not follow the standard formatting. A separate protologue is needed for both the genus, and the species description. A nomenclatural type has also not been proposed in the protologue, I believe the genome/MAG created from the mentioned BioSample must be deposited as well as the sample. I also see in the SeqCode system there are a few errors so this must still be resolved.

We thank the reviewer for his/her enthusiasm.

We have addressed the reviewer's concern by adding a separate description for the genus.

Description of *Anisomitus* gen. nov.

A.ni.so.mi'tus. **Gr. masc. adj.** *anisos*, uneven; **Gr. masc. n.** *mitos*, thread; **N.L. masc. n.** *Anisomitus*, referring to uneven filamentous form.

Genus of bacteria commonly referred to as Segmented filamentous bacteria (SFB); known also as *Candidatus Arthromitus* (Greek for "jointed thread"), as proposed by Snel *et al.* (1995). As "*Arthromitus*" also describes non-SFB bacteria⁶⁰, the genus name *Anisomitus* was chosen in keeping with a morphologically descriptive name for SFB and in recognition of the work by the French zoologist Pierre-Paul Grassé who described SFB attached to the intestinal epithelium of the domestic duck in 1925⁶². SFB are Gram variable and spore-forming bacteria that, based on 16S rRNA gene sequence analysis, form a monophyletic group within the *Clostridiaceae*. SFB grow from unicellular bacteria of approximately 1 micrometer in length into filaments reaching over 80 micrometer in length. Particular characteristics include a tip-like structure, present on both the unicellular and filamentous forms, as well as a biphasic filament morphology of thin and smooth to thick and bulbous as the filament ages and unicellular bacteria develop inside the filament to either form spores or to be released from the filament as intracellular offsprings. The genus is represented by closed genome sequences of SFB from mice^{29,63,64} and rat⁶³ as well as metagenome-assembled genomes from the gut of humans³³ and birds^{32,61,65}. Genomes are reduced and range in size from approximately 1.5 to 1.7 Mb, lack nearly all components of the TCA cycle, but include genes involved in flagella synthesis and chemotaxis. The nomenclatural type species for this genus is *Anisomitus miae*.

Seqcode has now endorsed the register list seqco.de/r:zlg3gn05 of the *Anisomitus* genus and the *Anisomitus miae* species as the nomenclature type. The genome sequence and accompanying raw sequence reads have been deposited at NCBI. The names will be formalized upon publication of the current work.

Minor edits:

- In 182: "IO formation" is mentioned but not explained as to what it means.

IO formation refers to the observation that IOs are formed within the mother cell prior to spore formation. The deposition of a spore coat surrounding IOs is documented for both mouse and rat SFB (10, 11). We have added these references now also at the end of the sentence for clarity.

Reviewer #2 (Remarks to the Author):

In this study, Kiran et al. identify and characterize a novel human-associated segmented filamentous bacteria (SFB) species, "Human-SFB-ML", from fecal samples of African cohorts, and then provide support for 4 major and 2 minor lineages worldwide. The revised manuscript is an improvement over the original and the responses to reviewers are constructive. Among my original critiques and the associated responses by the authors, a few deserve mention here, along with some additional ones.

We thank the reviewer for his kind comments and his/her help in improving the manuscript.

1. The discussion and defense of the term, "unique" in referring to the structure of SFB is appreciated. Perhaps my discomfort with the use of this term arises from the fact that we have sampled and examined visually only a tiny fraction of microbial diversity. "Unique" needs to be understood as referring to only this tiny fraction, in which case, it might be more appropriate to use, "unique, so far", or "unique among microbes characterized so far". If the intended meaning is to refer to all microbes on the planet, then "potentially unique" would be more appropriate.

We understand the reviewer's position and thank the reviewer for his/her lenience. We nevertheless did not make any further changes to the manuscript.

2. Fig S4 and the new phylogenetic analysis of SFB sequences together with the "49 closest bacterial species" is appreciated. However, the authors state (line 129) that these 49 species were identified using the "EzBioCloud" database. This is a commercial/proprietary database and according to the company's website (<https://help.ezbiocloud.net/ezbiocloud-16s-database/>), this database is dated "July 10, 2017". For both these reasons, this is not an appropriate reference/resource on which to conduct the analysis presented in this paper. Similarly, use of a non-aligned sequence database like NCBI may not necessarily provide the "most closely related SSU rRNA sequences. Use of a state-of-the-art, expert-curated rRNA sequence database, like Silva (<https://www.arb-silva.de/>) would be helpful. As an aside, in the title of Fig S4 "trees" should be "tree".

We have now included an analysis using the Silva database and a maximum likelihood phylogenetic tree (Fig. S4b) showing similar results to the EzBioCloud database analysis-derived phylogenetic tree (Fig. S4a). The Silva-based ML tree is composed of *Clostridium* species and a bacterium annotated as *Desnusiella massiliensis*, which however has not been validly published based on the DSMZ database and therefore may be revised in the future. It also includes less bacterial species than the EzBioCloud-based tree because the Silva database allowed only a limited recovery of nearest neighbour 16S rRNA gene sequences. Identical 16S rRNA gene sequences of the nearest neighbors were collapsed to a single entry.

We additionally thank the reviewer for the detailed reading of the figure legend. As two ML trees are now included, we kept the original phrasing of the title.

3. Lines 168-171: The authors now state, "Long filaments (normally >30 µm) with the thin and-

smooth to thick-and-bulbous filament transition...were more readily identified in fecal samples that, based on 16S rRNA gene amplicon analysis, had a higher SFB relative abundance.” It would be helpful to provide the data that support this finding, rather than just stating this finding.

As mentioned in the first revision, it is not possible to provide an accurate quantitation as samples were heterogenous (including both non diarrheal and diarrheal samples), sampling for the gram stain was not standardized, and visual quantitation of bacterial numbers remains too problematic. We nevertheless provide an estimation to support the mentioned statement through the comparison of the Human-SFB-ML frequency from at least 60 fields of view at a 40x magnification of the gram stains as compared to the relative abundance of Human-SFB-ML in the V1-V2 16S rRNA gene amplicon dataset of the published manuscript. Fields of view were scored as positive when they contained one or more filaments consistent with the SFB morphology. Notably, the diarrheal Human-SFB-KE sample was particularly mucoid and problematic to score as much of the sample did not spread but rather clustered in a mucoid aggregate (Fig. S9b), thereby skewing this type of estimation towards a lower frequency.

4. The authors state in their response to a critique of FISH, “It is clear from the FISH analysis that our staining is very specific as the positive bacteria are in a dense aggregation of bacteria”. This statement is not justified unless they happen to know that the surrounding bacteria have very similar (closely related) 16S rRNA sequences. All we know from this photo is that there are some bacteria that do not stain with the SFB probe. This is not a convincing argument for specificity. To evaluate specificity, the authors should obtain some of the *Clostridia* shown in Fig S4, or the most closely related bacteria to SFB that are available in culture and test these bacteria under the same conditions as used for the SFB staining.

To address the reviewer’s concerns, we repeated the fluorescent *in situ* hybridization using mouse SFB, purified from monoclonized mice, and a reference *Clostridium* of Fig. S4, *C. butyricum*, grown under anaerobic conditions in TY medium with glucose. We observed, under identical treatment and imaging conditions, equivalent staining with the EUB338 probe (A555-conjugated), but staining only of SFB with the SFB-specific probe (A488-conjugated). A zoom to unicellular IOs is provided in the mouse SFB panel for clarity. Notably, mouse SFB was used in this control as the 16S rRNA gene sequence complimentary to the SFB-specific probe is identical in all the SFB reference 16S rRNA gene sequences (Fig. S3). We thank Bruno Dupuis at Institut Pasteur for his kind culturing and sharing of the *C. butyricum* strain.

C. butyricum

Mouse SFB

Other minor changes:

Fig. 1b. To avoid confusion regarding the low nucleotide identities for 1470 bp long SFB 16S rRNA gene sequences when compared to SFB 16S rRNA gene sequences that do not include the V5-V9 regions (Shorebird, Mackarel, Pinfish, and Macaque), we have changed the graph to a comparison of the 1366 bp sequence and include in the comparison only those reference SFB 16S rRNA gene sequences that have the full 1366 bp length. We also increased the number of comparisons and now show a representative from each major grouping. This change has no impact on any other analysis or conclusions.

To avoid redundancy, we removed the 1366 bp 16S rRNA gene sequence comparison of Fig. S6 but instead now included a comparison of the V1-V4 region, which is present in all of the reference SFB 16S rRNA genes.

During this revision, some nucleotide sequence identity values were found to be erroneously transferred and therefore corrected in Suppl Table 1a-c. This has had no impact on any conclusions or analysis other than minor adjustments to the V1-V2 16S rRNA gene sequence nucleotide identity values plotted in Fig. S6 and Fig. 4a. In addition, the percent nucleotide identity values between Chicken-SFB-UK and Dog-SFB-US and SeaLion-SFB-US in the 1470 bp and 1366 bp analysis have also been corrected. This has had no impact on any conclusions or analysis.

The layout of Fig. 2 was slightly modified to increase the readability of the different panels.

Change of “ORFs” to “genes” in figure legend Fig. S13c for better consistency with Fig. 2f labelling.

REFERENCES

1. C. L. Thompson, R. Vier, A. Mikaelyan, T. Wienemann, A. Brune, *Environ Microbiol.* **14**, 1454–65 (2012).
2. P.-P. Grassé, *Ann. Parasitol. Hum. Comparée.* **3**, 343–348 (1925).
3. T. Kuwahara *et al.*, *DNA Res.* **18**, 291–303 (2011).
4. T. Prakash *et al.*, *Cell Host Microbe.* **10**, 273–84 (2011).
5. A. Bolotin *et al.*, *Genome Announc.* **2**, e00705-14 (2014).
6. H. Jonsson, L. W. Hugerth, J. Sundh, E. Lundin, A. F. Andersson, *Commun. Biol.* **3**, 485 (2020).
7. G. A. Hedblom, K. Dev, S. D. Bowden, D. J. Baumler, *BMC Genomics.* **23**, 659 (2022).
8. M. J. Pallen *et al.*, *Access Microbiol.*, in press, doi:10.1099/acmi.0.000910.v3.
9. R. Gilroy *et al.*, *PeerJ.* **9**, e10941 (2021).
10. D. J. P. Ferguson, A. Birch-Andersen, *Acta Pathol. Microbiol. Scand. Sect. B Microbiol.* **87B**, 247–252 (1979).
11. D. G. Chase, S. L. Erlandsen, *J Bacteriol.* **127**, 572–83 (1976).